# Observation of first- and second-order dissipative phase transitions in a two-photon driven Kerr resonator

Guillaume Beaulieu [1,2,9], Fabrizio Minganti [2,3,8,9], Simone Frasca [1,2], Vincenzo Savona [2,3], Simone Felicetti [4,5], Roberto Di Candia [6,7] & Pasquale Scarlino [1,2] ✉

In open quantum systems, dissipative phase transitions (DPTs) emerge from the interplay between unitary evolution, drive, and dissipation. While second-order DPTs have been predominantly investigated theoretically, first-order DPTs have been observed in single-photon-driven Kerr resonators. We present here an experimental and theoretical analysis of both first and second-order DPTs in a two-photon-driven superconducting Kerr resonator. We characterize the steady state at the critical points, showing squeezing below vacuum and the coexistence of phases with different photon numbers. Through time resolved measurements, we study the dynamics across the critical points and observe hysteresis cycles at the first-order DPT and spontaneous symmetry breaking at the second-order DPT. Extracting the timescales of the critical phenomena reveals slowing down across five orders of magnitude when scaling towards the thermodynamic limit. Our results showcase the engineering of criticality in superconducting circuits, advancing the use of parametric resonators for critically-enhanced quantum information applications.

Dissipative phase transitions (DPTs) are critical phenomena in which the steady state of the system—or an observable associated with it (e.g., the order parameter)—changes non-analytically upon an infinitesimal change in a control parameter (see Fig. 1a)[1–4]. DPTs extend the concepts of quantum and thermal phase transitions to systems out of their thermal equilibrium and placed in interaction with an environment[2,3]. The investigation of DPTs is of paramount importance given their occurrence in various physical systems, spanning the fields of quantum optics[5,6], condensed matter[7,8], and quantum information and technology[9–11]. Therefore, the lack of established extremal principles to describe the steady states associated with DPTs (such as the minimization of thermodynamic potentials)

calls for an effort to understand and characterize these critical phenomena[12].

DPTs can be either of first or second-order. First-order DPTs are characterized by a jump in the steady state and order parameter, together with phase coexistence, metastability, and hysteresis (see Fig. 1a–c)[1,2]. Other signatures that can accompany first-order DPTs include photon bunching and interference effects[13,14]. First-order DPTs have been observed experimentally in several systems, including trapped ions[15], ultracold bosonic gasses[16], nonlinear photonic or polaritonic modes[7,8,17], and circuit QED platforms[6,14,18,19]. Second-order DPTs are characterized by symmetries and their spontaneous breaking, and display a continuous but non-differentiable steady state and

[1]Hybrid Quantum Circuit Laboratory (HQC), Institute of Physics, École Polytéchnique Fédérale de Lausanne (EPFL), Lausanne, Switzerland. [2]Center for Quantum Science and Engineering, École Polytechnique Fédérale de Lausanne (EPFL), Lausanne, Switzerland. [3]Laboratory of Theoretical Physics of Nano-systems (LTPN), Institute of Physics, École Polytechnique Fédérale de Lausanne (EPFL), Lausanne, Switzerland. [4]Institute for Complex Systems, National Research Council (ISC-CNR), Rome, Italy. [5]Physics Department, Sapienza University, Rome, Italy. [6]Department of Information and Communications Engineering, Aalto University, Espoo, Finland. [7]Dipartimento di Fisica, Università degli Studi di Pavia, Pavia, Italy. [8]Present address: Alice & Bob, Paris, France. [9]These authors contributed equally: Guillaume Beaulieu, Fabrizio Minganti. ✉e-mail: pasquale.scarlino@epfl.ch

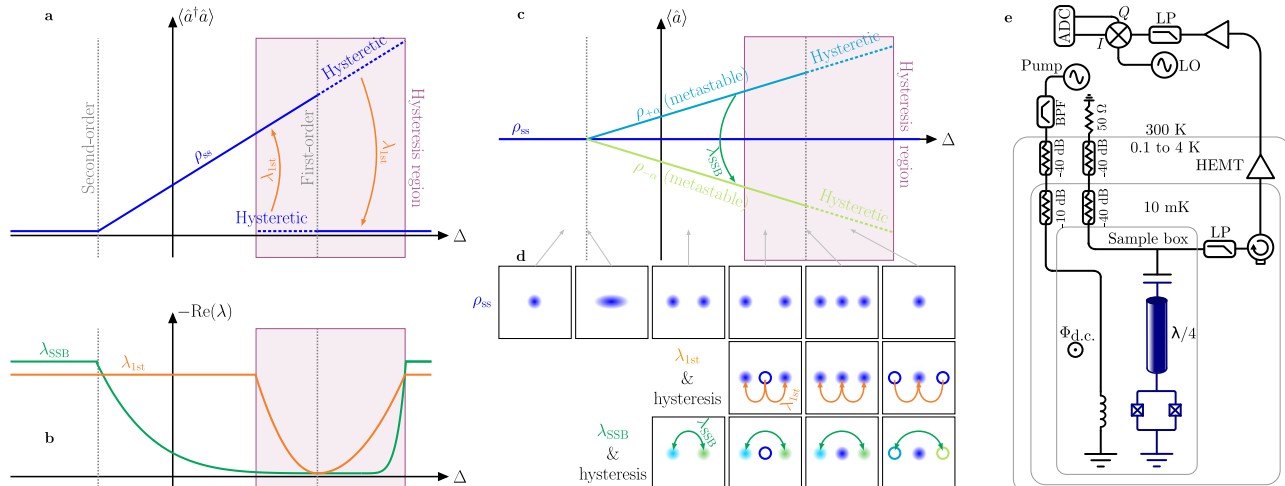

**Fig. 1 | Sketch of the theory of dissipative phase transitions and of the experimental set-up. a** Illustration of dissipative phase transitions (DPTs) according to Ref. 1 Sweeping a the pump-resonator detuning Δ, the photons in the resonator $\langle \hat{a}^{\dagger} \hat{a} \rangle$ (blue curve) changes discontinuously (first-order DPT), or continuously with non-continuous derivative (second-order DPT). The blue dashed lines indicate the metastable states associated with hysteresis across the first-order DPT, and the purple rectangle marks the hysteresis region. **b** The Liouvillian gaps $\lambda_{SSB}$ in green ($\lambda_{1st}$ in orange) associated with the second-order (first-order) DPT. **c** Throughout the detuning sweep $\langle \hat{a} \rangle = 0$ (blue curve). However, after second-order transition, the states $\rho_{\pm\alpha}$ shown in light blue and green, become metastable, with switching rate $\lambda_{SSB}$. The dashed lines indicate the metastable states associated with hysteresis across the first-order DPT. **d** Phase-space-like representation of the system across the DPTs. The top, middle, and bottom rows respectively represent the steady state (blue), the metastable state associated with $\lambda_{1st}$ (hollow blue), and

the metastable states associated with $\lambda_{SSB}$. The arrows within each panel indicate the decay of an initial state towards the steady state. The green arrows represent the decay of a non-symmetric state at a rate $\lambda_{SSB}$. The orange arrows are associated with the metastable state of the first-order DPT, decaying at a rate $\lambda_{1st}$. **e** Schematic illustrating the device and the experimental setup. The device is a hanger-type $\lambda/4$ coplanar waveguide resonator. The right side of the feedline is used to collect the emitted signal via heterodyne detection, whereas the left side is only used for spectroscopy measurements to extract the device parameters and is otherwise terminated by 50 Ω (see Supplementary Note 2). The other side of the cavity is terminated to ground via a SQUID. A magnetic field is applied through the SQUID, tuning both the resonance frequency and the Kerr nonlinearity. A second waveguide, inductively coupled to the SQUID, is used to supply a coherent pump tone around twice the resonant frequency of the cavity ($\omega_p \simeq 2\omega_r$). The pump results in a two-photon drive for the cavity[46,47].

order parameter as illustrated in Fig. 1a[1]. As such, they present a jump in the derivative of the order parameter, which requires an exceptional degree of controllability of the system to be observed[20,21]. The peculiar characteristics of second-order DPTs are predicted to enhance efficient encoding of quantum information[9] and bring advantageous metrological properties[10,22–27]. These predictions further motivate the interest in an experimental characterization of the static and dynamical properties of second-order DPTs.

Critical phenomena are commonly studied in many-body systems in the thermodynamic limit, where the number of constituents asymptotically diverges. However, quantum phase transitions can also take place in finite-component systems, where the thermodynamic limit corresponds to a rescaling of the system parameters[28–31]. A preeminent role in the study of finite-component first-order DPTs has been played by nonlinear quantum-optical oscillators[7,8,14,18]. These single-photon-driven Kerr resonators, however, cannot display second-order DPTs because the Hamiltonian term describing the drive is not invariant under the action of any symmetry group[1].

Consequently, to experimentally study the properties of both first- and second-order DPTs in such finite-component systems, it is necessary to engineer the drives and dissipative processes to ensure an underlying symmetry of the system. The parametrically-driven oscillators fulfill this symmetry requirement. These resonators have been the subject of extensive research, exploring their properties in both classical[21,32,33] and quantum configurations[13,34–38], as well as effective models connecting the two[12,20]. Notably, the two-photon driven-dissipative Kerr resonators are used to, e.g., create, stabilize, and manipulate photonic Schrödinger cat states in superconducting circuits[39,40], which have been proposed as fundamental components of quantum computing devices[41]. This shows that superconducting circuits[42] offer the necessary level of control to engineer such

processes[43–45], while also allowing the parameter rescaling required to witness finite-component phase transitions.

In this work, we use a two-photon driven superconducting Kerr resonator and conduct a thorough experimental analysis of both its first- and second-order DPTs. As a first step, we scale the system towards the thermodynamic limit and analyze its steady-state properties. We show squeezing below vacuum at the second-order DPT and observe the coexistence of multiple metastable states in the vicinity of the first-order DPT. Then, we focus on the dynamical properties associated with both transitions by probing the system dynamics through time-resolved measurements. We analyze the data with novel theoretical tools, based on quantum trajectories and Liouvillian spectral theory, and extract the characteristic timescales. From this analysis, we characterize the metastable states and quantify the critical slowing down of the two DPTs.

## Results

### Device and model

The device, shown in Fig. 1d, is a superconducting cavity made nonlinear by terminating one end to ground via a superconducting quantum interference device (SQUID). A two-photon, i.e., parametric, drive is applied to the cavity by modulating the magnetic flux through the SQUID at nearly twice the resonance frequency of the cavity[46–48]. The emitted signal is collected through a feedline coupled to the other end of the cavity, then filtered and amplified with a total gain $\mathcal{G}$ before being measured. Both signal quadratures are acquired using time-resolved heterodyne detection. This system is modeled by the Hamiltonian

$$\hat{H}/\hbar = \Delta \hat{a}^{\dagger} \hat{a} + \frac{U}{2} \hat{a}^{\dagger} \hat{a}^{\dagger} \hat{a} \hat{a} + \frac{G}{2} \left( \hat{a}^{\dagger} \hat{a}^{\dagger} + \hat{a} \hat{a} \right), \qquad (1)$$

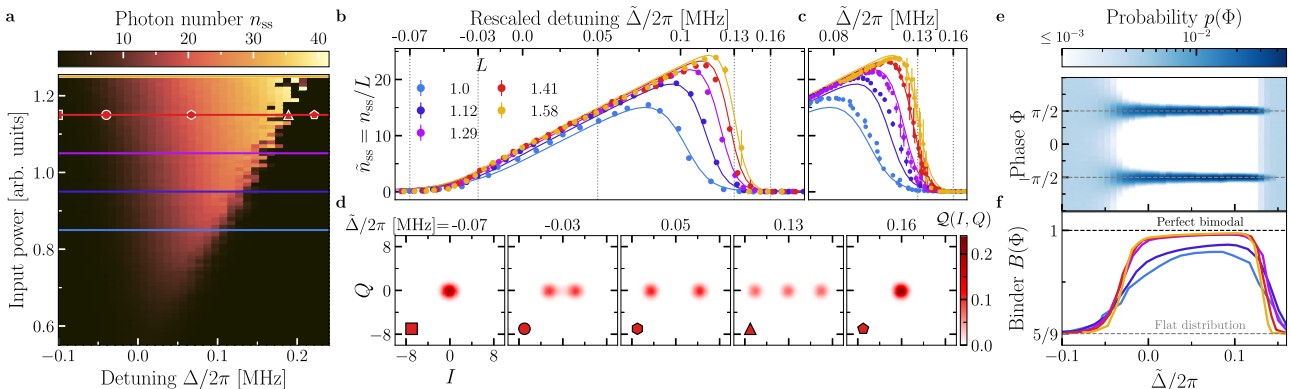

**Fig. 2 | Characterization of the steady state. a** Phase diagram showing the number of photons in the resonator as a function of the detuning $\Delta$ and input power, obtained by heterodyne detection of the emitted field. The three phases are indicated by: (i) square marker (the vacuum at negative detuning); (ii) hexagon marker (the bright phase); (iii) pentagon marker (the vacuum at positive detuning). The passage between these phases is accompanied by a second- [(i)→(ii), circle marker] and first-order DPTs [(ii)→(iii), triangle marker]. **b** Rescaled number of photons $\tilde{n}_{ss} = n_{ss}/L$ as a function of the rescaled detuning $\tilde{\Delta} = \Delta/L$ and rescaled drive $G = \bar{G}L$ for increasing scaling parameter $L$, with $\bar{G} = 65.5$ KHz (see also text and Methods for details). Circles indicate the experimental data, and solid lines are obtained from the numerical simulation of Eq. (2). The emergent discontinuities at negative and positive detuning with increasing $L$ signal the presence of a second- and first-order DPT in the thermodynamic limit, respectively. **c** Higher-resolution characterization of the abrupt change in $\tilde{n}_{ss}$ across the first-order DPT. The error bars correspond to the standard deviation over 4 experiments. **d** Husimi-$Q$ function estimated through heterodyne detection. The markers correspond to those in (**a**), and the values of $\bar{\Delta}$ corresponds to the vertical dotted gray lines in (**b**). **e** Histogram of the measured phase $\Phi$ for $L = 1.41$. **f** Bimodality coefficient (i.e., Binder cumulant) $B(\Phi)$, defined in the main text, as a function of rescaled detuning $\tilde{\Delta}$ for increasing scaling parameter $L$. Crossing of the cumulant corresponds to the critical points.

where $\hat{a}$ is the photon annihilation operator, $\Delta = \omega_r - \frac{\omega_p}{2}$ is the pump-to-cavity detuning, and $G$ is the two-photon drive field amplitude. In this study, we use $\Delta$ as the control parameter across the transition [see $\zeta$ in Fig. 1a–c]. Since the system interacts with the feedline, fluxline, and other uncontrolled bath degrees of freedom, its evolution is modeled via the Lindblad master equation

$$\frac{\partial \rho}{\partial t} = -\mathcal{L}\rho = -\frac{i}{\hbar}[\hat{H}, \rho] + \kappa(n_{th}+1)\mathcal{D}[\hat{a}]\rho \\ + \kappa n_{th}\mathcal{D}[\hat{a}^\dagger]\rho + \kappa_\phi \mathcal{D}[\hat{a}^\dagger\hat{a}]\rho + \kappa_2 \mathcal{D}[\hat{a}^2]\rho, \quad (2)$$

where $\mathcal{L}$ is the Liouvillian superoperator, whose spectrum is key in characterizing DPTs[1,2]. The dissipators are defined as $\mathcal{D}[\hat{A}]\rho = \hat{A}\rho\hat{A}^\dagger - \{\hat{A}^\dagger\hat{A}, \rho\}/2$, and the rates $\kappa$, $\kappa_\phi$, and $\kappa_2$ are associated with the total photon loss, dephasing, and two-photon loss, respectively. Finally, $n_{th}$ is the thermal photon number. Throughout the experiment, the resonator frequency is fixed at $\omega_r/2\pi = 4.3497$ GHz, corresponding to a Kerr nonlinearity of $U/2\pi = -7$ kHz, and $\kappa/2\pi = 77$ kHz. The other parameters of the experiment are theoretically estimated to be $\kappa_\phi/2\pi = 4.4$ kHz, $\kappa_2/2\pi = 78$ Hz, and $n_{th} = 0.055$. As described in Methods, the value of $G$ is measured and then refined through a theoretical estimation. The methods used for determining these parameters are described in Supplementary Note 5.

## Steady-state properties and phase diagram

We begin our study by characterizing the system steady-state $\rho_{ss}$, formally defined by $\partial_t\rho_{ss} = 0$. To this end, we initialize the system in the vacuum state, then switch on the two-photon drive $G$ at frequency $\omega_p$, and start acquiring the signal quadratures at frequency $\omega_p/2$ after a waiting time $\tau_{wait}$. Knowing the output gain $\mathcal{G}$ and the total loss rate $\kappa$, we rescale the signal quadratures measured at room temperature to obtain the field quadratures $I$ and $Q$ of the cavity, convoluted with the noise of the amplification chain. The amplifier noise is then removed when calculating the expectation value of the intracavity field moments ($\langle\hat{a}^{\dagger k}\hat{a}^l\rangle$) (see Methods and Supplementary Note 2). In Fig. 2a, $n_{ss} = \langle\hat{a}^\dagger\hat{a}\rangle_{ss}$ is reported as a function of $\Delta$ and input power, both tunable on demand. We stress that the required wait time $\tau_{wait}$ to reach the steady state can be orders of magnitude longer than the typical photon-lifetime $1/\kappa \sim 2$ μs (Supplementary Fig. 1), a clear indication of

critical slowing down[49]. From Fig. 2a, we distinguish three regimes: (i) at large negative detuning, the system is in the vacuum state; (ii) the system transitions from the vacuum state to a bright state without discontinuity. This happens at $\Delta \approx -G$ (see "Methods"); (iii) at large positive detuning, $n_{ss}$ falls abruptly from the high population phase to the vacuum[46].

To better characterize these regimes, we perform a rescaling of the parameters: $G = \bar{G}L$ and $\Delta = \tilde{\Delta}L$ (see "Methods" and refs. 1,50). The rescaling parameter is defined such that $L = 1$ corresponds to an estimated pump amplitude of $G = 65.5$ kHz. In the experiment, the rescaling is achieved by increasing the two-photon drive amplitude and correspondingly spanning a larger region of detuning. In Fig. 2b, we compare the curves of the re-scaled steady state intracavity population $\tilde{n}_{ss} = n_{ss}/L$ for the same $\bar{G}$ and range of $\tilde{\Delta}$, while increasing $L$. The solid lines in the figure are the theoretical curves obtained by numerical simulation of the model in Eq. (2) and show an excellent agreement with the experimental data. As $L$ increases, the emergence of a continuous but non-differentiable change in the photon number at negative detuning, and a discontinuous jump at positive detuning can be observed. A zoom on each of these regions is respectively shown in Figs. 2c and 3a. These abrupt changes are the fingerprints of second- and first-order DPTs, respectively, as also depicted in Fig. 1a. To better characterize the second-order transition, in Fig. 3b, c we show the first- and second derivative of the photon number with respect to detuning. Despite a continuous change in the photon number, the second order derivative increases with $L$ indicating that a second-order discontinuity is emerging at $\tilde{\Delta}/2\pi \simeq -0.07$ MHz. This value of the critical detuning is consistent with the value of $\Delta$ at which $n_{ss} = 0$ according to Eq. (6).

The histograms of the measured $I$ and $Q$ quadratures—i.e., the Husimi functions of the steady state convoluted with the noise of the amplifier—are plotted Fig. 2d for the three regimes mentioned above and near the critical points. As the detuning increases across the second-order DPT, the vacuum separates into two coherent-like states with opposite phases. It will be demonstrated below that at the second-order critical point, the vacuum is, in fact, squeezed. On the other hand, at the first-order critical point, two coherent-like states with a large photon number coexist with a vacuum-like state. In Supplementary Fig. 2, we also show the theoretical Wigner function based on the fitted experimental parameters, which confirms this interpretation.

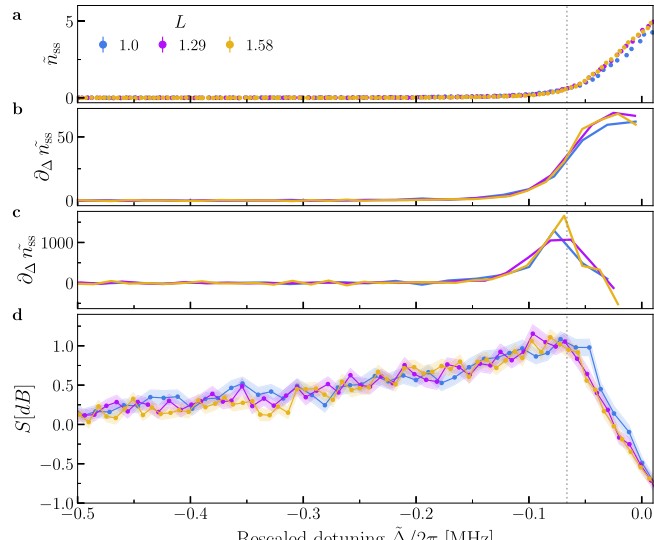

**Fig. 3 | Squeezing at the second-order DPT.** Rescaled photon number (**a**), its first (**b**), and second derivatives (**c**) calculated from the experimental data as a function of detuning. **d** The squeezing level $S = -10\log(\Delta x_\phi^2/0.5)$ evaluated across the second-order DPT. Notice that the maximum is in the vicinity of the critical point indicated by the maximum of the second derivative of the photon number in (**c**). The vertical dotted line indicates the expected maximum of the squeezing parameter obtained by numerical simulation of the steady state. The colored shaded region in (**c**) and the error bars in **a** represent the standard deviation calculated from 100 bootstrapped dataset of the measured data (see "Method for details).

We introduce here a new procedure to characterize criticality, rooted in the theory of quantum trajectories and DPTs (see Supplementary Note 6)[51]. The critical nature of the system can also be observed by considering the "conjugate" parameter of the photon number, i.e., the system's phase $\Phi(t) = \arg(I(t) + iQ(t))$. The probability distribution $p(\Phi)$ is reconstructed from the histogram of $I(t)$ and $Q(t)$. Figure 2e shows histograms of $\Phi$ as a function of the rescaled detuning for a fixed value $L$. While $\Phi$ is uniformly distributed in the vacuum phase, it displays two narrow peaks in the bright phase, corresponding to the coherent-like states in Fig. 2d. We report the bimodality coefficient (binder cumulant[52]) $B(\Phi) \equiv m_2^2/m_4$ in Fig. 2f, where the $j$th moments are $m_j = \int_{-\pi}^{\pi} p(\Phi)\Phi^j\,d\Phi$ with $p(\Phi)$ being the probability distribution of $\Phi$. This coefficient is a statistical measure used in the study of phase transitions and critical phenomena. It provides information about the probability distribution of an order parameter, capturing the "degree of order." Binder cumulants become independent of the lattice size around the transition (for large enough $L$), and the crossing of the cumulants provides an accurate determination of the critical point. This quantity, originally used for the study of paramagnetic-to-ferromagnetic second-order transitions has recently been proposed as a tool to study DPTs[51,53]. In the present case, the transitions are captured by a change from $B(\Phi) = 5/9$ (flat distribution) to $B(\Phi) \simeq 1$ (bimodality) that is smooth at the second-order and abrupt at the first-order critical points, thus reinforcing the evidence for DPTs.

We now examine the steady-state properties near the second-order transition in more detail. At the critical point of the second-order DPT, the steady state is squeezed below the vacuum. We define the squeezing level $S$ in decibels as the maximum of $S = -10\log(\Delta x_\phi^2/0.5)$, where $\Delta x_\phi^2 \equiv \langle \hat{x}_\phi^2 \rangle - \langle \hat{x}_\phi \rangle^2$ is the variance of the quadrature $\hat{x}_\phi = (\hat{a}e^{-i\phi} + \hat{a}^\dagger e^{i\phi})/\sqrt{2}$, spanning all possible $\phi$. We note that for $\Delta x_\phi^2 < 0.5$, the state is squeezed below the vacuum and $S$ is positive. Figure 3d shows the squeezing level as a function of the detuning. At large negative detuning $S$ equals zero ($\Delta x_\phi^2 = 0.5$) because the steady state is the vacuum. The maximum of squeezing level is above zero

(squeezing below vacuum). The position of this maximum closely aligns with the second-order critical point, i.e., the maximum of the second derivative of the photon number, as shown in Fig. 3a–c. The noise removal procedure followed to calculate the squeezing level is detailed in Methods and Supplementary Note 2.

## Dynamical properties of the second-order transition

Having characterized the steady state critical properties, we now focus on the dynamical properties. A distinctive feature of second-order DPTs is spontaneous symmetry breaking (SSB) (see Fig. 1a–c). The Eq. (2) is invariant under the transformation $\hat{a} \rightarrow -\hat{a}$. This weak $Z_2$ symmetry[54–56] imposes constraints on steady state of the system (see Methods). Namely, when collecting the signal, for each measured quadrature $(I, Q)$, it must be equally probable to measure $(-I, -Q)$. As such, the presence of a $Z_2$ symmetry ensures that $\langle \hat{a} \rangle = 0$. This implies that, although a single measure of $I$ and $Q$ can be different from zero, their average over many measurements must cancel out. SSB is defined as the presence of states $\rho_{\text{SSB}}^{\pm}$ that, despite being stationary, do not respect the previous condition[1,57]. These states can only emerge in the thermodynamic limit $L \rightarrow \infty$, or for classical analogs where the number of excitations can be taken to be infinite[21,58]. At finite values of $L$, however, the emergence of SSB is signaled by critical slowing down: $\rho_{\text{SSB}}^{\pm}$ are not stationary, but they decay towards $\rho_{\text{ss}}$ at a rate $\lambda_{\text{SSB}} \ll 1/\kappa^1$, as sketched in Fig. 1c. For the two-photon driven Kerr resonator model, $\rho_{\text{SSB}}^{\pm} \simeq |\pm\alpha\rangle\langle\pm\alpha|$ and $\rho_{\text{ss}} = (\rho_{\text{SSB}}^+ + \rho_{\text{SSB}}^-)/2$, where $|\alpha\rangle$ is a coherent state[13]. Theoretically, this rate corresponds to one of the Liouvillian eigenvalues (see "Methods" and Supplementary Note 5).

The continuous measurement traces shown in Fig. 4a display jumps between the states $\rho_{\text{SSB}}^{\pm}$. Notice how the observed rate of phase jumps is significantly larger than the typical photon lifetime $1/\kappa \sim 2\,\mu$s and further decreases with increasing value of $L$ (see Supplementary Fig. 5).

In order to quantify the critical slowing down, we have derived a method to extract $\lambda_{\text{SSB}}$ from the steady-state auto-correlation function. As proven in the Methods and Supplementary Note 6, in the limit in which critical slowing down takes place, one has

$$C_{\text{ss}}(t) = \lim_{\tau, T \to \infty} \frac{1}{T} \int_\tau^{\tau+T} \frac{I(\tau')I(t+\tau')}{I^2(\tau')} d\tau' \simeq \exp\{-\lambda_{\text{SSB}}t\} \quad (3)$$

where $I(\tau)$ is the measured quadrature at time $\tau$, integrated over a measurement time. A quadrature obtained from a single measurement trace is shown in Fig. 4a. In the experiment, given the discrete nature of the signal, $C_{\text{ss}}(t)$ is calculated by averaging over multiple times $\tau$ the product of $I(\tau)$ and $I(t + \tau)$. We plot the autocorrelation functions and their fit according to Eq. (3) in Fig. 4b. From this, we finally obtain $\lambda_{\text{SSB}}$, shown in Fig. 4c, as a function of the rescaled detuning $\tilde{\Delta}$ and for various $L$.

Remarkably, in our measurements, $\lambda_{\text{SSB}}$ spans five orders of magnitude. By fitting the minimum of $\lambda_{\text{SSB}}$ and plotting it as a function of $L$ (see inset of Fig. 4c), we clearly see an exponential behavior, characteristic of finite-component phase transitions, indicating the presence of a true SSB in the thermodynamic limit $L \rightarrow \infty$. The numerical simulations for $\lambda_{\text{SSB}}$ closely resemble the experimental data. It is worth emphasizing that the Liouvillian eigenvalues associated with the DPTs strongly depend on the model parameters. This is shown in Supplementary Fig. 3. Therefore, the validity of the model in Eq. (2) and of the chosen parameters is confirmed.

Finally, notice that $\lambda_{\text{SSB}}$ is associated to a bit-flip error rate in Kerr and dissipative cat qubits[40,57,59,60]. As our results demonstrate, $\lambda_{\text{SSB}}$ can be reduced by changing the detuning. Moreover, we see that $\lambda_{\text{SSB}}(\Delta, L) \propto e^{\alpha(\Delta)L}$, where $\alpha(\Delta)$ strongly depends on $\Delta$, as also shown in refs. 9,61,62. This is also highlighted in greater details in the Supplementary Fig. 5. These observations demonstrate how criticality could be exploited for quantum information processing[63].

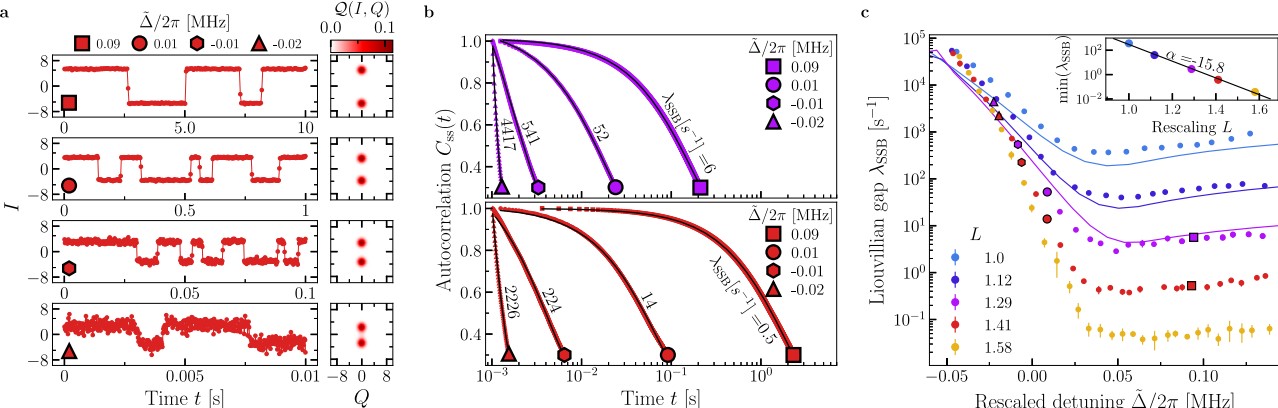

**Fig. 4 | Analysis of the second-order DPT. a** A segment of the measured quadrature. As a function of time, we plot $I(t)$ for $L = 1.41$ at various rescaled detunings $\tilde{\Delta} = \Delta/L$, indicated by the marker in each panel. Random jumps between two opposite values of the quadrature occur as time passes. These correspond to the switches between the states $\rho_{SSB}^+$ and $\rho_{SSB}^-$, as described in the main text. Using the entire collected signal, we recover a bimodal Husimi function shown on the right. **b** The autocorrelation function $C_{ss}(t)$ (see Eq. (3) and "Methods"), obtained from a single measurement trace as those shown in panel **a**. The markers at the end of the curves represent the values of $\tilde{\Delta}$, and the colors indicate the scaling parameter ($L = 1.29$: purple, $L = 1.41$: red). The Liouvillian gap can be extracted from fitting

these curves using Eq. (3). The fits are represented by the black lines. **c** The fitted Liouvillian gap $\lambda_{SSB}$ as a function of $\tilde{\Delta}$ for different scaling parameters $L$, such that $G = \bar{G}L$ with $\bar{G} = 65.5$ kHz. Points are the experimental data, while the solid lines describe the theoretical prediction obtained by diagonalizing the Liouvillian in Eq. (2). This task could be efficiently performed up to $L = 1.29$. After this value, simulations to optimize the parameters values become unreasonably long. The error bars correspond to the standard deviation over 4 experiments. The inset shows the minimum of $\lambda_{SSB}$ as a function of the rescaling parameter $L$. The black line shows the fit of the function $\lambda_{SSB} \propto \exp(\alpha L)$ to the data.

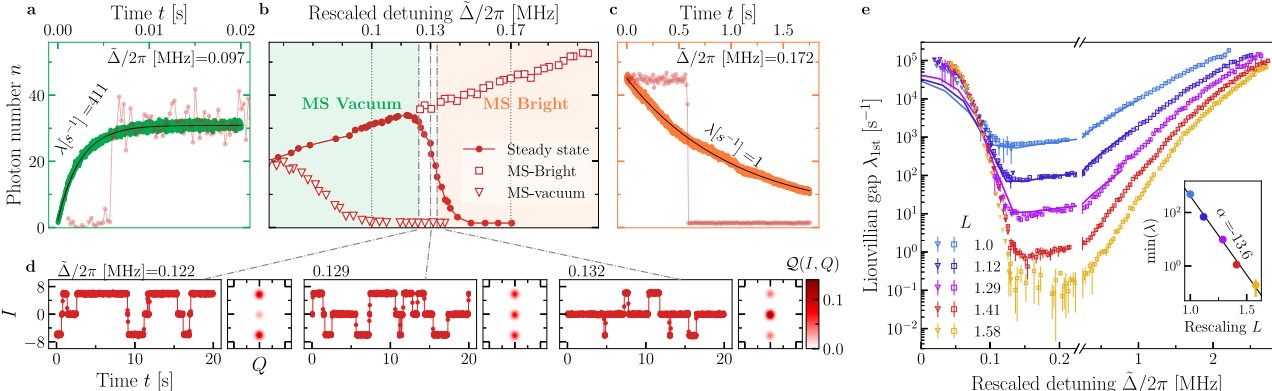

**Fig. 5 | Analysis of the first-order DPT. a–d** For $L = 1.41$, metastability around the critical detuning $\Delta_c/2\pi \approx 0.13$ MHz where the first-order transition takes place. $\Delta_c$ corresponds to the detuning for which the first-derivative of $n_{ss}$ with respect to detuning is maximal. **b** The photon number $n$ both in the steady state (circles) and in the metastable regimes (squares and triangles) as a function of detuning. The photon number in the metastable regimes has been obtained by initializing the system at $\Delta < \Delta_c$ ($\Delta > \Delta_c$) in the vacuum (in the high-population) phase and waiting for a time $1/\kappa$. **a** For $\Delta < \Delta_c$, the system is initialized in the vacuum, and it evolves towards the bright phase. The red curve is the measured photon number in a single measurement trace, while the green curve is the average over 1000 measurement traces, and is fitted by Eq. (4) (black line). **c** As in (**a**), but for $\Delta > \Delta_c$, where the system is initialized in the bright phase (see Supplementary Note 3). **d** Phase coexistence

takes place in proximity of the critical point $\Delta \simeq \Delta_c$. Once the system has reached the steady state, the signal of a single measurement trace displays random jumps between the vacuum and the bright phase. From left to right, $\Delta$ increases and the relative weights of the two phases change, as it can be observed in the Husimi functions. Note that at the time $t = 0$, the system has already reached the steady state. **e** Liouvillian gap $\lambda_{1st}$ extrapolated using Eq. (4) from data similar to those in (**a–c**). As in Fig. 4, markers indicate the experimental data, obtained by fitting the decay from either the vacuum or the bright phase towards the steady state, while the solid lines are the results of the numerical diagonalization of the Liouvillian in Eq. (2). The error bars correspond to the standard deviation over 4 experiments. The inset shows the minimum of $\lambda_{1st}$ as a function of the rescaling parameter $L$. The black line shows the fit of the function $\lambda_{1st} \propto \exp(\alpha L)$ to the data.

## Dynamical properties of the first-order transition

Similarly to the second-order DPT, criticality can only occur in the thermodynamic limit. In the case of finite $L$, however, the emergence of DPT results again in critical slowing down associated this time with a rate $\lambda_{1st}$. In particular, the photon number at a given time $t$ follows[1]

$$n(t) \simeq n_{ss} + \delta n\, e^{-\lambda_{1st} t}, \qquad (4)$$

where $\delta n$ depends on the initial state. Following this definition, we identify three regimes, summarized in Fig. 5b, in the proximity of the

critical point $\Delta_c$ of the first-order DPT: (i) $\Delta < \Delta_c$ shown in Fig. 5a; (ii) $\Delta \simeq \Delta_c$ in Fig. 5d; and (iii) $\Delta > \Delta_c$ in Fig. 5c.

In (i), the data obtained from a single measurement trace remain in the vacuum for a long time before randomly jumping to the bright phase (red curve). Once the bright phase is reached, the system never jumps back to the vacuum. Averaging over many measurement traces (the green curve) results in $n(t)$ following Eq. (4). We conclude that the steady state $\rho_{ss}$ is the bright phase, while the vacuum is metastable with lifetime $1/\lambda_{1st}$. In (ii), the measured quadrature along a measurement trace shows that the state jumps between the bright and the vacuum

phase. The relative time they spend in each of these phases determines the composition of $\rho_{ss}$. This is also evident from the Husimi functions that reflect the phase coexistence between the vacuum and the bright phase. This region of coexistence shrinks as the thermodynamic limit is approached (not shown). Finally, in (iii) the data display a jump between the bright phase and vacuum (red curve). Averaging over many measurement traces (orange curve) results in an exponential decay from $\delta_n \simeq n(t=0)$ to the vacuum following Eq. (4). In this regime, $\rho_{ss}$ is the vacuum, and $\lambda_{1st}$ describes the decay of the bright meta-stable phase.

We plot the Liouvillian gap $\lambda_{1st}$ in Fig. 5e, demonstrating the emergence of critical slowing down associated with the first-order DPT as we approach the thermodynamic limit. Notice that both data extrapolated in the regions (1) and (3) match at the critical point, confirming the theoretical prediction of ref. 1 (see also the zoom on the region in Supplementary Fig. 6). Furthermore, as shown in the inset, we also observe an exponential dependence for the minimum of $\lambda_{1st}$ with respect to the scaling parameter $L$. The numerical simulations for $\lambda_{1st}$ align well with the experimental data. We also note that, while in the previous simulations shown in Figs. 2 and 4, $\kappa_2$ played only a marginal role, it now determines the dependence of $\lambda_{1st}$ with respect to $\Delta$. This is shown in greater detail in the Supplementary Fig. 7.

As the critical region is characterized by metastable states, whose lifetime is of the order $1/\lambda_{1st}$, a hysteretic behavior in $\Delta$ is expected. Note that, although $\lambda_{SSB}$ is small in the hysteretic region, it plays no role in the determination of hysteresis, as we discuss below. As sketched in Fig. 6a, the detuning is ramped between $\Delta_{min}$ and $\Delta_{max}$, both outside the hysteresis range, according to $\Delta_\uparrow(t) = \Delta_{min} + D\,t$, with $D = (\Delta_{max} - \Delta_{min})/T$ and $\Delta_\downarrow(t) = \Delta_{max} - D\,t$, where $T$ is the sweep time. Hysteresis is immediately visible when comparing Fig. 6c, d. For a quantitative description of the effects of hysteresis, we calculate the loop area [see Fig. 6b] defined as

$$A(T) = \frac{\int_0^T dt\,\left[ n_\uparrow(t) - n_\downarrow(t) \right]}{T}. \tag{5}$$

where $n_\uparrow(t)$ [$n_\downarrow(t)$] is the average intracavity population at time $t$ along a sweep. As shown in Fig. 6e, by fitting the data by a power law, we find that $A(T) \propto T^x$.

Increasing $T$ allows the system to jump to the steady state with a higher probability, resulting in a smaller loop area. Additionally, $\lambda_{1st}$ becomes smaller with increasing parameter $L$, causing the system to remain in the metastable state for longer, thus resulting in a larger loop area. This proves that the hysteresis is indeed linked to $\lambda_{1st}$.

Finally, we do not observe hysteresis in the region where $|\lambda_{SSB}| \ll |\lambda_{1st}|$, nor do we observe two different rates of $A(T)$ that would suggest both eigenvalues are playing a role. We conclude that, as expected, only $\lambda_{1st}$ concurs with the hysteretic behavior, while $\lambda_{SSB}$ plays no role. Our analysis confirms the theoretical prediction[50] and aligns with other experimental verifications[7].

## Discussion

We have observed signatures of both first- and second-order dissipative phase transitions in a single superconducting Kerr resonator under a parametric drive. This was demonstrated through a study of both the static and dynamic properties of these finite-component DPTs, as we rescaled the system parameters towards the thermodynamic limit. The scaling was implemented by increasing the drive amplitude and correspondingly spanning a larger range of detuning. We measured the timescales characterizing the critical slowing down of both DPTs and developed an efficient method using autocorrelation measurements to extract these timescales. We framed and interpreted our results within the formalism of the Liouvillian theory.

The observation of squeezing below vacuum suggests that, at least in a specific operating region, the environment is sufficiently cold for the behavior of the resonator to not be dominated by thermal effects. This, combined with the agreement between our data and the predictions from our full quantum model, indicates that quantum fluctuations and quantum dissipative processes could play a role in the observed transitions[64,65]. However, since the device operates in a regime of weak nonlinearity ($U \ll \kappa$), the observed jumping events have clear classical analogs[32,66,67], and similar trends could be captured by a semiclassical model where various noise effects compete[68]. Consequently, while we cannot entirely rule out a classical explanation for the observed phenomena, our results showed excellent agreement with simulations based on Liouvillian theory. Our work thus proves the effectiveness of the Liouvillian theory in providing a general framework for discussing DPTs. It also highlights the need for studies of DPTs within quantum frameworks and different operating regimes to better understand the potential role of quantum processes in driving these transitions.

Although our manuscript primarily delves into the realm of fundamental physics in open systems, the control of the critical dynamics

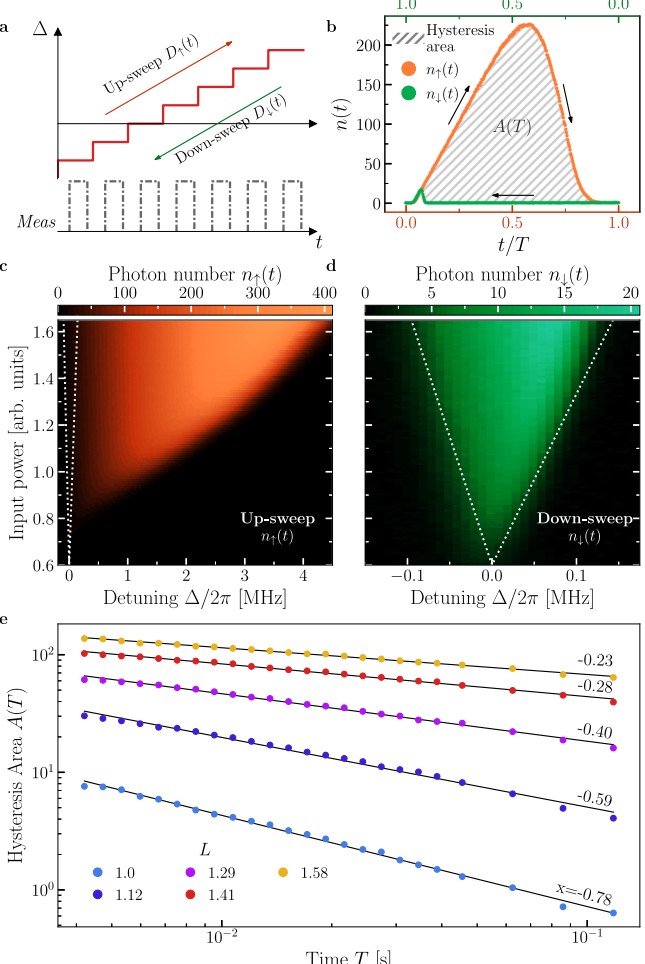

**Fig. 6 | Analysis of the hysteresis due to the first-order DPT. a** Schematic of the measurement protocol to obtain the hysteresis area. The up-sweep is $\Delta_\uparrow(t) = \Delta_{min} + D\,t$, for $D = (\Delta_{max} - \Delta_{min})/T$. Similarly, the down-sweep is $\Delta_\downarrow(t) = \Delta_{max} - D\,t$. Details of the measurement can be found in the Supplementary. **b** The area of hysteresis defined in Eq. (5) for $T = 3.5$ ms, $D/2\pi = 1000$ MHz s$^{-1}$ and $L = 1.41$. **c** Phase diagram of the photon number for an up-sweep with $D/2\pi = 1000$ MHz s$^{-1}$. **d** As in (c), but for a down-sweep. In both (**c** and **d**) the white dotted line indicates the same portion of the phase diagram. **e** As a function of $T$, the hysteresis area for various $L$. The black lines have been obtained by fitting the data with the power-law $A(T) \propto T^x$. $\Delta_{max}/2\pi = 4$ MHz and $\Delta_{min}/2\pi = -0.21$ MHz.

of a finite-component solid-state device paves the way for the technological application of critical phenomena. In particular, it serves as proof of concept towards the use of criticality and cat states for noise-biased bosonic codes[9], and it lays the foundation for the realization of dissipative critical quantum sensors[10].

## Methods

### Fabrication and setup

The device is made of a 150 nm thick aluminum layer deposited by e-beam evaporation on a 525 µm thick silicon substrate. The coplanar waveguides are fabricated by photolithography followed by wet etching. The 6.42 mm long CPW resonator is grounded through two Al/AlOx/Al Josephson junctions of area 0.56 µm², forming a SQUID.

The junctions were fabricated by e-beam lithography and deposited using double-evaporation technique inside a Plassys MEB550SL system. The participation ratio $\gamma$ of the SQUID nonlinear Josephson inductance over the bare cavity inductance is $\gamma = 3.13 \times 10^{-2}$. The finished device is bonded using Al wire to a custom printed circuit board, which is screwed to a copper mount anchored at the mixing chamber stage of a dilution refrigerator with base temperature of 10 mK. Two high-permeability magnetic shields protect the sample against external magnetic fields.

During the measurement, a NbTi coil placed underneath the sample provides a constant DC flux bias of $F = \phi_{ext}/\phi_0 = \pi/6$. Under this static field, the resonance frequency and Kerr nonlinearity are of $\omega_r/2\pi = 4.3497$ GHz and $U/2\pi = -7$ kHz. The internal and external photon loss rate, originating from the coupling to the feedline and other spurious baths are respectively of $\kappa_{ext}/2\pi = 60$ kHz and $\kappa_{int}/2\pi = 17$ kHz. All of these parameters are extracted by fitting the measured scattering coefficients using input-output relations.

A detailed description of the device, the fabrication process, the experimental setup, and the pulse sequence used in the measurements can be found in Supplementary Notes.

### Parameters estimation

In addition to the measured parameters, we need to quantify the pump amplitude $G$, the dissipative rates $\kappa_2$ and $\kappa_\phi$, and the number of thermal photons $n_{th}$ to model our system. $\kappa_2$ is a two-photon dissipation rate, arising through the same processes that convert the incoming pump tone into the two-photon drive[69]. $\kappa_\phi$ is the dephasing rate mainly due to the flux noise. To estimate these parameters, we explore the parameter space though a simulated annealing algorithm, and then search for the parameters that better fit the experimental data for photon number, $\lambda_{1st}$, and $\lambda_{SSB}$. Details on this procedure can be found in Supplementary Note 6.

When estimating $G$, its initial guess has been obtained by measuring the steady state photon number $n_{ss}$ as a function of the detuning $\Delta$. In the mean-field approximation, the stable solution for $n_{ss}$ is given by

$$n_{ss} = \frac{\sqrt{4G^2W - (UK - 2\Delta\kappa_2)^2} - \kappa_2 K - 2\Delta U}{2W}, \quad (6)$$

where $W = \kappa_2^2 + U^2$ and $K = \kappa_\phi + \kappa$. This formula simplifies to

$$n_{ss} = \frac{\Delta + \sqrt{|G|^2 - (\kappa + \kappa_\phi)^2}}{|U|} \quad (7)$$

when $\kappa_2 = 0$. We experimentally and theoretically find that such an approximation is valid far from the transition points. In a regime where $\kappa_\phi \ll \kappa$, $G$ can thus be easily estimated by extrapolating $n_{ss}(\Delta)$ to the x-intercept [$n_{ss}(\Delta) = 0$], even without knowing the Kerr nonlinearity. We find that the value of $G$ obtained via annealing simulation is within a few % of the initial guess.

### Acquisition of the signal

In the experiment, we measure $I_m$ and $Q_m$, demodulated at half of the pump frequency, of the emitted field using time-resolved heterodyne detection. Each acquired point is the integrated signal over a time interval $\tau_{int}$ varying from 2 to 50 µs, depending on the necessary time resolution. Given the timescale of the process and the desired accuracy, the data of a single measurement trace is constructed by concatenating $10^3$ to $10^7$ quadrature measurements obtained sequentially. Knowing the output gain $\mathcal{G}$ and the total loss rate $\kappa$, we rescale the measured signal quadratures $I_m$ and $Q_m$ to obtain the field quadratures $I$ and $Q$ of the cavity, convoluted with the noise of the amplification chain.

Having obtained the $I$ and $Q$ data, we can reconstruct the physics of other observables from the higher-order moments without the amplifier noise, as detailed in Supplementary Note 2. From a theoretical perspective, this procedure is equivalent to constructing the probability $p(I, Q)$ from the measured quadratures, and then computing the moments of this distribution. For instance, we estimate a generic $\langle\hat{I}\rangle = \iint dI\, dQ\, p(I, Q)\, I$. Notice that, as the measurement acquisition time is finite, the estimation of $p(I, Q)$ may be inaccurate in regimes characterized by rapid fluctuations.

Statistical error bars are obtained by taking the standard deviation over four repetitions of the experiment, with the exception of Fig. 3, where the error bars are obtained using a resampling technique, i.e., bootstrapping with replacement. In this case, the standard deviation is calculated from 100 bootstrapped datasets of the measured data. Systematic errors in the measurement mainly arise from frequency shifts, which are especially evident when measuring the photon number at the transition point. Other sources of systematic errors include the estimation of the gain (which, however, we find in good agreement with the theoretical results) and fluctuations in the pump power.

### Thermodynamic limit

The thermodynamic limit for the two-photon driven Kerr resonator has been discussed in details in ref. 13 Considers a lattice of $L$ coupled two-photon resonators, described by the Hamiltonian $\hat{H} = \sum_j \hat{H}_j + \sum_{i,j} \hat{H}_{i,j}$, with $\hat{H}_j = \Delta\hat{a}_j^\dagger\hat{a}_j + U\hat{a}_j^\dagger\hat{a}_j^\dagger\hat{a}_j\hat{a}_j/2 + G(\hat{a}_j^{\dagger 2} + \hat{a}_j\hat{a}_j)/2$ and $\hat{H}_{i,j} = J\hat{a}_i^\dagger\hat{a}_j + \text{h.c.}$. Re-writing the Hamiltonian using the Fourier modes, keeping only the mode $\hat{a}_0 = \hat{a}_{k=0}$, and fixing the Kerr nonlinearity $U$ as the unity of the model, results in

$$\hat{H}_{k=0} = (\Delta - 2J)L\,\hat{a}_0^\dagger\hat{a}_0 + \frac{U}{2}\hat{a}_0^\dagger\hat{a}_0^\dagger\hat{a}_0\hat{a}_0 + \frac{GL}{2}\left(\hat{a}_0^\dagger\hat{a}_0^\dagger + \hat{a}_0\hat{a}_0\right). \quad (8)$$

This leads, up to a shift in the detuning, to a rescaling of the single resonator Hamiltonian. Similarly, the photon loss term scales as $\kappa \rightarrow L\kappa$, while $\kappa_\phi$ and $\kappa_2$ remain unchanged. Scaling the parameter $L$ in the single resonator thus mimic the scaling of the uniform $k = 0$ mode of a lattice of $L$ resonators towards the thermodynamic limit.

In the experiment, we re-scale $\Delta$ and $G$, but not $\kappa$. As the data demonstrate, $\kappa$ plays only a marginal role in determining the proprieties of the second-order DPT and of the bright phase. Indeed, $\Delta \gg \kappa$ at the second-order critical point. However, $\kappa$ plays a more significant role in determining the critical point for the first-order DPT[9].

### Symmetry and Liouvillian eigenvalues

The equation of motion remains unchanged upon the transformation $\hat{a} \rightarrow -\hat{a}$, thereby establishing the model's invariance under the $Z_2$ symmetry. The presence of this weak $Z_2$ symmetry can be formalized through the action of the parity operator $\hat{\Pi} = \exp\{i\pi\hat{a}^\dagger\hat{a}\}$. Indeed, the steady state is such that $\rho_{ss} = \hat{\Pi}\rho_{ss}\hat{\Pi}$[54]. In a phase-space representation, this condition translates to $\rho_{ss}$ being symmetric upon a point reflection with respect to the origin, as one clearly sees in Fig. 4a where $\rho_{ss} \simeq (|\alpha\rangle\langle\alpha| + |-\alpha\rangle\langle-\alpha|)/2$.

As a consequence of the symmetry, the Liouvillian, represented as a matrix[1], has a block-diagonal structure, with two independent blocks, $\mathcal{L}_1$ and $\mathcal{L}_2$. By diagonalizing the Liouvillian, we obtain its eigenvalues $\lambda_j$ and eigenoperators $\rho_j$. In particular, the dynamics of any state can be recast as

$$\rho(t) = \rho_{ss} + \sum_j c_j e^{-\lambda_j t} \rho_j, \tag{9}$$

where the coefficients $c_j$ depend only on the initial state of the system.

Within this picture, we can directly assign a precise meaning to all the states and rates discussed in the paper. When we diagonalize $\mathcal{L}_1$, we find the eigenvalue $\lambda_0 = 0$ associated with the steady-state $\rho_{ss} \equiv \rho_0$. We then see that, in the critical region, a second eigenvalue $\lambda_{1st}$ approaches zero. Using Eq. (9), one can demonstrate that

$$n(t) = n_{ss} + \sum_j e^{-\lambda_j t} \delta n_j \xrightarrow{t \gg 1/\kappa} n_{ss} + \delta n\, e^{-\lambda_{1st} t}, \tag{10}$$

where $\delta n = c_{1st} \mathrm{Tr}(\rho_{1st} \hat{a}^\dagger \hat{a})$, $\rho_{1st}$ is the operator associated with $\lambda_{1st}$. This formula is Eq. (4), used to extrapolate the Liouvillian eigenvalue associated with the critical slowing down due to the first-order DPT.

The physics of SSB is theoretically described by $\mathcal{L}_2$. We call $\lambda_{SSB}$ the eigenvalue of $\mathcal{L}_2$ whose real part is closest to zero. An unambiguous signature of critical slowing down and SSB is then given by the observation $\lambda_{SSB} \to 0$ as $L$ is increased. $\lambda_{SSB}$ describes the rate at which $\rho_{SSB} \simeq |\pm\alpha\rangle\langle\pm\alpha|$ evolves towards $\rho_{ss}$. Experimentally, it corresponds to the rate at which the system jumps between the states of opposite phase.

### Autocorrelation function

A convenient way to extract $\lambda_{SSB}$ in the bright phase [see Fig. 2b] is to measure the system along a single quantum trajectory $|\psi_n(t)\rangle$, representing a single realization of the heterodyne measurement. The autocorrelation is defined as

$$C_n(\tau, t) = \left| \frac{\langle\psi_n(\tau)|\hat{a}|\psi_n(\tau)\rangle \langle\psi_n(t+\tau)|\hat{a}|\psi_n(t+\tau)\rangle}{\langle a \rangle_{ss}^2} \right|. \tag{11}$$

At each time $\tau$, the system is in one of the states $|\pm\alpha\rangle$, and the rate at which it jumps to the opposite state is $\lambda_{SSB}$. Averaging over several measurement traces (ideally, infinitely many) taken at a long enough time $t \gg 1/\kappa$ and for $\tau \gg 1/\kappa$, the steady state correlation function is then defined as

$$C_{ss}(t) = \sum_{n=1}^{N} \frac{C_n(\tau \gg 1/\kappa, t)}{N} \simeq \exp{-\lambda_{SSB} t}. \tag{12}$$

The last equality is proved in Supplementary Note 6, and it is rooted in a quantum trajectories interpretation of the Liouvillian dynamics, and of its symmetries. Given the ergodic nature of the system, $C_{ss}(t)$ can also be computed along a very long measurement trace $|\psi_1(t)\rangle$. In this case,

$$C_{ss}(t) = \frac{1}{T} \int_0^{T \gg 1/\kappa} dt'\, C_{1traj}(\tau + t', t) \tag{13}$$

This procedure provides the advantage of isolating the timescale of SSB. As such, this technique is a straightforward way to measure the Liouvillian eigenvalue.

### Data availability

The data used to produce the plots within this paper are available on Zenodo https://doi.org/10.5281/zenodo.12658150. All other data are available from the corresponding authors upon request.

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

## Acknowledgements

The authors thank Alberto Biella, Léo P. Peyruchat, Marco Scigliuzzo, and Gianluca Rastelli for the stimulating discussions, Alberto Mercurio for the support in developing the codes for the numerical simulations, Davide Sbroggio for helping in the fabrication process, and Vincent Jouanny, Fabian Oppliger and Franco De Palma for helping with the measurement setup.

## Author contributions

G.B. and F.M. contributed equally. F.M., R.D., S.Fe., and P.S. devised the research project. G.B., S.Fr., and P.S. designed the experiment. G.B. and S.Fr. fabricated the device. G.B. performed the measurements. F.M. numerically simulated the model. G.B. and F.M. analyzed the data. R.D. and S.Fe. derived the theory for the measurement and moment reconstruction. P.S. and V.S. supervised the experimental and theoretical parts of the project, respectively. All authors contributed to the writing of the paper.

## Funding

P.S. acknowledges support from the Swiss National Science Foundation (SNSF) through the grants Ref. No. 200021 200418 and Ref. No. 206021_205335, and from the Swiss State Secretariat for Education, Research and Innovation (SERI) under contract number REF-1131 -52105 /No SEFRI M822.00081. P.S. and V.S. acknowledge support from the EPFL Science Seed Fund 2021 and Swiss National Science Foundation project UeM019-16 - 215928. V.S. acknowledges support by the Swiss National Science Foundation through Projects No. 200020 185015 and 200020 215172. R.D. acknowledges support from the Academy of Finland, grants no. 353832 and 349199.

## Competing interests

The authors declare no competing interests.
