## [Transparent Peer Review file · Nature Communications]

Observation of first- and second-order dissipative phase transitions in a two-photon driven Kerr resonator

Corresponding Author: Mr Guillaume Beaulieu

Version 0:

Reviewer comments:

Reviewer #1

(Remarks to the Author)

In the manuscript, "Observation of first- and second-order dissipative phase transitions in a two-photon driven Kerr resonator", G. Beaulieu and F. Minganti et al. reported the signatures of the 1st- and 2nd-order dissipative phase transitions (DPTs) in a two-photon driven Kerr resonator. They perform not only steady-state but also time-domain measurements and show a good understanding of their observations with numerical simulations.

Phase transitions are a central topic in physics. The technology of quantum simulation enables an unprecedented level of control over system parameters, continuously advancing our understanding to a microscopic scale. Compared with the existing experimental studies of DPT, the novelty of the current paper is the two-photon drive. It leads to an observation of the less explored phenomenon of 2nd-order DPT. I think this paper is interesting, and it provides valuable experimental evidences for the 2nd-order DPT.

In addition to my appreciation for the novelty and interest of this work, I see a notable gap between the presented data and the authors' interpretations. On the one hand, the data quality is relatively weak, such that the actual signals of interest are often elusive. I also have several confusions regarding the authors' approach to the thermodynamic limit, and the interpretation of their data. On the other hand, the authors tend to make very strong claims with weak data support. I cannot fully agree with many of the claims, such as whether a phenomenon has been "unexplored" or a phenomenon can be seen from the data. I also cannot agree that this is a "comprehensive" study of DPT. These aspects compromise the solidity and the potential impact of this study.

My detailed comments are listed as follows:

{bf On the novelty and comprehensiveness}

\$bullet\$

Abstract, Lines 4-5: "We present here the first comprehensive experimental and theoretical analysis of both first and second-order DPTs..."\$bullet\$

Introduction, paragraph 4, lines 1-2: "Consequently, to experimentally study the unexplored properties of first- and second-order DPTs..."\$bullet\$

Discussion, paragraph 1, lines 4-5: "This was demonstrated by conducting a comprehensive study... as we rescaled the system parameters towards the thermodynamic limit."

While the investigation into the 2nd-order DPT is indeed novel, I find the term "unexplored properties" unclear. The authors have studied (i) transition speed, (ii) squeezing, (iii) critical slowing down, (iv) Liouvillian gap, and (v) power-law decay, but I believe all of them have been explored in the literature [PRL,118,\,247402\,(2017), Sci.\,Adv.\,7,\,eabe9492\,(2021), and Nat.\,Commun.\,14,\,2896\,(2023)].

Moreover, several more challenging experiments, such as (vi) coherent cancellation dip, (vii) photon bunching, and (viii) quantum state tomography, have been conducted in the literature but are not addressed in this paper [PRL118, 040402(2017), Nat.\,Phys.\,14,\,365-369\,(2018), and Nat.\,Commun.\,14,\,2896\,(2023)]. I think they might be even more interesting features of DPT from a theoretical point of view [Phys.\,Rev.\,A\,94,\,033841\,(2016), Phys.\,Rev.\,A\,93,\,033824\,

(2016), and Phys.\,Rev.\,A\,95\,012128\,(2017)].

I therefore cannot agree that this study is "comprehensive". It is not only because the more difficult measurements, (vi)-(viii), are not touched in this work, but also because the observations of (i)-(iv) seem elusive and a seeming negative result of (v) is observed. Although I agree with the authors that the signatures of DPTs are there, I am not fully convinced that the approach to the thermodynamic limit is correct.

On the approach to the thermodynamic limit

•

Introduction, paragraph 4, lines 3-5: "... it is necessary to engineer drives and dissipative processes to ensure an underlying symmetry of the system."

Results A, paragraph 3, lines 1-2: "To better characterize these regimes, we perform a rescaling of the parameters: $G = \tilde{G}L$ and $\Delta = \tilde{\Delta}L$ (see Methods)".

Methods (Thermodynamic limit), paragraph 2, line 1: "In the experiment, we rescale Δ and G , but not κ ."

The first quote actually aligns with the comment I am about to make. When approaching the thermodynamic limit, the dissipation process must be carefully controlled. However, the dissipation rate, κ , is not controlled at all throughout this study.

The authors argued in Methods that "As the data demonstrate, κ plays only a marginal role in determining the properties of the second-order DPT and of the bright phase." I may understand this argument as implying that the scaling factor, L , of this experiment varies in a very small range, so it will not lead to a very different value of κ . However, the issue emerges when the authors compare the data with different L .

Since a small range of L implies subtle differences in the data, it demands even more precise control of κ to ensure accurate conclusions. Neglecting the scaling factor of κ simply makes the "desired" result more evident, but it is artificial. On the other hand, the device used in this study actually allows for the control of κ . My question is why the authors choose not to do it? Is it because κ_{2} also varies at different biasing points? How large is the relative change of κ_{2} compared to that of κ ? I think the authors must argue very carefully why it is feasible to not control κ .

A minor complaint is that L varies only from 1 to 1.58 in this study. Ideally, we would hope it to change from 1 to ∞ . I see no obvious limit on L in this experiment. For example, the input power is only varied by 3 dB, which is far from the full range of a normal RF source. May the authors explain what limits L in this study?

Results A: Steady state properties and phase diagram

•

Fig.\,1a.

I would expect two metastable states around the 1st-order DPT, but it appears not to be the case in the schematic plot. The two metastable branches seem not to be the two metastable states around the 1st-order DPT. Why there is one metastable state and one steady state?

•

Fig.\,1c.

I am confused of the definition of the Liouvillian gap here, particularly the meaning of λ_{SSB} . If we take the usual definition of the Liouvillian gap (the minimum non-zero eigenvalue of the Liouvillian superoperator), why are there two different gaps at the same parameter?

I am trying to interpret it as that λ_{SSB} and $\lambda_{1\text{st}}$ are the 1st and 2nd smallest non-zero eigenvalues. $\lambda_{1\text{st}}$ becomes the Liouvillian gap in the region where λ_{SSB} is closed. Is this understanding correct?

In addition, why the critical point of the 2nd-order DPT is indicated at a position where the Liouvillian gap is not closed?

•

Fig.\,1d

Is the left side of the feedline left open, shorted, or terminated by a $50\ \Omega$ load in the experiment?

•

Paragraph 2, lines 7-8: "Knowing the output gain G and total loss rate κ , the field quadratures I and Q of the cavity are then reconstructed (see Methods)."

Methods (Acquisition of the signal), paragraph 1, lines 9-12: "From the measured I_m and Q_m , the quadratures of the intracavity field $I(Q)$ are obtained by removing the effect of the amplification chain and its associated noise."

Paragraph 4, lines 19-20: "The histograms of the measured I and Q quadratures, i.e., the Husimi functions of the steady

state convoluted by the noise of the amplifier, ... are plotted (in) Fig.\,2d."

By reading the Methods, the amplification noise is removed such that I and Q are the genuine intracavity field quadratures up to a scaling factor. However, by reading the last quote, it seems that I and Q are convoluted quadratures of the signal and the noise. Are the data in all the figures also not analyzed in this way? Which data in this study are measured with the described method?

I think this difference will lead to very different interpretations of the data. For example, the claim "The intracavity photon number is $n_{\text{ss}} = \langle I^2 \rangle + \langle Q^2 \rangle$." will not be correct. I point out this issue here and will comment on the specific data later.

•
Fig.\,2a.

Because $n_{\text{ss}} \neq \langle I^2 \rangle + \langle Q^2 \rangle$ if the noise is not removed, I am not convinced by the n_{ss} -axis values. I am assuming that the n_{ss} -axis is obtained by rescaling $\langle I^2 \rangle + \langle Q^2 \rangle$. It is not correct because of the convolution relation between signal and noise. I think the authors need to characterize and then subtract the noise photon number from $\langle I^2 \rangle + \langle Q^2 \rangle$ to get n_{ss} .

On the other hand, I see in the Supplementary Information that the authors have characterized the noise photon number. The question is how n_{ss} is obtained here, by scaling or subtracting the noise?

•
Fig.\,2b-c.

I am not convinced by the scaling, but let us leave this issue aside.

According to the Methods, the authors choose to rescale Δ , G , and κ (not controlled in reality) instead of U and κ_2 . I am thinking that the energy of the energy will be L times larger than the usual way of rescaling [Phys.\,Rev.\,A,94,033841,(2016)]. Thus, for comparing data with different L , shouldn't the n_{ss} -axis being n_{ss}/L^2 instead of n_{ss}/L ? Here, the physical meaning of n_{ss}/L^2 is the density of photons in an L -size system.

The authors wrote that "As L increases, the emergence of a continuous but non-differentiable change in the photon number at negative detuning, and a discontinuous jump at positive detuning can be observed [see also Fig. 2(c)]." However, I cannot agree that I see a "non-differentiable change" at $\tilde{\Delta}/2\pi = -0.04$ (MHz) nor a "discontinuous jump" at $\tilde{\Delta}/2\pi = 0.13$ (MHz) as indicated by the authors.

If we accept the scaling as it is, I would agree that the data trend in Fig. 2c indicates a discontinuous jump at $\tilde{\Delta}/2\pi = 0.13$ (MHz) at $L \rightarrow \infty$, corresponding to a 1st-order DPT. However, I cannot see the described data trend at the indicated critical point of the 2nd-order DPT. I think the authors should provide a zoom in of the data around $\tilde{\Delta}/2\pi = -0.04$ (MHz) to justify the observations of the 2nd-order DPT.

Besides a feature of "continuous but non-differentiable change", the slope should also be almost independent of L around the 2nd-order DPT. Ideally, a transition from a horizontal line to the same slope may be observed. I think an observation of this trend would be important to verify the existence of 2nd-order DPT.

•
Fig.\,2d.

Since the I , Q quadratures are convoluted signals with the noise (usually at the 10 -photon scale), I am somewhat surprised that the coherent "clouds" are well-separated at a low photon number. I guess that it is because of a very small measurement bandwidth and rescaling.

I may roughly estimate the measurement bandwidth as $B = 40$ (Hz) ~ 1 (kHz), since the Methods say that one data point corresponds to an integration time of 25 ~ 50 (μ s). In comparison, the linewidth of the nonlinear resonator is $\kappa/2\pi = 77$ (kHz). Here, $B \ll \kappa/2\pi$ so that I may naively interpret the small histogram clouds as heavily averaged results.

The very narrow measurement bandwidth is not an issue here since the expected features are in the mean value. The expected features are clear in the plot. However, I am wondering if it will cause issues in later discussions since the histogram does not correspond to the Q quasidistribution function anymore. The uncertainty relation of I and Q is averaged out.

For example, the authors wrote that "... $I(\tau)$ is the measured quadrature at time τ along a single quantum trajectory such as those shown in Fig.\,4(a)." I cannot agree that a heavily averaged measurement trace can be interpreted as a single quantum trajectory. On the other side, I understand that the usual requirement of $B > \kappa/2\pi$ is just a "safe" experimental condition. The authors interpretation may still be correct. Since the authors decide to not follow the "safe" configuration, I think that they need to justify their method.

Lastly, what are the units of the x - and y -axis for the histogram?

•

Paragraph 4, lines 19-20: "As the detuning increases across the second-order DPT, the vacuum becomes squeezed [see also Fig. 3(d)]."

Again, I cannot discern this phenomenon from the plots (I will comment on Fig. 3 later). I tend to believe that there are two closely spaced coherent clouds in the 2nd panel of Fig. 2d instead of one squeezed cloud. The claimed feature of squeezing seems elusive. Perhaps the problem can be settled by checking the Gaussianity of the histogram.

Here, the intracavity photon is almost 0 but the noise photon number is approximately 10. The latter will blur any small signal into a Gaussian distribution. Thus, strictly speaking, the authors need to do the noise removal before the Gaussianity check. However, working directly on the histogram data may also be acceptable but the claims such as "squeezing below vacuum" will not hold.

•

Fig. 2e-f:

Again, let us assume that we agree on not scaling κ . The observed jump at $\tilde{\Delta}/2\pi = 0.13$ (MHz) is a convincing signature of the 1st-order DPT. The curve also becomes more and more non-differentiable at $\tilde{\Delta}/2\pi = -0.04$ (MHz), which is a signature of the 2nd-order DPT.

However, I am puzzled by the caption: "... calculated from the probability distribution of Φ for $L = 1.41$." Shouldn't the different colors represent different values of L ? In addition, I would expect that the slope at the right side of the 2nd-order DPT is independent of L . Why it is not the case here?

{A1: Quantum nature of the transitions}

•

Paragraph 1, lines 12-15: "The position of this minimum closely aligns with the second-order critical point, i.e., the maximum of the second derivative of the photon number, as shown in Figs. 3(a-c)."

I am confused by Figs. 3a-c and the interpretation. Firstly, the 2nd-order DPT seems to happen at $\tilde{\Delta}/2\pi = -0.04$ (MHz) in Fig. 2, but it is indicated to be -0.06 (MHz) here. Why is that? Secondly, it seems that one data point in Figs. 3b-c corresponds to 5 data in Fig. 3a. Why did the authors average or throw out data? Most importantly, I think that a peak in the 2nd derivative would exist in any curves where two slopes are connected by a smooth transition. I am confused about what information do Figs. 3a-c provide for identifying the 2nd-order DPT.

•

Paragraph 6, line 4: "We define the squeezing parameter as the minimal variance $\Delta x_{\phi}^2 \equiv \langle x_{\phi}^2 \rangle - \langle x_{\phi} \rangle^2$ of the quadrature x_{ϕ} spanning all possible ϕ ."

Paragraph 1, lines 15-17: "This analysis supports the claim that quantum fluctuations play an important role at the second-order DPT." Fig. 3d:

If the amplification noise is not removed, the minimum value of Δx_{ϕ}^2 will be lower bounded by the noise photon number. The value should be much larger than $1/2$. I guess that the result, $\Delta x_{\phi}^2 = 1/2$, is obtained by an artificial rescaling the data. This is not correct because of the convolution relation. To extract the correct squeezing level, the authors need to remove the noise and perform a Gaussianity check in the meanwhile.

If we relax the criteria and look for signatures of squeezing, the noise-removal procedure may be skipped. Consequently, it is not appropriate to claim "squeezing below vacuum". I may ask the authors to show the anti-squeezing data alongside the squeezing data. The anti-squeezing data may help distinguish signatures of squeezing and two closely spaced coherent clouds. I would expect that the product of the squeezing level and the anti-squeezing level is a constant for the former but decrease with $\tilde{\Delta}/2\pi$ in the latter case.

In addition, it is confusing that the minimum variance, Δx_{ϕ}^2 , is larger than $1/2$ for $\tilde{\Delta}/2\pi > -0.04$ (MHz). It keeps increasing with $\tilde{\Delta}/2\pi$ onwards. The histogram shown in Fig. 2d and the authors' theory of two coherent clouds seem to set an upper bound of $1/2$ on the minimum variance. May the authors explain why $\Delta x_{\phi}^2 > 1/2$ can be observed?

•

Paragraph 2, lines 6-10: "From a theoretical viewpoint, in the one-photon driven resonator the presence of metastability, and thus criticality, can be argued using a semi-classical model, i.e., assuming a coherent state, and just one-photon loss."

I do not agree with this claim. Perhaps the authors can explain how the following results can be explained with a semi-classical model:

(i) The two power-law decay around the 1st-order DPT must have a quantum-mechanical origin, as argued in

Phys. Rev. A, 93, 033824, (2016) and observed in PRL, 118, 247402, (2017) and Nat. Commun., 14, 2896, (2023); (ii) The non-classical Wigner quasi-distribution function must have a quantum-mechanical origin, as argued in Phys. Rev. A, 39, 4675, (1989) and observed in Nat. Commun., 14, 2896, (2023); (iii) The so-called "coherent cancellation dip" must have a quantum-mechanical origin, as argued in J. Phys. A: Math. Gen., 13, 725-741, (1980) and observed in PRL, 118, 040402, (2017) and Nat. Commun., 14, 2896, (2023).

•

Paragraph 2, lines 14-18: "The region of metastability requires two-photon decays to be correctly captured by a coherent-state approximation. Criticality can only be theoretically obtained within a full quantum picture, as shown in Extended Data Fig. 2."

Introduction, paragraph 5, lines 8-11 that "Furthermore, we observe the coexistence of multiple metastable states in the vicinity of the first-order DPT, a feature that cannot be captured when neglecting the quantum effects of dissipation."
Results A, paragraph 1, lines 29-31: "... $U/2\pi = 7 \text{ kHz}$, and $\kappa/2\pi = 77 \text{ kHz}$. The other parameters of the experiment are theoretically estimated to be $\kappa_{\phi}/2\pi = 4.4 \text{ kHz}$, $\kappa_{\phi}/2\pi = 78 \text{ Hz}$, and $n_{\text{th}} = 0.055$ ".

Conclusion, paragraph 1, lines 16-18: "Our analysis unambiguously demonstrates the quantum nature of these critical phenomena..."

Conclusion, paragraph 2, lines 3-6: "They unambiguously demonstrate the importance of quantum processes in triggering DPTs, highlighting the necessity to study DPTs within a quantum framework."

I am puzzled why $\kappa_{\phi}/2\pi$ is being taken into consideration, given that it is 3-orders of magnitude smaller than κ . I read from the Extended Fig. 2 that a finite κ_{ϕ} is necessary to explain the metastable branches around $\tilde{\Delta}/2\pi = 0.13 \text{ MHz}$. I am not fully convinced with this claim since $\kappa_{\phi}/2\pi$ is extremely small. I think it will become more clear if the authors can plot all the three semi-classical branches in Extended Fig. 2.

Alternatively, the authors may compare the Liouvillian gaps with $\kappa_{\phi}/2\pi$ as a control parameter. My confusion is that a very small value of $\kappa_{\phi}/2\pi$ may not significantly change the Liouvillian gap, while a small Liouvillian gap around $\tilde{\Delta}/2\pi = 0.13 \text{ MHz}$ should allow metastable states [Phys. Rev. Lett., 116, 240404, (2016)].

To the best of my understanding, the non-zero rate of "two-photon loss" is regarded as unambiguously quantum nature by the authors. I do not understand this claim. I think the authors should explain in detail which specific aspects of this experiment unambiguously demonstrate the quantum nature of DPT.

Similarly, I also do not understand this claim in Discussion: "Furthermore, the results we obtained are universal, making our predictions applicable to any open quantum system." I think the authors should elaborate on the specific results and predictions they are referring to.

Results B: Dynamical properties **B1: Second-order**

•

Fig. 4c.

I find the parameter regime here puzzling. According to Figs. 2-3, the potential regime for the 2nd-order DPT is in the range $\tilde{\Delta}/2\pi = -0.07 \text{ to } 0 \text{ MHz}$. However, the shown Liouvillian gap indicates that it may happen at $\tilde{\Delta}/2\pi = 0.04 \text{ MHz}$ and onwards. Are these results consistent with each other?

In addition, I am puzzled that the Liouvillian gap, λ_{SSB} , in the range of $\tilde{\Delta}/2\pi = 0 \text{ to } 0.14 \text{ MHz}$ is different from the Liouvillian gap, $\lambda_{1\text{st}}$, in Fig. 4. Is there a subtle difference between the definitions of λ_{SSB} and $\lambda_{1\text{st}}$? Are the different values of them expected?

B2: First-order

•

Fig. 5a-c.

How is the initial "bright" state prepared in the experiment?

•

Fig. 5d.

The first data point of the left panel is different from the others. Is the initial state prepared in the bright state here while in the vacuum state for the panels?

•

Fig. 5e.

The Liouvillian gap, $\lambda_{1\text{st}}$, seems to close for a finite region ($\tilde{\Delta}/2\pi = 0.12 \text{ to } 0.2 \text{ MHz}$), which does not resemble a 1st-order DPT ($\lambda_{1\text{st}} = 0$ only at one point). Is this result expected?

•

Paragraph 6, lines 12-13: "... by fitting the data by a power law, we find that $A(T) \propto T^x$."

Paragraph 6, lines 16-18: "Our analysis confirms the theoretical prediction [48] and other experimental verifications [7]."

Fig. 6

I am also confused by the interpretation here. The main conclusion of Refs. [48,7] is the two power-law behavior, but the authors observed a single power-law behavior here. This seems to be a negative result of Refs. [48,7]. Is it because of the two-photon drive? Should we expect a two power-law for two-photon drive?

On the other hand, two power-law has been reported before [PRL, 118, 247402 (2017) and Nat. Commun., 14, 2896 (2023)]. I am confused about what new information does Fig. 6 provide compared with the existing results?

(Remarks on code availability)

The code has not been provided by the authors.

Reviewer #2

(Remarks to the Author)

The manuscript by Beaulieu, Minganti, et al. presents the experimental observation of both second- and first-order dissipative phase transitions in a two-photon driven superconducting Kerr resonator. The authors investigate the system's steady states and dynamics by adjusting the drive-cavity detuning at different input powers and monitoring the leaking cavity field in real time via heterodyne detection. They observe a continuous transition characterized by critical slowing down (CSD) of the dynamics and a squeezed vacuum state near the critical point. Additionally, a discontinuous transition is observed which is probed by a combination of CSD and hysteresis measurements between vacuum and bright phases. The experimental findings are supported by an exact master equation calculation encompassing single-, two-photon, and dephasing losses. The observed "quantum jumps" between vacuum and bright phases constitute clear experimental evidence of the significant role of quantum fluctuations in the system.

The manuscript is technically sound and well-written. The authors provide extensive supplementary material with technical details on the device, data evaluation, and theoretical model. To the best of my knowledge, this work presents the first observation of both first- and second-order dissipative transitions in (two-photon driven) Kerr resonators. Due to the pivotal role of quantum fluctuations, their observations offer a promising route for quantum-enhanced sensors, which further exploit the critical nonlinear behavior of phase transitions.

However, I have some questions and remarks before recommending publication:

Main Remarks:

1. In the main text, the authors should precisely elaborate on the role of the different energy scales (Δ , U , G) and dissipation scales (κ , κ_2 , κ_ϕ) for the observed second- and first-order transitions.

- Can the authors identify a microscopic mechanism giving rise to dissipative metastable states close to the first-order dissipative transition? Is it related to effective decay channels for collective excitations, as discussed in Refs. [4] and [20]?
- Can the authors provide an analytic estimate for the critical point of the second-order phase transition (either in the main text or the supplementary materials)? Is it solely determined by a competition between G and Δ , or do additional energy scales play a significant role?

2. The authors should provide more details on the experimental errors and uncertainties of their measurements. In particular:

- How large is the sensitivity of their heterodyne setup, i.e., how large is the minimal detectable photon number n_{ss} in the context of Fig. 2?
- Are there significant systematic errors in the (inferred) values of the quadratures I and Q ? What about the reconstructed Husimi distributions?
- The authors should provide (statistical) error bars for the minimal variance Δx^2_ϕ presented in Fig. 3(c). These can be obtained via resampling methods, such as jackknife or bootstrapping.

3. There seems to be some degree of circular argumentation when comparing the results with the Liouvillian theory in the discussion section: After stating the agreement between theory and experiment ("We framed and interpreted our results within the formalism of the Liouvillian theory [...]"), the authors state that their agreement showcases unambiguously the role of quantum fluctuations and dissipation ("Our analysis unambiguously demonstrates the quantum nature of these critical phenomena [...]"). However, they later claim that the agreement between theory and experiment also validates their Liouvillian model ("As such, our work validates the Liouvillian theory that underlies our findings."). I would strongly suggest the authors rephrase these conclusions to avoid potential misunderstanding. For example, they can elucidate the prominent role of quantum fluctuations in their system by highlighting experimental observations such as the "quantum jumps" between the different phases. To my understanding, such observations can only be captured by a full quantum mechanical description correctly incorporating quantum fluctuations and thus would escape a semiclassical approach. A suitable reference in this direction is Phys. Rev. A 102, 063702.

Additional Remarks:

Results section

- Can the authors comment on the degree of experimental tunability they have on the detuning Δ , Kerr nonlinearity U , and

two-photon drive G ?

- The authors should elaborate on the deviation of the Binder coefficient from one for small systems. Can this be attributed solely to finite-size effects or is there also a (technical) bias between the two Z_2 symmetry broken configurations observed?
- To provide better comparability with other platforms, the authors should introduce a squeezing parameter and provide a quantitative estimate of the maximal degree of squeezing in dB (decibel) in Fig. 3 (c), and its statistical uncertainty.
- The following statement is not completely self-explanatory, and merits further clarification or a suitable reference: "From a theoretical viewpoint, in the one-photon driven resonator the presence of metastability, and thus criticality, can be argued using a semiclassical model, i.e., assuming a coherent state, and just one-photon loss".
- Can the authors provide a physical argument on why the hysteresis-loop area in Fig. 6(e) decreases both with decreasing L and increasing probing time T ?

Discussion section

- Can the authors elaborate on the metrological usefulness of the observed vacuum-state squeezing and quantum jumps in the context of quantum-enhanced sensing of magnetic fields and other (technologically) relevant quantities?

(Remarks on code availability)

Reviewer #3

(Remarks to the Author)

The authors observe first- and second-order dissipative phase transitions in the squeezed Kerr resonator. The paper is easy to read and the authors are experts in their fields. The topic of the paper is also important and widely relevant.

Dissipative phase transitions are important and hard to study experimentally due to the high tunability and challenging parameter range required.

Experiments are, therefore, highly valuable and of foundational importance.

This work is interesting because it is a good example of theorists and experimentalists working closely together, probably making something larger than what can be achieved independently.

However, I am taken aback by the authors' claims regarding the experimental verification of "unambiguous" quantum effects. I find that all observations made, even if interesting and valuable in themselves, admit straightforwardly a classical explanation.

I agree that the observation may agree with a quantum Lindbladian treatment, but that is not sufficient to claim things like

"Our quantum treatment, based on the Liouvillian spectral theory, accurately reproduces the experimental data, demonstrating the quantum nature of these phenomena"

or

"The region of metastability requires two-photon decays to be correctly captured by a coherent-state approximation.

Criticality can only be theoretically obtained within a full quantum picture, as shown in Extended Data Fig. 2a"

Data Fig. 2a"

or

"Our analysis unambiguously demonstrates the quantum nature of these critical phenomena, showing that quantum fluctuations and quantum dissipative processes are the main drive of the observed transitions.

Furthermore, the results we obtained are universal, making our predictions applicable to any open quantum system."

These statements are incorrect and far too strong to be acceptable regarding the supporting evidence provided.

Decisively, the experiment is done in a classical regime where the nonlinearity is weaker than the trivial dissipation ($U \ll \kappa$). Note that the quantum nature of an experiment is washed away almost completely in almost any situation if the linewidth of the energy levels ($\sim \kappa$) is larger than the spectral (nonlinear) features ($\sim U$) of the Hamiltonian. This is the case for these experiments in particular. This is especially relevant for theoretical claims regarding quantum information, which are made in this paper.

I also think that the presence of two-photon dissipation can 1) have a perfectly classical origin in nonlinear noise mixing and 2) many spurious effects other than two-photon dissipation could cause the instability of the bright phase at large detuning (especially since there is no quantitative agreement between the quantum model and the data). Consequently, I don't think Data Fig. 2 is convincing and the claim of "unambiguity" is unsustainable.

Even if the authors would have achieved quantitative agreement between experiment and data, this would still not have been enough to support the claim that purely quantum effects were observed since alternative explanations have not been ruled out (nor discussed). The absence of quantitative agreement is in itself preoccupying.

The authors also insist in the observation of quantum jumps. The telegraphic ("jump") signal observed in this work is typical of a Brownian particle in a classical potential with more than one minimum. While I find the choice of a quantum treatment is a matter of taste, it is not at all a necessary description for this experiment.

Therefore, I find that not enough control experiments have been made to support the strong claims, and I don't think experiments in this regime can convincingly be proven to be more quantum than any other classical effect, which ultimately must admit a description in terms of open quantum systems. I do not think it is in the authors interest to insist in the quantum character of this otherwise very beautiful work.

If the authors wish to insist, the authors can try to explain (not to me, but in the paper) or define what they mean by "quantum". But I think the solution is to claim something like: "While we cannot rule out a classical explanation, there is a qualitative agreement with a simple yet rich full quantum model." This would be a fair claim.

I remain open to receiving a comeback from the authors in case they find I have missed their main point. I do wish to advance that I am not inclined to consider things like "squeezing below vacuum" to be necessarily quantum in origin. This also has a clear classical counterpart.

A last point: where is κ_2 coming from? is this engineered dissipation? Not by any additional drive, it seems. Or is it spurious? Was this explained and I missed it? What is the microscopic mechanism the authors think it can be attributed to?

(Remarks on code availability)

Version 1:

Reviewer comments:

Reviewer #1

(Remarks to the Author)

I thank the authors for clarifying their methods, and for providing more plots that help me understand their work. In the revised manuscript, the authors have weakened the strong claims that attracted most comments in the previous round of peer review. I agree with the authors that the signatures of the dissipative phase transition (DPT) are observed. The investigation of the 2nd-order DPT is indeed novel and interesting, especially the underlying symmetry and its spontaneous breaking.

I regret that my major concern regarding the thermodynamic limit is not directly addressed in the rebuttal letter. I therefore would remain my doubt on the correctness of the scaling. After reading through the revised manuscript, I think that the claims such as "fundamental role of quantum fluctuation" would still need more support from the data. More measurement data are also required to support the Z_2 symmetry story. In these regards, I maintain my opinion that the data do not fully support the conclusions. It would require a substantial revision of the story and addition of the supporting data.

My detailed comments are listed as follows:

1. Thermodynamic limit

Let me put it first that I have no doubt on whether there should exist a DPT. The signatures are there. My concerns have always been whether increasing L (i) leads us to the thermodynamic limit, or (ii) they simply confirm the signatures at different parameters.

While the approach (i) may be the only way towards a "smoking gun" proof of DPT, the approach (ii) is also acceptable depending on how the authors would interpret the data. Following (ii) but make claims as it has followed (i) is not acceptable in my point of view. In case (ii), data with different L are not comparable with each other.

The authors listed two possible ways towards the thermodynamic limit: (a) $U \rightarrow U/L$, $\kappa_2 \rightarrow \kappa_2/L$, and $\kappa_\phi \rightarrow \kappa_\phi/L$, and (b) $\Delta \rightarrow \Delta L$, $G \rightarrow GL$, and $\kappa \rightarrow \kappa L$. However, the experiment does not follow either of them. I would therefore conclude that L plays the role of (ii) and should not be interpreted as the scaling parameter.

I understand that a systemically control the parameters to follow the right scaling is difficult. But I believe that it is mandatory and it is experimentally feasible. Reasonable simplifications may circumvent the technical difficulty under careful clarification. I agree with the authors' rebuttal that if $\kappa \approx 0$, the current approach of scaling would already be correct. If the authors can carefully explain why κ can be fairly neglected in this experiment (by some transformation maybe?), I would be convinced that comparing data with different L is reasonable.

***** I did some quick calculations and have a different opinion on the two approaches, (a) and (b), of scaling: Assuming coherent-state approximation in Eq.(2), i.e., $\alpha = \text{tr}[\rho a]$, we have

$$\begin{aligned} \partial_t \alpha &= -i\Delta \alpha - iU|\alpha|^2\alpha - iG\alpha^{\dagger} \\ &- \frac{\left(\kappa + \kappa_{\phi}\right)^2}{\alpha} \\ &- \frac{\kappa_2}{\alpha^2} \end{aligned}$$

By defining $\tilde{\alpha} = \alpha/\sqrt{L}$, we see that the equation of $\tilde{\alpha}$ is invariant of L if LU and $L\kappa_2$ remain constant

$$\begin{aligned} \partial_t \tilde{\alpha} &= -i\Delta \tilde{\alpha} \\ &- i(LU)\tilde{\alpha}^2\tilde{\alpha} \\ &- iG\tilde{\alpha}^{\dagger} \\ &- \frac{\left(\kappa + \kappa_{\phi}\right)^2}{\tilde{\alpha}} \\ &- \frac{L\kappa_2}{\tilde{\alpha}^2} \end{aligned}$$

In other words, if we keep the parameters LU and $L\kappa_2$ as constants, U_0 and $\kappa_{2,0}$, respectively, we see DPT by comparing $\tilde{\alpha}$ with an increasing L .

Alternatively, we may keep Δ/L , G/L , and $\left(\kappa + \kappa_{\phi}\right)/L$ as constants, Δ_0 , G_0 and $\left(\kappa + \kappa_{\phi}\right)_0$, respectively. An equation invariant of L may be obtained by defining $t' = Lt$, i.e.,

$$\begin{aligned} \partial_{t'} \tilde{\alpha} &= -i\Delta_0 \tilde{\alpha} \\ &- iU\tilde{\alpha}^2\tilde{\alpha} \\ &- iG_0\tilde{\alpha}^{\dagger} \\ &- \frac{\left(\kappa + \kappa_{\phi}\right)_0}{\tilde{\alpha}} \\ &- \frac{\kappa_2}{\tilde{\alpha}^2} \end{aligned}$$

We also see DPT by comparing $\tilde{\alpha}$. However, one has to remember that the time derivative should be scaled by $1/L$ for comparison of time-domain quantities.

Because a mix of the two ways of scaling will also lead to a well-defined thermodynamic limit, I think that it is feasible for the current experiment to follow the right scaling.

I assume that the authors will successfully justify their scaling towards the thermodynamic limit, there are some other comments:

***** I see the second approach above is close to the authors' choice. It would require a different scaling of and the values of time axis in Figs. 4-6 as well as λ_{SSB} and $\lambda_{1\text{st}}$ for comparison.

***** The authors explained that the small range of L is limited by the measurement time, but would choosing a different biasing point solve this problem?

*** Fig. 2b.** I think zoom in of n_{ss}/L given in the rebuttal letter is very informative than Fig. 3a. Indeed, n_{ss} in Fig. 3a does not support the claims regarding the trend with L . I suggest to put the 2nd-order DPT zoom in beside Fig. 2c (the 1st-order DPT zoom in).

*** Fig. 3a-c.** I still cannot get the message of these plots. First, I think the y-axes should be n_{ss}/L and its derivatives instead of n_{ss} . Second, I think we would not only expect a higher peak of the 2nd derivative, but equally importantly also a narrower linewidth. These features are missing from the data.

2. Quantum signatures

I am not very skeptical on the claim "squeezing below vacuum" and its indication of "quantum", because the minimum fluctuation is smaller than the vacuum fluctuation. I know that the Hamiltonian is quantum mechanical, and it fits well with the experiment. However, I would become more critical when it leads to a conclusion such as "quantum fluctuations play a fundamental role". Questions arises in two categories: (i) How "quantum" it is? What specific "quantum fluctuation" is indicated here? and (ii) what does "classical fluctuation" predict? I think more evidences are needed to support such claims.

*** Fig. 3d.** The authors declined the Gaussianity check and the plot of anti-squeezing because they do not assume a Gaussian state to extract the squeezing level. But the question is whether it is a Gaussian state here? If Yes, then a Gaussianity check with zero thermal photon number would be a solid proof of the "quantum" claim. If it is not Gaussian, I am wondering what is role of the "squeezing" operator here? In both cases, the anti-squeezing may still provide useful information for understanding how "quantum" it is.

The authors also disagreed that (vi) comparing $\langle a^{\dagger 2} \rangle$ with $\langle a \rangle$ and $\langle a^{\dagger} a \rangle$, (vii) quantum state tomography, and (viii) $g^{(2)}(0)$ measurements are more difficult measurements. This comment was originally made on whether the current study is "comprehensive". In principle, I agree with the authors that the histogram they recorded should be adequate to generate these results. Indeed, a plot of these results may be very helpful here to support the "quantum" claim, especially $g^{(2)}(0)$.

One related comment on the state tomography: To the best of my knowledge, there have been two approaches for this purpose: (i) Assuming a small photon-number truncation and fit the density matrix via optimization, or (ii) assuming an analytical state and put in moments. Both methods rely solely on the measured histogram or moments. In Fig.\,S2, the authors have used $\langle a^\dagger a \rangle$, λ_{SSB} , and $\lambda_{1\text{st}}$ for tomography. It may be acceptable in the sense that all the variables are determined from the experiment, but I think it is very biased since a small or large value of $\lambda_{\text{SSB},1\text{st}}$ would already exclude most of the possibilities. I suggest use only moments for a less biased tomography.

{bf * Fig.\,S3.} I thank the authors for plotting all the relevant branches of metastability. I agree with the authors that considering a finite κ_2 leads to a better fitting between theory and experiment, so that it is a reasonable choice. What I do not understand is the interpretation of κ_2 : Why a negligibly small two-photon loss would indicate "quantum"? Shouldn't single-photon loss sounds even more "quantum"?

I have two more confusions on this plot: First, the claim that "only for $\kappa_2 \neq 0$ that the bright phase becomes unstable" is drawn from the semiclassical calculation. Second, I think the semiclassical calculation indicates that $\kappa_2 \neq 0$ results in hysteresis instead of instability. Third, I am wondering that if a different choice of parameter lead to very different comparison results? I agree that we should consider κ_2 , but I cannot follow the necessity of "quantum" here.

{bf *} In the rebuttal letter the authors states that "In the main text, we have tried to clarify this by specifying that the data are not quantum trajectories but rather collected along a single trajectories." But I fail to see this change in the revised manuscript.

I raised up this concern because photon fluctuation itself is broadband. In my opinion, using a narrow-band filter results in classical trajectories that jump from one could to another. The so called "quantum trajectories" are indeed very classical. Perhaps "single measurement trace" may be a better terminology?

It leads to the following questions on Figs.\,4a and 5d:

{bf * Figs.\,4a and 5d.} In my understanding, the two metastable states, ρ_{α} and $\rho_{-\alpha}$, should converge to a mixed state $\rho_{\text{ss}} = (\rho_{\alpha} + \rho_{-\alpha})/2$ in the symmetry-breaking region. If this understanding is correct, why the quadratures jump between two values with a notable time interval? I would expect that ρ_{ss} leads to $I=0$, considering that filter performs average. Even if the average is not sufficient, I would still expect a fast switching between μ I. Why the trajectory seems to stuck at either of the clouds for a while?

If I understand it correctly, the $t=0$ starts after a waiting time. What is the expected state at $t=0$? Is it a random choice between ρ_{α} and $\rho_{-\alpha}$? I am wondering whether $\langle a \rangle$ is a more straightforward measure of λ_{SSB} than the autocorrelation?

Or, is this measurement takes place after a sufficiently long waiting time where the system has reached ρ_{ss} ? If this is the case, why would autocorrelation function of steady-state measurement predict time dependence?

{bf 3. Z_2 symmetry}

The spontaneous breaking of Z_2 symmetry is very interesting, but I think it needs more data support.

Although λ_{SSB} provides sort of information, I think a direct observation related to the symmetry breaking is missing. In the symmetry-breaking phase, the two SSB states, ρ_{α} and $\rho_{-\alpha}$, would acquire a very long lifetime. The $\langle a \rangle$ measurement will show a very slow decay towards zero. Is this consistent with the measurement?

As commented above, I think $\langle a \rangle$ is a straightforward measure of λ_{SSB} ? What is the consideration that leads to the choice of the autocorrelation function?

{bf * Fig.\,4c.} In my understanding, the range for a small λ_{SSB} should cover the origin, since both positive and negative Δ would allow 2nd-order DPT. Two questions: (i) Where is the right boundary of the symmetry-breaking region? (ii) Why does the symmetry-breaking region looks solely positive?

{bf 4. Additional comments}

{bf * Fig.\,1a.} I know from the rebuttal letter that the red-dashed lines are covered by the blue lines. I would recommend making the dashed lines visible in the hysteresis region for clarity. Alternatively, replacing the label of "metastable" by "hysteric" may be more appropriate?

As shown in the schematic in the rebuttal letter, the x-axis can be understood as Δ . Because Δ can be either positive or negative, I think there should exist another critical point of the 2nd-order DPT on the positive side. Two SSB branches would exist between them, and may be drawn for clarity.

I understand that the authors want to keep the schematic general, but I do feel that the schematic in the rebuttal letter, with Δ as the x-axis and $\angle a \text{ or } \angle a^{\dagger} a$ as the y-axis, is much clearer to me.

As the authors explained in the rebuttal letter, λ_{1st} and λ_{SSB} are the minimum non-zero eigenvalues of different symmetry sectors corresponding to the order parameter μ , respectively. But a Liouvilian gap, as defined by the minimum non-zero eigenvalue of the whole matrix, should be the minimum of them, i.e., $\min\{\lambda_{1st}, \lambda_{SSB}\}$. Naming both of them as the Liouvilian gap is rather confusing to me. It is acceptable if this is the convention, otherwise I would suggest renaming them.

{bf * Fig.\,2d.} I understand from the rebuttal letter that the units of the x- and y-axes are not $\sqrt{\text{photon number}}$. I recommend add the "arb. units" label here and in Figs.\,4a and 5d.

{bf * Fig.\,5e.} In my understanding, the 1st-order DPT exists inside the symmetry-breaking region. Then, ρ_{α} and $\rho_{-\alpha}$ would also have a very long lifetime. How is ρ_{ss} distinguished from the metastable states ρ_{α} and $\rho_{-\alpha}$? I am thinking that looking only at $\angle a^{\dagger} a$ may end up in ρ_{α} and $\rho_{-\alpha}$ instead of ρ_{ss} . Will it mix up λ_{1st} and λ_{SSB} ? What does $\angle a$ look like?

In the last review round, I have asked whether the gap λ_{1st} looks wide. In the rebuttal letter, the authors provide a zoom out and indicate that it is indeed a narrow region. I am now wondering if it should be wide? I am considering that the symmetry-breaking region is wide with a small eigenvalue.

{bf * Fig.\,6.} I do not understand this claim: "we do not observe hysteresis in the region where $\lambda_{SSB} \ll \lambda_{1st}$." In my understanding, observation of hysteresis depends on the sweeping time, T , and the gap, λ_{1st} . The condition $\lambda_{SSB} \ll \lambda_{1st}$ should not exclude the existence of hysteresis for a small T . Do the authors mean a very small λ_{SSB} but a very large λ_{1st} ?

(Remarks on code availability)
Data and code are not available.

Reviewer #2

(Remarks to the Author)

In the revised version of the manuscript, the authors have addressed both my major and minor concerns while improving the presentation of their results. In particular, they provide a more systematic treatment of experimental errors and clarify the relation between their results and the expectations of Liouvilian theory.

Given the relevance of their work in the context of dissipative phase transitions and the pivotal role of quantum fluctuations, as stated in my previous report, I recommend now the manuscript for publication in Nature Communications.

(Remarks on code availability)

Reviewer #3

(Remarks to the Author)

I believe that this work has a notable gap between the presented data and the interpretation offered by the authors. The quality of the data is relatively weak, the signals of interest are elusive, and the authors tend to make very strong claims. As a consequence, the experiment does not support the claims made. I think the authors believe too much in their theory and see in their data what they want to see. I do not find the work convincing.

The work itself is valuable as it sets a precedent for more experiments in an original research direction. If it is toned down further, it deserves publication. In its current form, I just find it misleading, claiming an experimental real state that is out of bounds for the setup.

Below, I review the point on which I based my judgment.

The experiment is done in the classical regime

a) As far as I am concerned, the experiment is done in an entirely classical regime.

In particular, I completely disagree with Referee 2 (and with the authors who agree with the referee here) that "The observed quantum jumps (...) constitute a clear observation of the significant role of quantum fluctuations", and they have a "pivotal role".

Please see Figure 2e in PhysRevLett.123.254102. These jumps are entirely classical and entirely analogous to the jumps observed here by Beaulieu et al.

In fact, I propose to replace the words "jump" and "quantum jump" in the manuscript with "hop" or "classical jump" to avoid

the confusion that Referee 2 was led into.

I find this misconception, purposefully induced by the article, entirely unacceptable.

In general, the authors have not provided convincing experimental evidence that what they observed will not be observed too in the setup of PhysRevLett.123.254102. This is a question beyond the theoretical model.

I believe that the quantitative and qualitative data the authors present can also be acquired in a setup like that of PhysRevLett.123.254102. The theory analysis is indeed based on quantum trajectories and jumps. The experiment does not require it or demonstrate at all its need.

b) In their reply to my first review, the authors state that "As Heisenberg scaling cannot be obtained using classical resources, we argue that some quantum effect must be present around the critical point, despite the very dissipative nature of the system".

However, the paper does not contain any equivalent explanation or justification. The paper does not demonstrate any "metrological advantage" nor mentions that "the Heisenberg scaling" is a signature that (some) "quantum mechanics" must be present.

The word "some" used in the author's private reply to me seems to be much more adequate and honest in spirit. Still, I do not believe the toned-down argument either and I still wonder what do the authors really mean by "quantum" (I explicitly asked for a clarification on this in my first review and I do not find their answer).

Quantum fluctuations are related to the circuit properties (it's "mass"), and one needs quantum-limited amplification to directly observe these quantum fluctuations. The experiment does not have a quantum-limited amplifier and, therefore, needs to subtract extra noise in post-processing. This is not an elegant experimental technique. This is why quantum-limited amplifiers are important. They are hardly optional if the aim is to demonstrate the purely quantum nature of an observation.

Noise removal in post-processing can lead to all kinds of experimental artifacts, in particular, an overestimation of vacuum fluctuations. I judge that there is no evidence in the paper to decide if this calibration was made with sufficient care. If the fluctuation obtained after noise subtraction (a very delicate and error-prone process) contains classical noise, it is easy to reduce the fluctuations with squeezing without ever having witnessed any quantum mechanics.

This is related to the first referees' concerns about the calibration of the quadrature measurement. (See referee one comment 19)

At no time is it made clear (by the data) that " $\Delta x \Delta p = 1/4$ ", showing the measurements reach the Gaussian quantum noise limit. How was this uncertainty relation calibrated?

To put it in simpler language, the authors have not proven that the fluctuations they see are quantum in origin (even without any squeezing involved). In fact, I do not believe they are.

c) I find uncomfortable the Wigner plots they now show as "experimental reconstruction". The plots are clearly not a tomographic reconstruction. Tomographic reconstruction is a usual technique nowadays but the experiment in question does not have the possibility to measure it. In the field of quantum optics and quantum circuits "Wigner reconstruction" means full tomography. It is disappointing to read "Wigner reconstruction" and find the analysis made.

I think the Wigner plots shown are some sort of theory fit to some partial data, but they reflect the full phase-space distribution. Therefore, the "reconstruction" conveys much more information than the information the authors have experimental access to. It is, yet again, a way to overrepresent the experimental data.

The data does not show discontinuities

a) The authors insist on the "discontinuous jump" in Fig. 2(b) and the "a discontinuous jump at positive detuning can be observed" and the "continuous but non-differentiable change in photon number at negative detuning" in Fig. 3(c). I see no discontinuities or nondifferentiabilities in the data, because there are none. Sporadically the authors tone down the claim by introducing the word "emergence". This word should be systematically used and in the title of the paper.

It is true that the observation the authors are after is very (very) hard to measure experimentally and the word "emergence" is sufficiently ambiguous to be tolerated. The terrible experimental challenge faced is why this observation has not been reported directly in previous literature (to my knowledge) and I judge it has not yet been overcome yet. To do so, a highly refined experiment will be needed.

b) Regarding Figures 2b and c.

It is understood that here there are no discontinuities but only the "emergence of a discontinuity."

This is fine. As we already know from theory that in some limit ($L \rightarrow \infty$) there should be a discontinuity, we theoretically interpret this set of experimental points following over a smooth curve as the "emergence".

Here we have theory speaking for the data, but it is reasonable and acceptable as $L > 2$ is hard to handle experimentally as the authors clearly explain. Now, where do we see this is caused by unambiguous quantum effects? Where do we see this is caused by κ^2 ?

I believe that introducing a phenomenological κ^2 to the simpler model makes the effect emerge. But what about the alternatives? What about also including phenomenologically κ^4 ? Why not? or Hamiltonian terms other than Kerr which naturally become relevant at higher amplitudes? Or just broad-band noise? (All of this, including κ^2 , admits a classical

explanation unless several caveats are addressed experimentally and theoretically).

In other words, I would not use this type of experiment to calibrate a κ_2 in a setup (as the authors propose). It is too indirect evidence of the existence of a two-photon dissipation, and the effect could be caused by anything else. Furthermore, the authors do not spend any effort justifying κ_2 microscopically for their flux-driven SQUID.

Microscopically, κ_2 can be generated by noise in the flux drive at the two-photon drive frequency. What noise-temperature do the authors have in their drive at the sample level at the drive frequency? Why they do not consider $D[\text{adag}^2]$ too? There are thermal photons at 4GHz (n_{th}) but not at 8GHz? Is this temperature roughly compatible with the κ_2 inferred from the model applied to the data?

Without these sanity checks, the claim is not believable, and I see no reason to trust the analysis.

I am sure κ_2 plays a role in the theory model, but it is unclear if it exists or plays a role at all in the experiment (no reasonable microscopic model is presented, no sanity checks were made, no comparison to other hypotheses is presented). I don't think it necessarily does.

c) Regarding Figure 3c (Where are the error bars? Where is the theory curve for the derivatives?)

Indeed, the authors are after a quantity that is very elusive experimentally. The data needs to be presented with more touch or a better experiment needs to be designed. Let me make this clear by the following example.

Consider the function $f(x)$ which is $f(x) = 0$ if $x < 0$ and $f(x) = x^2$ if $x \geq 0$.

This function is continuous, its first derivative is continuous, and its second derivative is discontinuous. Consider now a signal consisting of this function plus a very small amount of noise and compute numerically its derivatives. I have done this in the plot below (the Python code to generate this figure is attached at the end of the report).

Just like the author's true experimental data, the blue numerical data I created has very good signal-to-noise. But what can one say, from the data (dots), about the second derivative?

As the authors explain to referee one (comment 18), they can smooth the noise in the calculation of the second derivative (taking a midpoint derivative) to see the "peak," but this average is a poor experimental technique (what are the error bars?). I do not think the information about the "continuity" is possibly there.

I think the "peak" in Figure 3c is essentially noise, an artifact of the freedom in choosing which midpoint derivative using many points. Exploiting that freedom, the plot could have been made to look very different. I believe it looks like a peak because that is what the authors are expecting to see from theory. I, as a reader, yet ignore completely what an unbiased analysis of the experimental data revealed to the authors in their exploration.

Other comments:

-Typo: "behavior behavior"

-How is "the dephasing rate mainly due to the nonlinearity of the resonator"? Isn't it flux noise?

-Why is the "five order of magnitude slowdown" a demonstration of quantum criticality?

What would one expect for a classical parametric oscillator? Isn't the strong slowdown a well-known fact? This slowdown has been observed in classical and quantum systems already (citations of this scaling shine by their absence. Quantum dissipative cats dominated by κ_2 are a bad example for this work, where κ_2 is very small and the two bright state is due to Kerr)

-"Our work impacts all major quantum technological platforms." I think the claims about quantum technology and quantum information processing are all unwarranted and beyond the scope of this setup. The experiment is not close to a regime that would be useful for quantum information. Turning this experiment into a quantum information experiment will be a mayor task and it is unfair to extrapolate this observations to that setting. That defeats the point of experimental science.

-The words "establish" or "demonstrated" are too strong, and the data cannot back them up. Maybe "seen signatures of" would be better.

-"Quantum fluctuations and quantum dissipative processes could be the main drives of the observed transitions". This is a better statement but it contradicts the statements made before. "Could be" or "demonstrably is"?

"They show how quantum processes could trigger DPTs, highlighting the necessity to study DPTs within a quantum framework." I disagree with the necessity. It contradicts the previous statement. "Could" or "necessitates"?

Code for figure:

```
span = 10
samples = 101
xs = np.linspace(-span/2,span/2,samples)
```

```
def func(xs):
y = np.zeros_like(xs)
for i, x in enumerate(xs):
```

```
y[i] = (x+np.abs(x))**2/4
return y
```

```
function = func(xs)
signal = function + np.random.randn(samples)/250
```

```
dx = span/samples
plt.plot(xs,function,'-b', label = r'$f=(x+|x|)^2/4$ (noiseless)')
plt.plot(xs,signal,'ob', label = 'singal = f+small noise')
```

```
plt.plot(xs[:-1],np.diff(function)/dx, '-r',label = 'first numeric derivative (noisless)')
plt.plot(xs[:-1],np.diff(signal)/dx, 'or',label = 'first numeric derivative (signal)')
```

```
plt.plot(xs[:-2],np.diff(np.diff(function))/dx**2, 'k-',label = 'second numeric derivative (noiseless)')
plt.plot(xs[:-2],np.diff(np.diff(signal))/dx**2, 'ok',label = 'second numeric derivative (signal)')
```

```
plt.legend(loc='center left', bbox_to_anchor=(1, 0.5), ncol = 1)
plt.xlabel('x')
plt.ylim(-1,4)
```

(Remarks on code availability)

Version 2:

Reviewer comments:

Reviewer #3

(Remarks to the Author)

The new version of the manuscript has been considerably tuned down, and the analysis seems reasonable to me now. I appreciate the effort the authors put into clarifying my doubts.

Yet, regrettably, I do not see in this work any technical breakthrough or new theoretical insight that would justify publication in Nature Communications. In particular, I think this work is far from advancing state-of-the-art quantum information applications.

(Remarks on code availability)

Reviewer #4

(Remarks to the Author)

This manuscript presents a detailed experimental analysis of the parametric instability and bistability of a Kerr resonator. Such instabilities can also be described as a mean-field dynamical phase transitions and occur in both classical and quantum driven-dissipative systems. The phase transitions are “rounded up” by non-linear corrections that become negligible in the correct scaling limit. As mentioned by the authors and by the reviewers, the topic has been studied extensively in earlier works both theoretically and experimentally and the present study contributes incrementally to the field. Nevertheless, considering that the manuscript has already undergone two rounds of reviews and that the authors made noteworthy efforts in replying to all comments by the previous referees, I suggest accepting the manuscript in its present form.

(Remarks on code availability)

Reply of manuscript NCOMMS-24-15284-T

*** Reply to Reviewer 1 ***

We first want to thank the Reviewer 1 for acknowledging the novelty of this work and for their thorough review, which has helped improve the quality of the manuscript and clarify some important points. Below are our responses to the concerns and questions raised by the Reviewer.

Comment 1: In addition to my appreciation for the novelty and interest of this work, I see a notable gap between the presented data and the authors' interpretations. On the one hand, the data quality is relatively weak, such that the actual signals of interest are often elusive. I also have several confusions regarding the authors' approach to the thermodynamic limit, and the interpretation of their data. On the other hand, the authors tend to make very strong claims with weak data support. I cannot fully agree with many of the claims, such as whether a phenomenon has been "unexplored" or a phenomenon can be seen from the data. I also cannot agree that this is a "comprehensive" study of DPT. These aspects compromise the solidity and the potential impact of this study.

The Reviewer raises significant concerns about the quality of the data and the interpretations presented. Throughout the Reviewer's comments, it seems that the main concerns about the data quality relate to the removal of noise. Additionally, doubts regarding the thermodynamic limit and our interpretation are raised. We will clarify both aspects in the comments below. For now, we will leave these issues aside to address the comprehensiveness of our study.

Our work shows signatures of both first- and second-order dissipative phase transitions in the two-photon driven Kerr resonator, which are in agreement with the underlying theory. Although other signatures of the transition can be derived, we argue that the ones we present provide a convincing demonstration of the critical nature of the system. We hope that the reply below, along with the updated version of the manuscript, will also convince the Reviewer.

On the novelty and comprehensiveness

Comment 2: While the investigation into the 2nd-order DPT is indeed novel, I find the term "unexplored properties" unclear. The authors have studied (i) transition speed, (ii) squeezing, (iii) critical slowing down, (iv) Liouvillian gap, and (v) power-law decay, but I believe all of them have been explored in the literature [PRL 118, 247402 (2017), Sci. Adv. 7, eabe9492 (2021), and Nat. Commun. 14, 2896 (2023)].

Although we agree with the Reviewer that these properties have been studied in other models, we note that all of the given examples relate to first-order transitions and single-photon drive, as the Reviewer acknowledges. The features of first-order transitions are extremely relevant and interesting, which is why all of the articles above are cited in our manuscript.

That being said, the "unexplored" aspect and novelty of our work lies in studying similar and different properties due to the two-photon drive, which gives rise to second-order transitions and spontaneous symmetry breaking. The models in the cited articles, due to their lack of symmetry (from having a single photon pump), cannot, for instance, show any signature of the competition between two critical timescales, such as the metastability of the bright states with the vacuum and the passage between the two states of opposite phases. In fact, one of the main interests of our article is showing that many features of the first-order transition differ significantly from those of the second-order transition. We provide more details and examples below while addressing the Reviewer's questions (in particular see comments 7 and 8). Nevertheless, to avoid any ambiguity, we have removed the word "unexplored" from the manuscript.

Comment 3: Moreover, several more challenging experiments, such as (vi) coherent cancellation dip, (vii) photon bunching, and (viii) quantum state tomography, have been conducted in the literature but are not addressed in this paper [PRL118, 040402(2017), Nat. Phys. 14, 365- 369 (2018), and Nat. Commun. 14, 2896 (2023)]. I think they might be even more interesting features of DPT from a theoretical point of view [Phys. Rev. A 94, 033841 (2016), Phys. Rev. A 93, 033824 (2016), and Phys. Rev. A 95, 012128 (2017)].

Some clarifications are important here.

Cancellation dip (vi): due to the Z_2 symmetry of the model considered in our work, $\langle \hat{a} \rangle = \langle -\hat{a} \rangle = 0$. As such, $|\langle \hat{a} \rangle| \neq \langle \hat{a}^\dagger \hat{a} \rangle$ everywhere. For this reason, we did not plot the coherent cancellation dip, as it would simply be the plot of $\langle \hat{a}^\dagger \hat{a} \rangle$, which is already shown.

Photon bunching (vii): as indicated in the articles cited by the Reviewer, photon bunching is a signature of first-order dissipative phase transitions. Correlation functions are also expected to be a clear signature of both transitions for the model under consideration here, but this may not be the case in general. For instance, consider [Phys. Rev. A 97, 013853 (2018)], where a lattice of resonators is studied. In this case, the $g^{(2)}(0)$ shows far less impressive features than those of the thermodynamic limit of a single cavity. On the other hand, critical slowing down and observable behavior are common features across all first-order transitions, including cases where $g^{(2)}$ is not as relevant. For this reason, we decided to focus on the Liouvillian spectral structure, which we present in the main article. That is to say, notwithstanding the signature provided by $g^{(2)}$, we show other features that capture the transition as well.

We have added the clarification in the main text that "Other signatures of first-order DPTs can accompany the transition, including photon bunching and interference effects," with additional citations.

Quantum state tomography (viii): let us note that the Wigner function in [Nat. Commun. 14, 2896 (2023)] was obtained by fitting the two parameters c and d appearing in Eq.(5) of the Supplementary Material of that article, and then plotting the analytical solution for the function (as explained in Supplementary Note 4, paragraph C of the cited reference).

Similarly, we have fitted comparable parameters (using a different procedure based not only on the moments of the field but also on the Liouvillian eigenvalues). From these fitted parameters, we can also plot the corresponding phase space representations. Below, we show the obtained Wigner function across the phase diagram using a similar approach to [Nat. Commun. 14, 2896 (2023)]. In the original version, we decided to present only the experimental data, but we have now added this figure to the extended data figure.

For all the reasons mentioned above, we believe that our measurements are not less advanced than those implemented in the references. Incidentally, one of the authors of the current article is also an author of the referenced theoretical work. The difficulties of the measurements are discussed in comment 4.

Comment 4: I therefore cannot agree that this study is “comprehensive”. It is not only because the more difficult measurements, (vi)-(viii), are not touched in this work, but also because the observations of (i)-(iv) seem elusive and a seeming negative result of (v) is observed. Although I agree with the authors that the signatures of DPTs are there, I am not fully convinced that the approach to the thermodynamic limit is correct.

We disagree on the difficulty of measurements (vi)-(viii). As explained in comment 3, (vi) is already shown, (vii) has the same difficulty as the squeezing (it is a moment reconstruction measurement), and (viii) has been obtained using a similar procedure to that of the cited article without any additional data.

Results of (v) are addressed in comment 26 and the thermodynamic limit in comment 5 below. Given these points, and also given our replies below, we do not see the “elusiveness” of our results. That being said, we have removed the term “comprehensive” from the manuscript.

On the approach to the thermodynamic limit

Comment 5: The first quote actually aligns with the comment I am about to make. When approaching the thermodynamic limit, the dissipation process must be carefully controlled. However, the dissipation rate, κ , is not controlled at all throughout this study.

The authors argued in Methods that “As the data demonstrate, κ plays only a marginal role in determining the properties of the second-order DPT and of the bright phase.” I may understand this

argument as implying that the scaling factor, L , of this experiment, varies in a very small range, so it will not lead to a very different value of κ . However, the issue emerges when the authors compare the data with different L .

Since a small range of L implies subtle differences in the data, it demands even more precise control of κ to ensure accurate conclusions. Neglecting the scaling factor of κ simply makes the “desired” result more evident, but it is artificial. On the other hand, the device used in this study allows for the control of κ . My question is why did the authors choose not to do it? Is it because κ_2 also varies at different biasing points? How large is the relative change of κ_2 compared to that of κ ? I think the authors must argue very carefully why it is feasible to not control κ .

Several remarks are in order here.

The thermodynamic limit in the model under consideration can be reached in two ways [Phys. Rev. A 94, 033841 (2016)]:

1. Simultaneously changing $U \rightarrow U/L$, $\kappa_2 \rightarrow \kappa_2/L$ and $\kappa_\phi \rightarrow \kappa_\phi/L$. This is not possible, as we can only systematically control U by changing the biasing point. Changing the biasing point results in effective uncontrolled changes of κ , κ_ϕ and κ_2 . Therefore, the actual scaling in the experiment would be $U \rightarrow U/L$, $\kappa \rightarrow f(L)\kappa$, $\kappa_2 \rightarrow \kappa_2g(L)$, and $\kappa_\phi \rightarrow \kappa_\phi h(L)$, where $f(L)$, $g(L)$, and $h(L)$ are function that would need to be experimentally characterized.
2. Simultaneously changing $\Delta \rightarrow \Delta L$, $G \rightarrow GL$ and $\kappa \rightarrow \kappa L$. This is also not possible. Even though κ could be changed by adjusting the biasing point, this approach lacks control [New J. Phys. 15 105002 (2013)] and, more importantly, would also result in a change in U , κ_2 , and κ_ϕ . These changes would effectively invalidate the scaling. Therefore, the actual scaling in the experiment would be $\Delta \rightarrow \Delta L$, $G \rightarrow GL$, and $\kappa \rightarrow \kappa$.

Of these two scalings, only the second one corresponds to a well-defined thermodynamic limit. Indeed, if we go to the thermodynamic limit using the second scaling even while setting $\kappa \rightarrow 0^+$, the phase transitions—both the first- and second-order ones—would still be present. In this limit, DPTs would be triggered by two-photon dissipation, detuning, Kerr nonlinearity, and dephasing. As this limit is much more experimentally controllable than the other one, it is the one we used for scaling.

Obviously, we are still concerned by finite-size effects. That is the reason for the Binder cumulant study. Binder cumulants are employed to mitigate finite-size effects in phase transitions and distinguish between phase transitions and crossovers. Please note that in the figure, we check the curves for all the L values considered in the paper (there was a minor typo in the caption which is addressed in comment 17). As the Binder cumulant criterion was satisfied, we were assured that the considered scaling and the considered L were sufficient to make this claim for the second-order transition. As it is evident from the data in Fig.2, however, finite-size effects are still present for the first-order transition. We now commented more on the Binder cumulants in the main text.

Incidentally, also note that the chosen scaling is more appropriate in light of quantum technological applications such as the generation of Schrödinger cat states (where single-photon dissipation is a nuisance rather than a desired feature) [PRX Quantum 4, 020337 (2023)]. These underlying theoretical reasons also guided our decision to opt for the chosen scaling.

Comment 6: *A minor complaint is that L varies only from 1 to 1.58 in this study. Ideally, we would hope it to change from 1 to ∞ . I see no obvious limit on L in this experiment. For example, the input power is only varied by 3 dB, which is far from the full range of a normal RF source. May the authors explain what limits L in this study?*

The Reviewer is correct in suggesting that the power of the RF source doesn’t impose a limit on the value of L . The limiting factor in this experiment was instead the timescale of the phenomena under study. The critical slowing down for $L = 1.58$ already leads to timescales on the order of minutes for single jumps. Collecting enough statistics for each detuning requires hundreds of jumps and therefore hours of measurements. According to the predicted extrapolations, considering $L = 2$ would lead to timescales of $\lambda \sim 10^{-5} \text{ s}^{-1}$ which means several hours for a single jump. Putting aside

the impracticality of conducting such long measurements, instrument drift or resonator frequency drift, as commented on in the supplementary material, would occur, making the results rather noisy.

We also want to stress that going to higher values of L is not strictly required. The Binder cumulant study (see comment 5) and the collapsing of the curves on top of each other are sufficient to demonstrate the onset of a second-order phase transition.

Results A: Steady state properties and phase diagram

Comment 7: Fig. 1a. I would expect two metastable states around the 1st-order DPT, but it appears not to be the case in the schematic plot. The two metastable branches seem not to be the two metastable states around the 1st-order DPT. Why there is one metastable state and one steady state?

For all finite-size systems, such as the one considered in the article, the steady state is unique. There are, however, three possible “metastable” states that are involved in the precursors of the transition: the vacuum-like state (for the sake of brevity, here ρ_{vac}), and the two coherent-like states (here $\rho_{+\alpha}$ and $\rho_{-\alpha}$) [Phys. Rev. A 98, 042118 (2018)].

Before the first-order transition: the steady state is $\rho_{\text{ss}} \propto \rho_{+\alpha} + \rho_{-\alpha}$ and ρ_{vac} and $\rho_{\pm\alpha}$ are metastables. The metastable ρ_{vac} decays to ρ_{ss} at a rate given by $\lambda_{1\text{st}}$, while Each of the $\rho_{\pm\alpha}$ decays to ρ_{ss} in a time λ_{SSB} .

After the first-order transition: the steady state is $\rho_{\text{ss}} = \rho_{\text{vac}}$ and $(\rho_{+\alpha} + \rho_{-\alpha})/2$, and $\rho_{\pm\alpha}$ are metastables. The metastable state $(\rho_{+\alpha} + \rho_{-\alpha})/2$ decays at a rate $\lambda_{1\text{st}}$, while each of the states $\rho_{\pm\alpha}$ decays to $\rho_{+\alpha} + \rho_{-\alpha}$ at a rate λ_{SSB} .

At the first-order transition: the steady state is three modal $\rho_{\text{ss}} \simeq A\rho_{+\alpha} + B(\rho_{-\alpha} + \rho_{\text{vac}})/2$ with A and B real coefficients such that $A + B = 1$ in the thermodynamic limit.

The whole schematic is shown below:

In the schematic of Fig.1(a), we specify, “the purple dashed lines indicate the metastable states associated with hysteresis across the first-order DPT.” In the schematic above, these would be the red and green states. Plotting the SSB state in panel (a) as well would make it extremely crowded. Nonetheless, the presence of metastability associated with SSB is indicated in the sketch of the quasi-probability function in Figure 1. This is, however, a style choice, and we have no issue expanding the figure if the Reviewer believes it would be more beneficial to the reader.

It might also be clearer now that the model is far richer than that of a first-order phase transition in the absence of spontaneous symmetry breaking (single-photon drive) and what we mean by the “unexplored” property and competition between critical timescales discussed in comment 2.

Comment 8: *Fig. 1c. I am confused of the definition of the Liouvillian gap here, particularly the meaning of λ_{SSB} . If we take the usual definition of the Liouvillian gap (the minimum non-zero eigenvalue of the Liouvillian superoperator), why are there two different gaps at the same parameter? I am trying to interpret it as that λ_{SSB} and $\lambda_{1\text{st}}$ are the 1st and 2nd smallest non-zero eigenvalues. $\lambda_{1\text{st}}$ becomes the Liouvillian gap in the region where λ_{SSB} is closed. Is this understanding correct? In addition, why the critical point of the 2nd-order DPT is indicated at a position where the Liouvillian gap is not closed?*

This is briefly explained in the section “Symmetry and Liouvillian eigenvalues” of the Methods and further detailed in [Phys. Rev. A 98, 042118 (2018)]. Both $\lambda_{1\text{st}}$ and λ_{SSB} are Liouvillian gaps (the minimum non-zero eigenvalues of the Liouvillian superoperator), but they belong to two different symmetry sectors of the Liouvillian. While $\lambda_{1\text{st}}$ sets the timescales associated with the first-order transition, λ_{SSB} describes both the second-order transition and the corresponding spontaneous symmetry breaking.

As for the fact that the point where the second-order DPT occurs is not the minimum of the Liouvillian gap λ_{SSB} , this is the main difference compared to the point-like closure of the Liouvillian gap associated with a first-order DPT. Let us further note, incidentally, that the minimum of λ_{SSB} may even occur in a region where the system is in the vacuum, indicating a breaking of the Z_2 symmetry in a metastable region. This is the principle of the critical Schrödinger cat, recently proposed by some of the authors [PRX Quantum 4, 020337 (2023)].

Finally, we stress again that the features associated with both transitions combine and compete. Using a single Liouvillian eigenvalue cannot capture the characteristics of a first-order transition in the presence of symmetry. This is also one of the main novel points of the article, and the reason why we emphasize symmetry and the importance of correctly distinguishing between the two processes and the two timescales. This further resonates with our previous comments 1 and 2 on the novelty of our work relative to the previous literature experimentally studying first-order dissipative phase transitions with single-photon drive.

Comment 9: *Fig. 1d Is the left side of the feedline left open, shorted, or terminated by a 50Ω load in the experiment?*

We thank the reviewer for helping us clarify the schematic, which we tried to simplify but ended up being ambiguous. The left-side of the feedline is connected to 50 Ω outside the fridge. The device is a hanger-type $\lambda/4$ resonator, and the left side of the feedline is only used for spectroscopy measurements and is necessary for future experiments. We have now adjusted the figure and part of the caption to “the device is a hanger-type $\lambda/4$ coplanar waveguide resonator. The right side of the feedline is used to collect the emitted signal via heterodyne detection, whereas the left side is only used for spectroscopy measurements to extract the device parameters and is otherwise terminated by 50 Ω”.

Comment 10: *Paragraph 2, lines 7-8: “Knowing the output gain G and total loss rate κ , the field quadratures I and Q of the cavity are then reconstructed (see Methods).” Methods (Acquisition of the signal), paragraph 1, lines 9-12: “From the measured I_m and Q_m , the quadratures of the intracavity field (I, Q) are obtained by removing the effect of the amplification chain and its associated noise.”*

Paragraph 4, lines 19-20: “The histograms of the measured I and Q quadratures, i.e., the Husimi functions of the steady state convoluted by the noise of the amplifier, ... are plotted (in) Fig. 2d.”

By reading the Methods, the amplification noise is removed such that I and Q are the genuine intracavity field quadratures up to a scaling factor. However, by reading the last quote, it seems that I and Q are convoluted quadratures of the signal and the noise. Are the data in all the figures also not analyzed in this way? Which data in this study are measured with the described method? I think this difference will lead to very different interpretations of the data. For example, the claim “The

intracavity photon number is $n_{ss} = \langle I^2 \rangle + \langle Q^2 \rangle$." will not be correct. I point out this issue here and will comment on the specific data later.

We apologize for any confusion here.

The measured quadratures acquired by the analog-to-digital converter (Quantum Machine OPX+) are referred to as I_m and Q_m . Up to a scaling factor, these quadratures correspond to output signal field at cryogenic temperature $\hat{b}_{out}^{(r)}(t)$ (escaping the feedline on the right side) convoluted with the field operator of the amplification noise \hat{h}^\dagger . More precisely, we define the complex envelope operator \hat{S} as :

$$\hat{S} \equiv \frac{1}{\sqrt{Z_0 h f}} \frac{\hat{I}_m + i \hat{Q}_m}{\sqrt{\mathcal{G}}} = \hat{b}_{out}^{(r)} + \hat{h}^\dagger, \quad (1)$$

where \mathcal{G} measurement-line gain, $h = 6.63 \times 10^{-34}$ Js is the Planck constant, f the frequency and Z_0 is the impedance. The measured histograms in Fig.2(d), Fig.4(a), and Fig.5(d) are obtained from the quadratures of the complex envelope referred to the cavity

$$\hat{S} \sqrt{\frac{2}{\kappa_{ext}}} = \hat{I} + i \hat{Q} \quad (2)$$

As such, these quadratures are a convolution of the cavity field and amplifier noise.

The expectation value of the intracavity field moments are obtained using the same approach as [Phys. Rev. A 86, 032106 (2012), New J. Phys. 16 015001 (2014) and Nat. Commun. 14, 2896 (2023)], the expectation value of the moments $\langle (\hat{b}_{out}^{(r)\dagger})^i (\hat{b}_{out}^{(r)})^j \rangle$ can be computed using

$$\langle (\hat{S}^\dagger)^n \hat{S}^m \rangle_{\rho_{\hat{b}_{out}^{(r)}}} = \sum_{i=0}^n \sum_{j=0}^m \binom{n}{i} \binom{m}{j} \langle (\hat{b}_{out}^{(r)\dagger})^i (\hat{b}_{out}^{(r)})^j \rangle \langle \hat{h}^{n-i} (\hat{h}^\dagger)^{m-j} \rangle. \quad (3)$$

The expectation value of intracavity field moments $\langle \hat{a}^{\dagger k} \hat{a}^l \rangle$ are then derived (see supplementary section F) as

$$\langle \hat{b}_{out}^{(r)\dagger k} \hat{b}_{out}^{(r)l} \rangle = \left(\frac{\kappa_{ext}}{2} \right)^{\frac{k+l}{2}} \langle \hat{a}^{\dagger k} \hat{a}^l \rangle. \quad (4)$$

Consequently, the steady-state photon number $n_{ss} = \langle \hat{a}^\dagger \hat{a} \rangle$ corresponds to the true photon number in the cavity with the noise photon number of the amplification chain correctly removed. This corresponds to the photon number shown in Fig 2(a,b,and c) and Fig 3(a). **All the expectation values** plotted in the paper, including the moments of the intracavity field, are obtained by removing the effects of the amplification chain and its associated noise.

As accurately pointed out by the Reviewer, there was some imprecision regarding the use of I and Q in the main text and methods. However, it is important to emphasize that the data treatment is correct and consistent across all figures. This point has been clarified by in the main text, methods and in the supplementary.

Comment 11: Fig. 2a. Because $n_{ss} \neq \langle I^2 \rangle + \langle Q^2 \rangle$ if the noise is not removed, I am not convinced by the z-axis values. I am assuming that the n_{ss} -axis is obtained by rescaling $\langle I^2 \rangle + \langle Q^2 \rangle$. It is not correct because of the convolution relation between signal and noise. I think the authors need to characterize and then subtract the noise photon number from $\langle I^2 \rangle + \langle Q^2 \rangle$ to get n_{ss} .

The Reviewer is correct that the noise photon number needs to be subtracted from the rescaled $\langle I^2 \rangle + \langle Q^2 \rangle$ to obtain the correct n_{ss} value. As detailed in the previous comment 10, we correctly remove the noise photon number and provide the detailed calculation in supplementary sections E and F.

Comment 12: According to the Methods, the authors choose to rescale Δ , G , and κ (not controlled in reality) instead of U and κ_2 . I am thinking that the energy of the energy will be L times larger than the usual way of rescaling [Phys. Rev. A 94, 033841 (2016)]. Thus, for comparing data with different L ,

shouldn't the y-axis being n_{ss}/L^2 instead of n_{ss}/L ? Here, the physical meaning of n_{ss}/L^2 is the density of photons in an L -size system.

As discussed in comment 5, one way to reach the thermodynamic limit is by rescaling the parameters $\Delta \rightarrow \Delta L$, $G \rightarrow GL$ and $\kappa \rightarrow \kappa L$ with $L \rightarrow \infty$. As described in the work cited by the Reviewer [Phys. Rev. A 94, 033841 (2016)], "in the thermodynamic limit, one expect $\langle a^\dagger a \rangle \propto |g|$ [with $g = G/(U - i\kappa_2)$]" . By substituting these rescaled parameters, the photon number n_{ss} consequently exhibits a universal behavior in the rescaled picture $\langle n_{ss} \rangle / L$ for $L \rightarrow \infty$. This can further be highlighted by taking the mean-field approximation with $\kappa_2 = 0$ (Methods Eq. 7):

$$n_{ss} = \frac{\Delta + \sqrt{|G|^2 - (\kappa + \kappa_\phi)^2}}{|U|} \quad (5)$$

where n_{ss} must be divided by L to be invariant under the transformation $\Delta \rightarrow \Delta L$, $G \rightarrow GL$ and $\kappa \rightarrow \kappa L$.

Comment 13: *The authors wrote that "As L increases, the emergence of a continuous but non-differentiable change in the photon number at negative detuning, and a discontinuous jump at positive detuning can be observed [see also Fig. 2(c)]." However, I cannot agree that I see a "non-differentiable change" at $\tilde{\Delta}/2\pi = -0.04$ MHz nor a "discontinuous jump" at $\tilde{\Delta}/2\pi = 0.13$ MHz as indicated by the authors.*

If we accept the scaling as it is, I would agree that the data trend in Fig. 2c indicates a discontinuous jump at $\tilde{\Delta}/2\pi = 0.13$ MHz at $L = \infty$, corresponding to a 1st-order DPT. However, I cannot see the described data trend at the indicated critical point of the 2nd-order DPT. I think the authors should provide a zoom in of the data around $\tilde{\Delta}/2\pi = -0.04$ MHz to justify the observations of the 2nd-order DPT.

Although we completely agree with the reviewer that a true phase transition occurs in the thermodynamic limit $L = \infty$, reaching this limit experimentally is not feasible (see comment 6). What we observe experimentally is only the "emergence of [...] a discontinuous jump at positive detuning", i.e., the feature in the process of becoming visible. As L increases, it is clear from Fig. 2(c), that a discontinuous jump at positive detuning is emerging.

Concerning the second-order DPT, we fully agree with the Reviewer that "the emergence of a continuous but non-differentiable change in the photon number at negative detuning" in Fig. 2 could be questionable. For this reason, we provide a zoom of the n_{ss} as a function of the detuning in the region of $\tilde{\Delta}/2\pi = -0.04$ MHz in Fig. 3(a). We apologize if this was not clear and have now clearly indicated at this point in the main text that a zoom on that region is provided in Fig.3(a).

Also note that to further demonstrate that we observe the emergence of a continuous but non-differentiable change in the photon number at the second-order DPT, we also present the first and second derivatives of n_{ss} in Fig.3(b) and (c). From Fig.3(c), it is clear that as L increases, the non-differentiability of the photon number becomes more pronounced (it increases with L).

Finally, the Binder cumulant analysis of the emergence of a second-order discontinuity is in line with these previous results (see comment 5), further corroborating the presence of a second-order discontinuity.

Comment 14: *Besides a feature of "continuous but non-differentiable change" the slope should also be almost independent of L around the 2nd-order DPT. Ideally, a transition from a horizontal line to the same slope may be observed. I think an observation of this trend would be important to verify the existence of 2nd-order DPT.*

We completely agree with the Reviewer that the slope of \tilde{n}_{ss} as a function of $\tilde{\Delta}$ should become independent of L (for sufficiently large L) after the 2nd-order DPT. This is one of the reasons \tilde{n}_{ss} is shown as a function of $\tilde{\Delta}$ in Fig.2(b), where it can be seen that, away from the first-order DPT, all the lines collapse on top of each other when L becomes sufficiently large, showing that they have the same slopes. We also provide below a zoom similar to Fig.3(a), but in the rescaled picture of \tilde{n}_{ss} and

$\tilde{\Delta}$, showing that for sufficiently large values of L and slightly past the second-order phase transition, all curves collapse on top of each other. This also shows that scaling of κ plays a marginal role in that region (comment 5).

We also want to stress that for finite-size L , finite-size effects will always be visible when zooming in closely enough around the critical point as can be seen from the plot above. As previously explained in comment 5, a tool to avoid finite size effects and associated artifacts is the Binder cumulant analysis that, as shown in panel (f) of Fig 2, confirms the trend of an emergent critical phenomenon. All these observations point to the existence of the 2nd-order DPT. This analysis is further corroborated by the study of the critical slowing down of λ_{SSB} .

Furthermore, we also calculate the slope between the second- and first-order DPT from the data in Fig. 2(a). In the figure below, we plot the value of the slopes of n_{ss} as a function of the detuning Δ for different pump amplitudes, clearly showing that it saturates when L becomes sufficiently large.

Comment 15: Fig. 2d. Since the I, Q quadratures are convoluted signals with the noise (usually at the 10-photon scale), I am somewhat surprised that the coherent “clouds” are well-separated at a low photon number. I guess that it is because of a very small measurement bandwidth and rescaling.

I may roughly estimate the measurement bandwidth as $B = 40 \text{ Hz} - 1 \text{ kHz}$, since the Methods say that one data point corresponds to an integration time of $2\text{-}50 \mu\text{s}$. In comparison, the linewidth of the nonlinear resonator is $\kappa/2\pi = 77 \text{ kHz}$. Here, $B \ll \kappa/2\pi$ so that I may naively interpret the small histogram clouds as heavily averaged results.

The very narrow measurement bandwidth is not an issue here since the expected features are in the mean value. The expected features are clear in the plot. However, I am wondering if it will cause issues in later discussions since the histogram does not correspond to the Q quasidistribution function anymore. The uncertainty relation of I and Q is averaged out. For example, the authors wrote that "... $I(\tau)$ is the measured quadrature at time τ along a single quantum trajectory such as those shown in Fig. 4(a)." I cannot agree that a heavily averaged measurement trace can be interpreted as a single quantum trajectory. On the other side, I understand that the usual requirement of $B > \kappa/2\pi$ is just a "safe" experimental condition. The authors interpretation may still be correct. Since the authors decide to not follow the "safe" configuration, I think that they need to justify their method.

Lastly, what are the units of the x - and y -axis for the histogram?

The Reviewer is correct that the coherent "clouds" are well-separated due to the long integration time of the measurement and that the histograms represent heavily averaged results. This is motivated by the fact that the timescale of interest is dictated not by κ , but rather by the Liouvillian gaps $\lambda_{1\text{st}}$ and λ_{SSB} , which range from approximately 100 kHz (1 jump event every $10 \mu\text{s}$) to 0.01 Hz (1 jump event every 100 seconds). The goal of the measurement is to capture these events. Consequently, collecting samples every $2\text{-}50 \mu\text{s}$, depending on the jump rate, is more than sufficient to distinguish the events clearly. This is also evident in Fig. 4(a) and Fig. 5(d), where measurements after rescaling are directly shown.

We also stress the extremely long time scales of the events. Collecting a single trace (for one value of detuning) to extract λ_{SSB} at $L = 1.58$ with 100 jumps conservatively takes approximately 15 minutes. Sampling this trace at intervals on the order of 100 ns such that $B \ll \kappa/2\pi$, would require a staggering amount of data points (on the order of 10^{10}).

The Reviewer states that a heavily averaged measurement trace cannot be interpreted as a single quantum trajectory. While we agree that some of the fast dynamics, such as the ring-up or ring-down of the resonator, are lost by collecting subsequent points averaged over a long time, the captured trace still qualitatively corresponds to the behavior along a quantum trajectory. Each "snapshot" taken from the trajectory (i.e., each point collected) is simply an averaged one, which does not affect the interpretation of the dynamics if the timescales we are interested in are much longer than the averaging time. In the main text, we have tried to clarify this by specifying that the data are not quantum trajectories but rather collected along a single trajectories.

Finally, the x - and y -axis for the histograms are unitless since they represent the rescaled field quadratures convoluted with the amplifier noise (see comment 10).

• **Comment 16:** Paragraph 4, lines 19-20: "As the detuning increases across the second-order DPT, the vacuum becomes squeezed [see also Fig. 3(d)]."

Again, I cannot discern this phenomenon from the plots (I will comment on Fig. 3 later). I tend to believe that there are two closely spaced coherent clouds in the 2nd panel of Fig. 2d instead of one squeezed cloud. The claimed feature of squeezing seems elusive. Perhaps the problem can be settled by checking the Gaussianity of the histogram.

Here, the intracavity photon is almost 0 but the noise photon number is approximately 10. The latter will blur any small signal into a Gaussian distribution. Thus, strictly speaking, the authors need to do the noise removal before the Gaussianity check. However, working directly on the histogram data may also be acceptable but the claims such as "squeezing below vacuum" will not hold.

We agree with the Reviewer that a squeezed vacuum state cannot be discerned directly from Fig.2(d). This figure is not intended to make such a state discernible. As also correctly pointed out by the reviewer, it consists of two closely spaced coherent clouds. The statement “as the detuning increases across the second-order DPT, the vacuum becomes squeezed [see also Fig.3(d)]” was intended to provide the reader with already a complete overview of the phase diagram even though the squeezing below the vacuum level is demonstrated in Fig.3(d). We apologize if this led to any confusion, and we now write “As the detuning increases across the second-order DPT, the vacuum separates into two coherent-like states with opposite phases. It will be demonstrated in the following section that at the second-order critical point, the vacuum is, in fact, squeezed.” Furthermore, we have updated Fig. 2(d) with data taken at slightly higher detuning to remove any possible confusion.

Furthermore, we agree with the Reviewer that photon noise is large compared to the intracavity population and noise removal is required to claim squeezing below vacuum. This has already been done when calculating the squeezing in Fig. 3 as detailed in comment 10 and in supplementary section F.

The Reviewer may find the Wigner functions above, obtained with a similar method as the Reviewer’s suggested paper [Nat. Commun. 14, 2896 (2023)], which clearly show the presence of squeezing.

Finally, the Gaussianity check is discussed in comment 19.

Comment 17: *Fig. 2e-f: Again, let us assume that we agree on not scaling κ . The observed jump at $\tilde{\Delta}/2\pi = 0.13$ MHz is a convincing signature of the 1st-order DPT. The curve also becomes more and more non-differentiable at $\tilde{\Delta}/2\pi = -0.04$ MHz, which is a signature of the 2nd-order DPT. However, I am puzzled by the caption: “... calculated from the probability distribution of Φ for $L = 1.41$.” Shouldn’t the different colors represent different values of L ? In addition, I would expect that the slope at the right side of the 2nd-order DPT is independent of L . Why it is not the case here?*

The Reviewer is rightfully puzzled by the caption. The part of the caption “... calculated from the probability distribution of Φ for $L = 1.41$.” is a typo and has been removed. The different colors in Fig. 2(f) correctly correspond to different values of L .

As shown in comment 14, the slope at the right side of the 2nd-order DPT is independent of L (in the rescaled picture). In addition, this can also be seen in Fig.2 (f) where the Binder cumulants overlap for the three curves with the highest value of L , indicating that finite size effects become sufficiently small.

Comment 18: *Paragraph 1, lines 12-15: “The position of this minimum closely aligns with the second-order critical point, i.e., the maximum of the second derivative of the photon number, as shown in Figs. 3(a-c).”*

I am confused by Figs. 3a-c and the interpretation. Firstly, the 2nd-order DPT seems to happen at $\tilde{\Delta}/2\pi = -0.04$ MHz in Fig. 2, but it is indicated to be -0.06 MHz here. Why is that? Secondly, it seems that one data point in Figs. 3b-c corresponds to 5 data in Fig. 3a. Why did the authors average or throw out data? Most importantly, I think that a peak in the 2nd derivative would exist in any curves where two slopes are connected by a smooth transition. I am confused about what information do Figs. 3a-c provide for identifying the 2nd-order DPT.

We apologize for any confusion, the vertical lines in Fig. 2 do not indicate the transition point, but the vertical line at $\tilde{\Delta}/2\pi = -0.04$, just aims just to describe the quantum state at different points. We have updated Fig. 2 with data taken at slightly higher detuning to remove any possible confusion.

As for panels 3(a-c), for robustness (to reduce noise) we compute the derivative using multiple points. As we are considering a second-order derivative, noise effects and uncertainty can be amplified, and this “midpoint” derivative procedure allows for a better estimation of the actual derivative.

Finally, any second-order transition is, by definition, witnessed by a discontinuous behavior of the derivative of the order parameter. For finite-size systems, a large second-order derivative increasing with L captures this phenomenon, as we show in our data.

Comment 19: *Paragraph 6, line 4: “We define the squeezing parameter as the minimal variance*

$\Delta x_\phi^2 \equiv \langle x_\phi^2 \rangle - \langle x_\phi \rangle^2$ of the quadrature x_ϕ spanning all possible ϕ .”

• Paragraph 1, lines 15-17: “This analysis supports the claim that quantum fluctuations play an important role at the second-order DPT.”

• Fig. 3d: If the amplification noise is not removed, the minimum value of Δx_ϕ^2 will be lower bound by the noise photon number. The value should be much larger than 1/2. I guess that the result, $\Delta x_\phi^2 = 1/2$, is obtained by an artificial rescaling of the data. This is not correct because of the convolution relation. To extract the correct squeezing level, the authors need to remove the noise and perform a Gaussianity check in the meanwhile.

If we relax the criteria and look for signatures of squeezing the noise-removal procedure may be skipped. Consequently, it is not appropriate to claim “squeezing below vacuum”. I may ask the authors to show the anti-squeezing data alongside the squeezing data. The anti-squeezing data may help distinguish signatures of squeezing and two closely spaced coherent clouds. I would expect that the product of the squeezing level and the anti-squeezing level is a constant for the former but decreases with $\tilde{\Delta}/2\pi$ in the latter case.

In addition, it is confusing that the minimum variance, Δx_ϕ^2 , is larger than 1/2 for $\tilde{\Delta}/2\pi > -0.04$ MHz. It keeps increasing with $\tilde{\Delta}/2\pi$ onwards. The histogram shown in Fig. 2d and the authors’ theory of two coherent clouds seems to set an upper bound of 1/2 on the minimum variance. May the authors explain why $\Delta x_\phi^2 > 1/2$ can be observed?

We absolutely agree with the Reviewer that claims of squeezing below the vacuum cannot be made without noise removal. That is why in Fig. 3(c), we show the moments having removed the amplifier noise. This follows the same procedure as explained in comment 10 above and is further described in the Supplementary Material. For the reasons mentioned above, this is indeed a measurement of squeezing below the vacuum.

Concerning the Gaussianity check, this would be necessary if we were to assume a Gaussian structure of our state and use it to extract the squeezing parameter. This is done, for instance, in [Nat. Commun. 14, 2896 (2023)], where the Gaussian hypothesis needs to be checked to apply the formula to reconstruct the squeezing parameter. Instead, the procedure we use does not depend on any assumption about the state.

Concerning the increase in the minimum variances, this is because the system’s states are not a mixture of coherent states, but rather a mixture of squeezed states, with an amplifying direction orthogonal to the displacement direction. This can be seen in the Wigner function plotted above in comment 3.

Comment 20: Paragraph 2, lines 6-10: “From a theoretical viewpoint, in the one-photon driven resonator the presence of metastability, and thus criticality, can be argued using a semi-classical model, i.e., assuming a coherent state, and just one-photon loss.”

I do not agree with this claim. Perhaps the authors can explain how the following results can be explained with a semi-classical model:

- (i) The two power-law decay around the 1st-order DPT must have a quantum-mechanical origin, as argued in Phys. Rev. A 93, 033824 (2016) and observed in PRL 118, 247402 (2017) and Nat. Commun. 14, 2896 (2023);
- (ii) The non-classical Wigner quasi-distribution function must have a quantum-mechanical origin, as argued in Phys. Rev. A 39, 4675 (1989) and observed in Nat. Commun. 14, 2896 (2023);
- (iii) The so-called “coherent cancellation dip” must have a quantum-mechanical origin, as argued in J. Phys. A: Math. Gen. 13, 725-741 (1980) and observed in PRL 118, 040402 (2017) and Nat. Commun. 14, 2896 (2023).

What we claimed is that a semiclassical model retrieves multiple steady states in a finite region of parameters. As the quantum theory predicts a unique steady state for any finite-size system, these cannot be stationary. Hence, from a semiclassical model, one can argue that one of these two needs to be metastable and that at some specific detuning, some passage from a high-to-low population phase must occur.

We conclude that the presence of metastability can be argued from a classical model. As the semiclassical model is expected to recover some (although not all) of the features of the quantum system

in the limit of small nonlinearity and far from the transition point, and since we know that in a full quantum treatment these states must be metastable, we can expect that the full quantum model will display criticality in the thermodynamic limit.

We apologize if we have been too brief in explaining this reasoning. We have now expanded the discussion.

Let us stress that we did not comment on any other effect emerging in the driven critical system, but only on the fact that, from semiclassical arguments and knowing the quantum results for finite-size systems, one may expect the presence of a jump in the photon number and thus a critical phenomenon.

Comment 21: *Paragraph 2, lines 14-18: “The region of metastability requires two-photon decays to be correctly captured by a coherent-state approximation. Criticality can only be theoretically obtained within a full quantum picture, as shown in Extended Data Fig. 2.”*

- *Introduction, paragraph 5, lines 8-11 that “Furthermore, we observe the coexistence of multiple metastable states in the vicinity of the first-order DPT, a feature that cannot be captured when neglecting the quantum effects of dissipation.”*
- *Results A, paragraph 1, lines 29-31: “... $U/2\pi = -7$ kHz, and $\kappa/2\pi = 77$ kHz. The other parameters of the experiment are theoretically estimated to be $\kappa_\phi/2\pi = 4.4$ kHz, $\kappa_2/2\pi = 78$ Hz, and $n_{\text{th}} = 0.055$ ”.*
- *Conclusion, paragraph 1, lines 16-18: “Our analysis unambiguously demonstrates the quantum nature of these critical phenomena...”*
- *Conclusion, paragraph 2, lines 3-6: “They unambiguously demonstrate the importance of quantum processes in triggering DPTs, high-lighting the necessity to study DPTs within a quantum framework.”*

I am puzzled why $\kappa_2/2\pi$ is being taken into consideration, given that it is 3-orders of magnitude smaller than κ . I read from the Extended Fig. 2 that a finite κ_2 is necessary to explain the metastable branches around $\tilde{\Delta}/2\pi = 0.13$ MHz. I am not fully convinced with this claim since $\kappa_2/2\pi$ is extremely small. I think it will become more clear if the authors can plot all the three semi-classical branches in Extended Fig. 2.

Alternatively, the authors may compare the Liouvillian gaps with $\kappa_2/2\pi$ as a control parameter. My confusion is that a very small value of $\kappa_2/2\pi$ may not significantly change the Liouvillian gap, while a small Liouvillian gap around $\tilde{\Delta}/2\pi = 0.13$ MHz should allow metastable states [Phys. Rev. Lett. 116, 240404 (2016)].

To the best of my understanding, the non-zero rate of “two-photon loss” is regarded as unambiguously quantum nature by the authors. I do not understand this claim. I think the authors should explain in detail which specific aspects of this experiment unambiguously demonstrate the quantum nature of DPT.

Similarly, I also do not understand this claim in Discussion: “Furthermore, the results we obtained are universal, making our predictions applicable to any open quantum system.” I think the authors should elaborate on the specific results and predictions they are referring to.

An updated version of the extended data Fig.2 is shown below where we now show all of the semi-classical solutions: the states $\pm\alpha$ (have the same photon number), the vacuum (which is always a possible solution for the equation of motion) and a non-zero unstable branch.

Also, note that in this model, the "S" of bistability at the first-order transition has a different shape than in the single-photon drive and is only present when $\kappa_2 \neq 0$, as further discussed in the next paragraph.

The importance of κ_2 is already discussed in main text in the paragraph above Eq. (5). "Two-photon dissipation plays a fundamental role in the correct theoretical description of the observed first-order DPT. While in the previous simulations shown in Figs. 2 and 4, κ_2 played only a marginal role, it now determines the dependence of λ_{1st} with respect to Δ . This is shown in greater detail in the Extended Data Fig. 7. Given the sensitivity of λ_{1st} to very small changes in the value of κ_2 , measuring λ_{1st} is a promising tool for determining κ_2 in Kerr-cat based quantum devices". Furthermore, extended data figure 7 shows the effect of small changes in κ_2 with respect to the original value $\kappa_2^{(0)}$. While (a) the photon number and (b) λ_{SSB} are only marginally affected by a change in κ_2 , (c) λ_{1st} is particularly sensitive to its value, especially at large detuning where the bright phase is metastable. (d) This dependence becomes more pronounced as the detuning is increased. The chosen detunings in (d) correspond to the vertical dashed lines in (c).

Finally, the universality claim resides in the presence of symmetry, i.e. all spontaneous breaking of Z_2 symmetry is expected to occur in the same way. We have clarified this point.

Results B: Dynamical properties

B1: Second-order

Comment 22: Fig. 4c. I find the parameter regime here puzzling. According to Figs. 2-3, the potential regime for the 2nd-order DPT is in the range $\tilde{\Delta}/2\pi = -0.07$ -0 MHz. However, the shown Liouvillian gap indicates that it may happen at $\tilde{\Delta}/2\pi = 0.04$ MHz and onwards. Are these results consistent with each other?

In addition, I am puzzled that the Liouvillian gap, λ_{SSB} , in the range of $\tilde{\Delta}/2\pi = 0$ -0.14 MHz is different from the Liouvillian gap, λ_{1st} , in Fig. 4. Is there a subtle difference between the definitions of λ_{SSB} and λ_{1st} ? Are the different values of them expected?

We apologize for any confusion here. In this part of the article, we are characterizing the spontaneous symmetry breaking, a dynamical consequence of a second-order transition.

The second-order DPT is indeed occurring, according to our data, around $\tilde{\Delta}/L \simeq -0.07$ MHz. The Liouvillian gap, however, does not directly indicate the critical point but rather captures the decay time of $\rho_{\pm\alpha}$. At the critical detuning, this timescale is ill-defined (the critical detuning is associated with an exceptional point). Furthermore, around the critical point, other effects (such as the saturation of the two-level system) prevent accurate estimations of the Liouvillian gap associated with spontaneous symmetry breaking. Nonetheless, this is not the main point, as all the data show a clear trend where, increasing L , the SSB becomes more and more visible.

Notice also that, compared to the first-order transition where the minimum of λ_{1st} coincides with the point where the transition occurs, this is not the case for the second-order DPT. λ_{SSB} and λ_{1st} , are completely different objects, both theoretically (one signals the occurrence of SSB and the other of a first-order transition) and experimentally (they can be measured separately, as we shown in our

article). In this regard, they can be different. For instance, one can add engineered noise that increases $|\lambda_{\text{SSB}}|$ (making loss of coherence faster) without changing $\lambda_{1\text{st}}$.

B2: First-order

Comment 23: Fig. 5a-c. How is the initial “bright” state prepared in the experiment?

This is explained in the supplementary material, in particular with the scheme in Fig. S6. We now better refer to this section in the caption.

Comment 24: Fig. 5d. The first data point of the left panel is different from the others. Is the initial state prepared in the bright state here while in the vacuum state for the panels?

We apologize for any confusion. These measurement points are obtained after the drive has been on for a very long time, ensuring that the average dynamic has reached the steady state (here t refers to the measurement time, not the dynamic time). Whether the system is in the bright or vacuum phase is then simply a probabilistic event, dependent on the relative weight of the two phases. We have now clarified this in the caption.

Comment 25: Fig. 5e. The Liouvillian gap, $\lambda_{1\text{st}}$, seems to close for a finite region ($\tilde{\Delta}/2\pi = 0.12\text{-}0.2\text{ MHz}$), which does not resemble a 1st-order DPT ($\lambda_{1\text{st}} = 0$ only at one point). Is this result expected?

Notice that there is a cut in the figure, and the transition is much sharper when plotted with the same scale as the right part of the panel (the minimum looks like a single point). This is shown in the figure below and in Fig. S7 in the supplementary.

This is indeed the expected behavior, as the theory curves match the experimental data.

Comment 26: Paragraph 6, lines 12-13: “... by fitting the data by a power law, we find that $A(T) \propto T^x$.”

• Paragraph 6, lines 16-18: “Our analysis confirms the theoretical prediction [48] and other experimental verifications [7].”

• Fig. 6

I am also confused by the interpretation here. The main conclusion of Refs. [48,7] is the two power-law behavior, but the authors observed a single power-law behavior here. This seems to be a negative result of Refs. [48,7]. Is it because of the two-photon drive? Should we expect a two power-law for two-photon drive?

On the other hand, two power-law has been reported before [PRL 118, 247402 (2017) and Nat. Commun. 14, 2896 (2023)].

I am confused about what new information does Fig. 6 provide compared with the existing results?

Our results are not in contrast with previous literature, but rather the analogy to the other results in literature. We observed the long-time behavior of the power law, confirming the presence of this feature also for the 1st order transition. This was done in the spirit of the discussion in [Phys. Rev. A 93, 033824 (2016)]. We did not check the shorter-time behavior, as we were interested in demonstrating that the hysteresis was associated with λ_{1st} and not with λ_{SSB} . Indeed, λ_{SSB} is small and different from λ_{1st} , but we do not see its effect on hysteresis. We have now clarified this point.

*** Reply to Reviewer 2 ***

The manuscript by Beaulieu, Minganti, et al. presents the experimental observation of both second- and first-order dissipative phase transitions in a two-photon driven superconducting Kerr resonator. The authors investigate the system's steady states and dynamics by adjusting the drive-cavity detuning at different input powers and monitoring the leaking cavity field in real time via heterodyne detection. They observe a continuous transition characterized by critical slowing down (CSD) of the dynamics and a squeezed vacuum state near the critical point. Additionally, a discontinuous transition is observed which is probed by a combination of CSD and hysteresis measurements between vacuum and bright phases. The experimental findings are supported by an exact master equation calculation encompassing single-, two-photon, and dephasing losses. The observed "quantum jumps" between vacuum and bright phases constitute clear experimental evidence of the significant role of quantum fluctuations in the system.

The manuscript is technically sound and well-written. The authors provide extensive supplementary material with technical details on the device, data evaluation, and theoretical model. To the best of my knowledge, this work presents the first observation of both first- and second-order dissipative transitions in (two-photon driven) Kerr resonators. Due to the pivotal role of quantum fluctuations, their observations offer a promising route for quantum-enhanced sensors, which further exploit the critical nonlinear behavior of phase transitions. However, I have some questions and remarks before recommending publication:

We thank the Reviewer 2 for the detailed reading of our paper and its appreciation. We reply to the Reviewer's comment below.

Main Remarks:

1. In the main text, the authors should precisely elaborate on the role of the different energy scales (Δ , U , G) and dissipation scales (κ , κ_2 , κ_ϕ) for the observed second- and first-order transitions.

- Can the authors identify a microscopic mechanism giving rise to dissipative metastable states close to the first-order dissipative transition? Is it related to effective decay channels for collective excitations, as discussed in Refs. [4] and [20]?

- Can the authors provide an analytic estimate for the critical point of the second-order phase transition (either in the main text or the supplementary materials)? Is it solely determined by a competition between G and Δ , or do additional energy scales play a significant role?

In the context of the present transition, the microscopic transition rate is indeed determined by an "effective heating and tunneling rate" which dictates the switching rate between the metastable branches. This effective rate emerges from the one- and two-photon dissipation mechanisms, as well as dephasing, and sometimes takes the name of quantum heating. In this "coarse-grained" picture, these terms can indeed be seen as emergent decay rates for collective excitations.

As for the critical point, an exact analytic estimation using the quantum theory is possible only if $\kappa_\phi \rightarrow 0$ and $n_{\text{th}} \rightarrow 0$. Furthermore, one also needs to neglect the presence of effects such as the two-level saturation effectively changing the photon dissipation rate for few photons in the resonator.

Instead, we use an extrapolation of the semiclassical formula in Eq.(6) to derive the critical point, once the thermodynamic limit is taken. We obtain $\tilde{\Delta}_c \simeq -65.5$ kHz. This estimate closely matches the experimental data $\tilde{\Delta}_c \simeq 70$ KHz. We now briefly comment on this in the main text.

2. The authors should provide more details on the experimental errors and uncertainties of their measurements. In particular:

- How large is the sensitivity of their heterodyne setup, i.e., how large is the minimal detectable photon number n_{ss} in the context of Fig. 2?

- Are there significant systematic errors in the (inferred) values of the quadratures I and Q ? What about the reconstructed Husimi distributions?

- The authors should provide (statistical) error bars for the minimal variance Δx_ϕ^2 presented in Fig.

3(c). These can be obtained via resampling methods, such as jackknife or bootstrapping.

Fig. 2 shows the reconstructed number of photons, i.e., erasing the measurement noise. In principle the limit on the heterodyne setup is set by the number of bits in the ADC and the amplitude. For our 12bit for 1Vpp, we have a resolution is 122 μ V which would coincides to an average of roughly 0.2 photons in the resonator. However, since we the signal is convoluted with white noise larger than a few least significant bit, we have different distribution on the bit levels depending on the averaged signal such that we can detect much smaller peaks with sufficient averaging. In this work, we collect many datapoints and average enough to easily resolve signals on the order of 0.01 photon. For a number of independent measurements of the order of $M = 10^5$ (the amplifier noise is white, so all measurements are independent), the error on the reconstructed n_{ss} is often smaller than the point size .

The most problematic systematic errors are due to frequency shifts. These become especially evident when we are measuring the photon number at the transition point, as characterized in Fig. S5 of the supplementary material. Other sources of systematic errors include the estimation of the gain (which, however, we find in good agreement with the theoretical results) and fluctuations in the pump power.

As for the Husimi distributions, these data are convoluted with the amplifier noise. Here too, the main source of noise (outside the amplifier) is frequency shifts, with calibration errors on gain also playing a role.

In the current version of the manuscript, we provide more details on the noise and error estimation, and we have also added error bars in Fig. 3(c). As now explained in the caption, the error bars are obtained via a resampling method of bootstrapping, as proposed by the Reviewer.

3. There seems to be some degree of circular argumentation when comparing the results with the Liouvilian theory in the discussion section: After stating the agreement between theory and experiment ("We framed and interpreted our results within the formalism of the Liouvilian theory [...]"), the authors state that their agreement showcases unambiguously the role of quantum fluctuations and dissipation ("Our analysis unambiguously demonstrates the quantum nature of these critical phenomena [...]"). However, they later claim that the agreement between theory and experiment also validates their Liouvilian model ("As such, our work validates the Liouvilian theory that underlies our findings."). I would strongly suggest the authors rephrase these conclusions to avoid potential misunderstanding. For example, they can elucidate the prominent role of quantum fluctuations in their system by highlighting experimental observations such as the "quantum jumps" between the different phases. To my understanding, such observations can only be captured by a full quantum mechanical description correctly incorporating quantum fluctuations and thus would escape a semiclassical approach. A suitable reference in this direction is Phys. Rev. A 102, 063702.

We agree with the Reviewer here. We have changed the manuscript accordingly.

Additional Remarks:

Results section

- Can the authors comment on the degree of experimental tunability they have on the detuning Δ , Kerr nonlinearity U , and two-photon drive G ?

The detuning $\Delta = \omega_r - \frac{\omega_p}{2}$ depends on the resonance frequency ω_r and the pump frequency ω_p . By adjusting the flux bias in the SQUID, we can tune the resonance frequency ω_r . In our experimental setup, to limit the effect of flux noise in the symmetric SQUID, it is best to keep the flux $F = \frac{\pi\Phi_{DC}}{\Phi_0}$ between 0 and $\frac{\pi}{3}$. This range allows us to tune the resonance frequency between 4.24 GHz and 4.37 GHz (see supplementary Fig.S3 (a)). On the other hand, the pump frequency ω_p is constrained only by the instrument and can range between 0 and 18 GHz with our source. Consequently, the detuning $\Delta/2\pi$ could therefore range from -4 GHz to 4 GHz, far beyond the scale of the experiment. The minimal change in detuning is limited by the pump, and we estimate that we can safely change Δ in steps of $\simeq 10$ Hz.

The Kerr nonlinearity U can be tuned by changing the participation ratio of the cavity inductance

to SQUID inductance through a change in the flux bias. Using the measured circuit parameters and equation S30, and constraining the flux F between 0 and $\frac{\pi}{3}$, the Kerr nonlinearity U can be varied between 4.6 kHz and 32 kHz. This change, however, also entails changes in κ , κ_ϕ and κ_2 .

The two-photon drive G scales with the amplitude of the pump tone at ω_p . In our experiment, it varies between $G = 65.5$ kHz to 103.5 kHz. Considering our operating flux point and our experimental setup, the pump amplitude can be straightforwardly increased to reach a G of 200 kHz. Higher values could be reached by adding an amplifier to further increase the amplitude of the pump tone or by increasing the flux to achieve more efficient three-wave mixing. The bottleneck will be the point at which the superconductivity of the junction breaks. The smallest increment in the pump amplitude is set by the number of bits of the DAC of the pump, which gives a smallest possible increment on the order of 5 Hz.

- The authors should elaborate on the deviation of the Binder coefficient from one for small systems. Can this be attributed solely to finite-size effects or is there also a (technical) bias between the two Z_2 symmetry broken configurations observed?

This is indeed a finite-size effect. The deviation of the Binder cumulant trend is observed also in numerical simulations of the model. Nonetheless, let us also note that the estimation of the phase is a higher-moment reconstruction, and as such, it is noisier for a smaller-sized system. We now clarify this point in the main text.

- To provide better comparability with other platforms, the authors should introduce a squeezing parameter and provide a quantitative estimate of the maximal degree of squeezing in dB (decibel) in Fig. 3 (c), and its statistical uncertainty.

We agree with the Reviewer and now plot a squeezing parameter in dB with error bars obtained with bootstrapping.

- The following statement is not completely self-explanatory, and merits further clarification or a suitable reference: “From a theoretical viewpoint, in the one-photon driven resonator the presence of metastability, and thus criticality, can be argued using a semiclassical model, i.e., assuming a coherent state, and just one-photon loss”.

As also discussed in the Reply to Reviewer 1, this may have been too cryptic. What we wanted to claim is the following: a semiclassical model retrieves multiple steady states in a finite region of parameters. As the quantum theory predicts a unique steady state for any finite-size system, these cannot be stationary. Hence, from a semiclassical model, one can argue that one of these two needs to be metastable and that at some specific detuning, a passage from a high-to-low population phase must occur. We conclude that we can argue the presence of metastability from a classical model. As the semiclassical model is expected to recover some (*but not all*) of the features of the quantum system in the limit of small nonlinearity and far from the transition point, and since we know that in a full quantum treatment, these states must be metastable, we can expect that the full quantum model will display criticality. We apologize if we have been too brief in explaining this reasoning. We have now expanded the discussion.

- Can the authors provide a physical argument on why the hysteresis-loop area in Fig. 6(e) decreases both with decreasing L and increasing probing time T ?

The hysteresis area in the regime we considered is mainly characterized by the “breakdown” of adiabaticity associated with λ_{1st} . The smaller λ_{1st} , the larger the area expected to be. Furthermore, increasing T allows for the system to jump to a steady state with higher probability. In this sense, combining the two effects leads to the prediction that, increasing L and decreasing T , the area of hysteresis increases. This plot also shows how only λ_{1st} is associated with hysteresis. We now comment on this fact in the main text.

Discussion section - Can the authors elaborate on the metrological usefulness of the observed vacuum-state squeezing and quantum jumps in the context of quantum-enhanced sensing of magnetic fields and other (technologically) relevant quantities?

The high susceptibility developed in the proximity of phase transitions makes critical systems compelling candidates to realize high-precision sensing devices. Critical systems are already widely used in sensing applications, two notable examples are particle detectors based on bubble chambers and transition-edge photodetectors. Notice that, although the physics behind these devices is quantum, their working principle is energy deposition and they do not follow an optimal detection strategy from a quantum metrology viewpoint. However, recent theoretical and experimental works show that optimal quantum sensing strategies can indeed be realized using quantum systems in the proximity of phase transitions. This field is currently very active and substantial efforts are dedicated to the design and implementation of optimal critical quantum sensing protocols. Accordingly, the development of experimental platforms where phase transitions can be controlled is an important step for quantum metrology.

In particular, the device analyzed in this work has been proposed [npj Quantum Inf. 9, 23 (2023)] as a platform to implement optimal critical quantum sensing protocols for magnetometry and for superconducting-qubit readout. The working principle of the proposed protocol is the following: the resonator frequency ω_r is a critical parameter, as it establishes the value of the pump-to-resonator detuning $\Delta = \omega_r - \omega_p/2$. By varying Δ the system can be forced to switch between the vacuum and a symmetry-broken state. Then, in the proximity of the transition point, the system susceptibility to small variations of ω_r is critically enhanced. As theoretically shown [npj Quantum Inf. 9, 23 (2023)], this susceptibility can be used to achieve a quantum-enhanced estimation of ω_r . In turn, the effective resonator frequency depends on the magnetic flux threading the SQUID. Hence, once the device is calibrated, there is a one-to-one correspondence between the estimation of ω_r and of the magnetic field intensity. The same applies in the case in which the magnetic flux is controlled, but the resonator is dispersively coupled to a qubit. In this case, a change in the qubit state induces a (known) frequency shift. A parameter-discrimination protocol between the two possible values of ω_r allows one to infer the qubit state.

We would like to stress out that these protocols are *conceptually different* from the standard passive approach where, for example, a squeezed state is generated and then used as a probe. In a recent work [arXiv:2402.15559 (2024)] it has been demonstrated that critical quantum sensing protocols based on the of 2-nd order phase transition (which was observed for the first time in the submitted manuscript) can outperform any passive strategy.

*** Reply to Reviewer 3 ***

The authors observe first- and second-order dissipative phase transitions in the squeezed Kerr resonator. The paper is easy to read and the authors are experts in their fields. The topic of the paper is also important and widely relevant.

Dissipative phase transitions are important and hard to study experimentally due to the high tunability and challenging parameter range required. Experiments are, therefore, highly valuable and of foundational importance. This work is interesting because it is a good example of theorists and experimentalists working closely together, probably making something larger than what can be achieved independently. However, I am taken aback by the authors' claims regarding the experimental verification of "unambiguous" quantum effects. I find that all observations made, even if interesting and valuable in themselves, admit straightforwardly a classical explanation.

We thank the Reviewer for their critical reading and the appreciation of our work.

We agree on the possible classical nature of some of the phenomena when taken singularly (as explained, for instance, in the seminal works by Dickman and co-authors). For example, the jump time rate may be associated with a latent temperature and a classical underlying model.

We have now strongly tempered our claims on the quantum nature of the phenomena and have extended the discussion to include an alternative classical explanation.

I agree that the observation may agree with a quantum Lindbladian treatment, but that is not sufficient to claim things like

"Our quantum treatment, based on the Liouvillian spectral theory, accurately reproduces the experimental data, demonstrating the quantum nature of these phenomena" or "The region of metastability requires two-photon decays to be correctly captured by a coherent-state approximation. Criticality can only be theoretically obtained within a full quantum picture, as shown in Extended Data Fig. 2a" or "Our analysis unambiguously demonstrates the quantum nature of these critical phenomena, showing that quantum fluctuations and quantum dissipative processes are the main drive of the observed transitions. Furthermore, the results we obtained are universal, making our predictions applicable to any open quantum system."

These statements are incorrect and far too strong to be acceptable regarding the supporting evidence provided.

We apologize for any misunderstanding, but the Liouvillian theory reproduces the experimental data (for the sizes we were able to efficiently and precisely simulate numerically). The solid lines in Figs. 2, 4, and 5 represent the numerical results obtained with the Liouvillian theory and match well with the experimental data.

As for the two-photon loss, we discuss its importance for a semiclassical analysis, and in Extended Data Fig. 7, we also show its significance in correctly determining the switching rates.

Nonetheless, we have modified the main text to clarify what we mean and have significantly lowered our claims on the quantum nature of the phenomena.

Decisively, the experiment is done in a classical regime where the nonlinearity is weaker than the trivial dissipation ($U \ll \kappa$). Note that the quantum nature of an experiment is washed away almost completely in almost any situation if the linewidth of the energy levels ($\sim \kappa$) is larger than the spectral (nonlinear) features ($\sim U$) of the Hamiltonian. This is the case for these experiments in particular. This is especially relevant for theoretical claims regarding quantum information, which are made in this paper.

I also think that the presence of two-photon dissipation can 1) have a perfectly classical origin in nonlinear noise mixing and 2) many spurious effects other than two-photon dissipation could cause the instability of the bright phase at large detuning (especially since there is no quantitative agreement between the quantum model and the data). Consequently, I don't think Data Fig.2 is convincing and the claim of "unambiguity" is unsustainable.

We agree with the referee that, in general, $U \ll \kappa$ means an almost complete lack of quantum properties. However, some of these properties may manifest around the critical points.

For instance, concerning the use of quantum information protocol, we agree that it is surprising, but quantities such as quantum Fisher information predict that, at the second-order critical point, the system is characterized by Heisenberg scaling in sensitivity, despite the presence of large one-photon loss. This is discussed more in detail in [npj Quantum Information 9, 23 (2023)], where these data are theoretically shown. As Heisenberg scaling cannot be obtained using classical resources, we argue that some quantum effect must be present around the critical point, despite the very dissipative nature of the system. In this paper, we show for instance squeezing below vacuum, a resource that is known to provide an advantage in quantum metrological protocols.

Concerning the two-photon dissipation, it is true that classical nonlinear damping processes can occur. However, we find very good qualitative and quantitative agreement between the model with two-photon loss and the experimental data (the solid lines of theory are provided up to detunings of order 1 MHz and match the experimental data). We apologize if these were not so visible.

Even if the authors would have achieved quantitative agreement between experiment and data, this would still not have been enough to support the claim that purely quantum effects were observed since alternative explanations have not been ruled out (nor discussed). The absence of quantitative agreement is in itself preoccupying.

Again, apologies if it was not clear, we have a quantitative agreement. We now discuss alternative explanations and, in general, why we opt for the explanation in terms of the Liouvillian that we provide.

The authors also insist in the observation of quantum jumps. The telegraphic ("jump") signal observed in this work is typical of a Brownian particle in a classical potential with more than one minimum. While I find the choice of a quantum treatment is a matter of taste, it is not at all a necessary description for this experiment.

Therefore, I find that not enough control experiments have been made to support the strong claims, and I don't think experiments in this regime can convincingly be proven to be more quantum than any other classical effect, which ultimately must admit a description in terms of open quantum systems. I do not think it is in the authors interest to insist in the quantum character of this otherwise very beautiful work. If the authors wish to insist, the authors can try to explain (not to me, but in the paper) or define what they mean by "quantum". But I think the solution is to claim something like: "While we cannot rule out a classical explanation, there is a qualitative agreement with a simple yet rich full quantum model." This would be a fair claim.

I remain open to receiving a comeback from the authors in case they find I have missed their main point. I do wish to advance that I am not inclined to consider things like "squeezing below vacuum" to be necessarily quantum in origin. This also has a clear classical counterpart.

We have changed the text following the Reviewer's suggestion, that now state: "a classical explanation where multiple temperature temperature and noise effects compete could result in similar trends. This is the theory of quantum activation, where the non-commutative nature of the system's operator results in a heating-like effect that is captured by a corresponding classical model."

A last point: where is κ_2 coming from? is this engineered dissipation? Not by any additional drive, it seems. Or is it spurious? Was this explained and I missed it? What is the microscopic mechanism the authors think it can be attributed to?

Although we cannot assign a precise mechanism to it, its source is probably the process of downconversion occurring between the resonator and the feedline, mediated by the SQUID, and the presence of a long-lived stationary mode in the feedline. If this is the case, the stationary mode induces some sort of Purcell effect, resulting in the two-photon dissipation.

In conclusion, the most probable source of two-photon dissipation in the resonator is the combined presence of a squeezed bath and a lossy stationary mode.

Reply of manuscript NCOMMS-24-15284-T

*** Reply to Reviewer 1 ***

I thank the authors for clarifying their methods, and for providing more plots that help me understand their work. In the revised manuscript, the authors have weakened the strong claims that attracted most comments in the previous round of peer review. I agree with the authors that the signatures of the dissipative phase transition (DPT) are observed. The investigation of the 2nd-order DPT is indeed novel and interesting, especially the underlying symmetry and its spontaneous breaking.

I regret that my major concern regarding the thermodynamic limit is not directly addressed in the rebuttal letter. I therefore would remain my doubt on the correctness of the scaling. After reading through the revised manuscript, I think that the claims such as “fundamental role of quantum fluctuation” would still need more support from the data. More measurement data are also required to support the Z_2 symmetry story. In these regards, I maintain my opinion that the data do not fully support the conclusions. It would require a substantial revision of the story and addition of the supporting data. My detailed comments are listed as follows:

We thank Reviewer 1 for their in-depth review of the paper and for acknowledging the novelty and significance of our work, particularly the investigation of the second-order DPT. Below, we provide our responses to the Reviewer’s questions.

Specifically, we offer a very detailed explanation regarding the correctness of our scaling approach, supported by additional numerical simulations. We also elaborate on how our measurements strongly support the interpretation of spontaneous symmetry breaking. Finally, we have updated the manuscript to provide a more nuanced interpretation of the role of quantum fluctuations. Below, we explain how this interpretation is supported by the experimental data. We hope that, in light of these modifications and further explanations, the Reviewer will find that the data sufficiently support our interpretation and recommend the paper for publication in Nature Communications.

1. Thermodynamic limit *Let me put it first that I have no doubt on whether there should exist a DPT. The signatures are there. My concerns have always been whether increasing L (i) leads us to the thermodynamic limit, or (ii) they simply confirm the signatures at different parameters.*

While the approach (i) may be the only way towards a “smoking gun” proof of DPT, the approach (ii) is also acceptable depending on how the authors would interpret the data. Following (ii) but make claims as it has followed (i) is not acceptable in my point of view. In case (ii), data with different L are not comparable with each other.

The authors listed two possible ways towards the thermodynamic limit: (a) $U \rightarrow U/L$, $\kappa_2 \rightarrow \kappa_2/L$, and $\kappa_\phi \rightarrow \kappa_\phi/L$, and (b) $\Delta \rightarrow \Delta L$, $G \rightarrow GL$, and $\kappa \rightarrow \kappa L$. However, the experiment does not follow either of them. I would therefore conclude that L plays the role of (ii) and should not be interpreted as the scaling parameter.

I understand that a systemically control the parameters to follow the right scaling is difficult. But I believe that it is mandatory and it is experimentally feasible. Reasonable simplifications may circumvent the technical difficulty under careful clarification. I agree with the authors’ rebuttal that if $\kappa \approx 0$, the current approach of scaling would already be correct. If the authors can carefully explain why κ can be fairly neglected in this experiment (by some transformation maybe?), I would be convinced that comparing data with different L is reasonable.

** I did some quick calculations and have a different opinion on the two approaches, (a) and (b),*

of scaling: Assuming coherent-state approximation in Eq. (2), i.e., $\alpha = \text{tr}[\rho a]$, we have

$$\partial_t \alpha = -i\Delta \alpha - iU|\alpha|^2 \alpha - iG\alpha^* - \frac{(\kappa + \kappa_\phi)}{2} \alpha - \frac{\kappa_2}{2} |\alpha|^2 \alpha. \quad (1)$$

By defining $\tilde{\alpha} = \alpha/\sqrt{L}$, we see that the equation of $\tilde{\alpha}$ is invariant of L if LU and $L\kappa_2$ remain constant

$$\partial_t \tilde{\alpha} = -i\Delta \tilde{\alpha} - i(LU)|\tilde{\alpha}|^2 \tilde{\alpha} - iG\tilde{\alpha}^* - \frac{(\kappa + \kappa_\phi)}{2} \tilde{\alpha} - \frac{(L\kappa_2)}{2} |\tilde{\alpha}|^2 \tilde{\alpha}. \quad (2)$$

In other words, if we keep the parameters LU and $L\kappa_2$ as constants, U_0 and $\kappa_{2,0}$, respectively, we see DPT by comparing $\tilde{\alpha}$ with an increasing L . Alternatively, we may keep Δ/L , G/L , and $(\kappa + \kappa_\phi)/L$ as constants, Δ_0 , G_0 and $(\kappa + \kappa_\phi)_0$, respectively. An equation invariant of L may be obtained by defining $t' = Lt$, i.e.,

$$\partial_{t'} \tilde{\alpha} = -i\Delta_0 \tilde{\alpha} - iU|\tilde{\alpha}|^2 \tilde{\alpha} - iG_0 \tilde{\alpha}^* - \frac{(\kappa + \kappa_\phi)_0}{2} \tilde{\alpha} - \frac{\kappa_2}{2} |\tilde{\alpha}|^2 \tilde{\alpha}. \quad (3)$$

We also see DPT by comparing $\tilde{\alpha}$. However, one has to remember that the time derivative should be scaled by $1/L$ for comparison of time-domain quantities.

Because a mix of the two ways of scaling will also lead to a well-defined thermodynamic limit, I think that it is feasible for the current experiment to follow the right scaling.

We thank the Reviewer for their clear question on the thermodynamic limit and for providing some quick calculations to help us understand their reasoning.

Below, we first clarify that our scaling of the parameters is correct and explain why we disagree with the scaling proposed by the Reviewer obtained using a coherent-state approximation. Secondly, we justify why L is an appropriate scaling parameter and correctly leads to the thermodynamic limit [approach (i) above].

Scaling of the parameters :

Based on a *semiclassical* approximation, the Reviewer proposes two scalings:

$$\text{(I) :} \quad \Delta \rightarrow \Delta L \quad G \rightarrow GL \quad (\kappa + \kappa_\phi) \rightarrow (\kappa + \kappa_\phi)L \quad (4)$$

$$\text{(II) :} \quad U \rightarrow U/L \quad \kappa_2 \rightarrow \kappa_2/L \quad (5)$$

As expected, these two scalings are equivalent. However, as we demonstrate in the two plots below, using numerical simulations of our full quantum model, these two scalings do not result in the expected collapse of curves, which is a key indicator of DPTs in this setup. This is particularly visible near the first-order transition.

The two scalings we propose to reach the thermodynamic limit are as follows:

$$(I) : \quad \Delta \rightarrow \Delta L \quad G \rightarrow GL \quad \kappa \rightarrow \kappa L \quad (6)$$

$$(II) : \quad U \rightarrow U/L \quad \kappa_2 \rightarrow \kappa_2/L \quad \kappa_\phi \rightarrow \kappa_\phi/L. \quad (7)$$

By performing the same numerical simulations with these two scalings, we observe the expected collapse of the curves at both the first- and second-order DPTs for sufficiently large L .

The primary issue with the Reviewer’s proposed scaling is the assumption that it is predicted by the semiclassical approximation. The semiclassical approximation equates the effect of dephasing to that of single-photon loss. However, a detuning mechanism does not result in particle emission, so conclusions drawn from the semiclassical approximation should be approached with caution. More specifically, when discussing DPTs, the difference in the effect of dephasing between a full quantum model and the semiclassical approximation comes from the non-Gaussian nature of the state near the first-order DPT. Near the first-order DPT, the system becomes “bananized” (see, for example, the review by Blais et al. [Rev. Mod. Phys. 93, 025005, 2021] on the presence of these states in cQED setups, and [Phys. Rev. A 94, 033841 (2016)] for Wigner plots). This highlights the importance of correctly treating dephasing using a full quantum model, such as the one we proposed. A more detailed example of the complete failure of the semiclassical approximation near a DPT can be found in Fig. 4 of Phys. Rev. Research 3, 043197.

The Reviewer points out that, even if our proposed scalings of Eqs. (6) or (7) are correct, our experiment does not follow them, but suggests that doing so would be both experimentally feasible and necessary. We agree with the Reviewer that we do not follow the scalings of Eqs. (6) or (7). However, we respectfully disagree with the Reviewer and argue that implementing these scalings is not necessary and would be extremely challenging in practice.

Experimental implementation :

In typical cQED configurations, such as our experiment and those investigating DPTs cited in the main text, it would be extremely difficult, if not unfeasible, to fully implement the proposed scaling. Fully implementing the scaling would require controlling all dissipation rates and Hamiltonian parameters to scale precisely as required. To the best of our knowledge, no experimental protocols exist that would allow to control dephasing without affecting any of the other parameters (dissipation and Hamiltonian terms). If we acknowledge that fully implementing the scaling is not experimentally viable, we are left with two possible scalings that can be achieved experimentally :

$$(I) : \quad \Delta \rightarrow \Delta L \quad G \rightarrow GL \quad (8)$$

$$(II) : \quad U \rightarrow U/L \quad (9)$$

Using the same numerical simulation as above, we plot the effect of these two scalings below

We observe that, of these two scalings, only the scaling of Eq. (8) (used in the experiment) shows the expected collapse of curves at both the first- and second-order DPTs.

Thermodynamic limit :

Even if it would be experimentally feasible to implement the full scalings (Eqs. (6) or (7)), we claim that it is actually not necessary since the chosen scaling of Eq. (8) correctly admits a thermodynamic limit. For both scalings, there exist a characteristic L at which finite-size effects become negligible. This is already shown by the numerical simulations above, where we observe that for sufficiently large values of L , all the curves collapse onto each other even when using the experimental scaling ($\Delta \rightarrow \Delta L$, $G \rightarrow GL$). We provide further support to this claim by plotting below the minimum of the Liouvillian gap λ_{1st} for the full scaling (Eq. 6) and for the chosen scaling (Eq. (8)). For sufficiently large L , we observe that, unsurprisingly, the full scaling follows an exponential behavior, but so does the scaling implemented experimentally. The remaining question, which was asked by the Reviewer in the first rebuttal letter, is whether we reach sufficiently large values of L in the experiment to observe the “true” scaling and not just finite-size effects.

Finite size effect :

Here, we provide justification for our claim that we are already observing the precursors of the effects that emerge in the thermodynamic limit. Our claim is supported in the paper by investigating three indicators of criticality:

- The curves for the photon number overlap near second-order point (see also the previous rebuttal letter);
- The Binder cumulant, a quantity explicitly designed to minimize finite-size effects, aligns with this analysis, giving us confidence that the points we are experimentally observing correspond to

that predicted in the thermodynamic limit (see also the numerical simulations above).

- The minima of the Liouvillian gaps show an exponential scaling.

The first two points were already discussed in detail in the initial rebuttal letter. We therefore elaborate more on the last point. When studying a DPT in finite-size system, Liouvillian gaps show an exponential dependence on L , for large-enough L . Finite-size effects manifest as deviations from this exponential law at small values of L . This can be seen in the numerical simulations presented above. The fact that we experimentally observe an exponential scaling of the minima of the Liouvillian gaps is a strong indicator that we are already witnessing the precursors of the effects emerging in the thermodynamic limit. Since our three indicators are in agreement, we have no reason to believe that the quantities we are investigating are still displaying finite-size effects. This doesn't mean that all observables and quantities have converged (as one could always find a quantity that has not yet converged for any finite value of L), but at least it applies to those we investigate in the main text and upon which we base our analysis. We believe, that providing more results deviating from the expected scaling (those occurring for $L < 1$) would only confuse the reader.

I assume that the authors will successfully justify their scaling towards the thermodynamic limit, there are some other comments:

** I see the second approach above is close to the authors' choice. It would require a different scaling of and the values of time axis in Figs. 4-6 as well as λ_{SSB} and $\lambda_{1\text{st}}$ for comparison.*

The scaling used in the experiment ($\Delta \rightarrow \Delta L$, $G \rightarrow GL$) doesn't require scaling the values of time axis or λ_{SSB} and $\lambda_{1\text{st}}$.

As explained above, we disagree with the second approach proposed by the Reviewer using a semiclassical approximation, as it does not result in the collapse of curves at both the first- and second-order DPTs. The semiclassical approximation provides incorrect intuition regarding the scaling and the relevance of dephasing and other parameters. As such, arguments based on it should not be extended to the computation of time and rescaling. A more detailed example of the complete failure of the semiclassical approximation near a DPT can be found in Fig. 4 of Phys. Rev. Research 3, 043197.

Since L is the correct scaling parameter, we should not rescale the Liouvillian gaps. Both the experiments and numerical simulations agree that we are observing the correct features associated with emergent criticality in the thermodynamic limit. Therefore we maintain that there is no need to include alternative scalings or additional effects. Finally, we note that rescaling the gaps λ_{SSB} and $\lambda_{1\text{st}}$ by L would not change the observe exponential behavior.

** The authors explained that the small range of L is limited by the measurement time, but would choosing a different biasing point solve this problem?*

No, selecting a different biasing point would not resolve the issue of the measurement time.

If the value of U is increased by adjusting the operating flux point, the value of L required to reach the thermodynamic limit also increases. Since our primary goal is to study values of L where finite-size effects are not dominant, this would require larger values of L , where the timescales would remain the same. In other words, if U is doubled, we then need to double L to observe the same effect as we were observing, resulting in approximately the same timescales.

Consequently, if we want to maximize how "far" we go in the thermodynamic limit, there is no advantage in choosing a different biasing point. Increasing U by changing the biasing point would only lead to higher sensitivity to flux noise which is detrimental for very long measurement.

** **Fig. 2b.** I think zoom in of n_{ss}/L given in the rebuttal letter is very informative than Fig. 3a. Indeed, n_{ss} in Fig. 3a does not support the claims regarding the trend with L . I suggest to put the 2nd-order DPT zoom in beside Fig. 2c (the 1st-order DPT zoom in).*

We have updated Fig. 3a to provide n_{ss}/L for $L = 1, 1.29$ and 1.58 near the second-order DPT. Since Fig. 2 already contains a lot of information, we felt it would be clearer for the reader to present this

additional information in Fig. 3a.

** Fig. 3a-c. I still cannot get the message of these plots. First, I think the y-axes should be n_{ss}/L and its derivatives instead of n_{ss} . Second, I think we would not only expect a higher peak of the 2nd derivative, but equally importantly also a narrower linewidth. These features are missing from the data.*

The message of Fig. 3 is that the maximum of the second-order derivative (at the second-order critical point) increases with L and that the position of this maximum also aligns with the maximum of the squeezing parameter.

To clarify the message of Fig. 3, we follow the suggestion of the Reviewer and use the y axis n_{ss}/L and its derivative.

Regarding the linewidth, we agree with the Reviewer that this feature should emerge in the thermodynamic limit. However, even in theoretical curves, capturing the emergence of a Dirac δ -like is nearly impossible. For example, we show below the theoretical prediction for the full width at half maximum (FWHM) (in the rescaled picture) of the second-order derivative peak as a function of L .

The difference in FWHM between the minimum and maximum L is predicted to be of the order of ~ 0.01 MHz. Given the current level of averaging required to observe the second-order peak, we do not have sufficient resolution in our data to capture this feature. Capturing this feature would require acquiring an even greater number of points, and given the slow decrease in FWHM that we observe, it would not have provided a convincing demonstration of criticality.

We also want to emphasize that, based on previously studied examples, this feature is known to emerge only for very large values of L . For example, in the inset of Fig. 8 of Phys. Rev. A 94, 033841 (2016), finite-size feature of the first-order derivative are still visible even for effective values of L far beyond those considered here.

2. Quantum signatures *I am not very skeptical on the claim “squeezing below vacuum” and its indication of “quantum”, because the minimum fluctuation is smaller than the vacuum fluctuation. I know that the Hamiltonian is quantum mechanical, and it fits well with the experiment. However, I would become more critical when it leads to a conclusion such as “quantum fluctuations play a fundamental role”. Questions arises in two categories: (i) How “quantum” it is? What specific “quantum fluctuation” is indicated here? and (ii) what does ‘classical fluctuation’ predict? I think more evidences are needed to support such claims.*

As pointed out by both Reviewers 1 and 3, the claims regarding quantum fluctuations detract from our main focus on first- and second-order DPTs and how the Liouvillian gaps capture them. Consequently, in the revised version of the manuscript, we have removed the unnecessary discussion and moved the mention of quantum fluctuations to the discussion/outlook section. In the discussion section, we now provide more nuanced explanation of how quantum fluctuations could contribute to the observed effects, while clearly indicating that further experiments would be needed to provide undeniable proof. The conclusion “quantum fluctuations play a fundamental role” has been removed.

In view of these changes, we address the Reviewer’s question to justify why it is not an overstatement to say that our results *hint* at the *potential* effect of quantum fluctuations.

Measuring squeezing below vacuum demonstrates that, at least in a specific operating region, the environment of the resonator is sufficiently cold for its behavior to not be dominated by thermal effects. If there were strong “classical fluctuations”, we would expect the noise of these fluctuations to increase the fluctuations along the squeezed quadrature above the vacuum level. For example, we theoretically show in the figure below the effect of adding thermal photons on the squeezed states obtained near the second-order critical point. The red line is simply a reference corresponding to the squeezed variance Δx^2 of the ground state (i.e. without any dissipation and noise $\kappa = 0$, $\kappa_\phi = 0$, $n_{\text{th}} = 0$, $\kappa_2 = 0$). The blue line corresponds to the squeezed variance including dissipation ($\kappa \neq 0$, $\kappa_\phi \neq 0$ and $\kappa_2 \neq 0$) as a function of the temperature (n_{th}). As clearly shown, by increasing the number of thermal photons the state is no longer squeezed below the vacuum ($1/2$).

This suggests that, at least in the operating point where we measure squeezing below vacuum, our system is well shielded and sufficiently cold to avoid significant classical fluctuations. Since there is no apparent reason why this would differ at other operating points, and given that fluctuations are necessary for jumps to be observed, this *hints* at the *potential* role of quantum fluctuations (i.e. zero-point fluctuations).

Finally, we would like to highlight that the results of our experiment need to be considered in the context of the current literature. Although we have not conducted a thorough experimental study comparing the effects of classical and quantum fluctuations on the behavior of our system, other experiments on very similar systems have done so. It is undeniable that current cQED setup can achieve regime where thermal and classical noise sources are negligible relative to quantum noise with both linear resonator (Nature volume 454, pages 310–314 (2008), Nature volume 459, pages 546–549 (2009)) and nonlinear resonator with $U < \kappa$ (Appl. Phys. Lett. 91, 083509, Phys. Rev. B 83, 134501). Nevertheless, it is known that switching events still occur under these conditions (Phys. Rev. A 73, 042108, Phys. Rev. E 75, 011101). In this regime, the observed switching rate can be explained by quantum fluctuations/zero-point fluctuations. The rates can be partially described by a semiclassical model, where quantum fluctuations are incorporated through an effective temperature $T_{eff} = \hbar\omega_r/k_B$. This is the theory of quantum activation, as extensively reviewed in arXiv:1112.2407v1. This theory has been experimentally demonstrated in Rev. Sci. Instrum. 80, 111101, where the authors monitored the switching rate at various temperatures of a driven superconducting nonlinear resonator near a first-order DPT. They showed that near the minimum temperature of their cryostat, the effective temperature matched the predictions based on quantum fluctuations (see Fig. 17 of that paper). This is one of the many experimental works showing that quantum fluctuations can play a significant role in this type of system. Further relevant experimental demonstration can be found in PRL 110, 047001 (2013) : Quantum Heating of a Nonlinear Resonator Probed by a Superconducting Qubit and Nature volume 479, pages 376–379 (2011) : Demonstration of dynamical Casimir effect in a superconducting circuit. To the best of our knowledge, no similar study has been conducted for a *parametrically* driven nonlinear resonator, but these previous results suggest that quantum fluctuations *could* also play a role in the switching rate here, and that is now our only claim.

* **Fig. 3d.** *The authors declined the Gaussianity check and the plot of anti-squeezing because they do*

not assume a Gaussian state to extract the squeezing level. But the question is whether it is a Gaussian state here? If Yes, then a Gaussianity check with zero thermal photon number would be a solid proof of the “quantum” claim. If it is not Gaussian, I am wondering what is role of the “squeezing” operator here? In both cases, the anti-squeezing may still provide useful information for understanding how “quantum” it is.

We have removed the section titled “Quantum interpretation of the transition” and moved any discussion on the possible quantum nature of the transitions to the conclusion as an outlook for future experiments which could strengthen these claims.

For transparency and in the spirit of academic discussion, we have nonetheless provide answers to the Reviewer’s questions. There is no Gaussian state expected here. As for the anti-squeezing operator, it does not provide useful information for understanding how “quantum” the state is. In a nutshell, our analysis in the following shows that (i) the state at criticality should be non-Gaussian; (ii) the anti-squeezed quadrature provides no information on how “quantum” the state is; (iii) regardless of being Gaussian or non-Gaussian, squeezing below vacuum alone is a “solid” witness of non-classicality.

Gaussianity check :

We have two important points here:

- Numerical simulations show that the state is non-Gaussian at the second-order DPT, in the broken-symmetry phase, and at the first-order DPT. At the second-order DPT, deviations from a Gaussian state are due to the presence of Kerr nonlinearity, giving the state an “hourglass” shape (see wigner for $\tilde{\Delta}/2\pi = -0.03$). In the broken-symmetry phase, the state is a superposition of *two* quasi-Gaussian states, so comparing it to a single Gaussian would fail. Finally, near the first-order transition DPT, we expect the state to be a mixture of two non-Gaussian and banana-shaped states (“a process that is sometimes referred to as bananization”, verbatim from [Rev. Mod. Phys. 93, 025005 (2021)] (see wigner for $\tilde{\Delta}/2\pi = 0.13$). These non-Gaussian states can be observed in the theory Wigner plots presented below.

- Despite being non-Gaussian, a state can still have a variance in one quadrature that is below that of the vacuum. In this case, a system is said to be squeezed below vacuum, and this is a property that does not rely on the Gaussianity of the state.

Anti-squeezing :

Concerning the anti-squeezing, we agree with the Referee that if the state was Gaussian, this would give information about the purity, or mixedness, of the state, and an effective temperature of the state could be retrieved. However, we can demonstrate that in our system, the anti-squeezed variance doesn't provide information. Similarly to the previous comment, we theoretically show in the figure below the effect of adding thermal noise on the squeezed variance Δx^2 (left panel), on the anti-squeezed variance Δp^2 (middle panel) and on the product of the two $\Delta x^2 \Delta p^2$ (right panel). In each panel, the red line is simply a reference corresponding to the result obtained for the ground state (i.e. without any dissipation and noise $\kappa = 0$, $\kappa_\phi = 0$, $n_{\text{th}} = 0$, $\kappa_2 = 0$). Looking at the three panels, we see that for the ground state Δx^2 is squeezed ($\Delta x^2 < 1/2$), Δp^2 is anti-squeezed ($\Delta p^2 > 1/2$) and the product of the two saturates Heisenberg uncertainty $\Delta x^2 \Delta p^2 = 1/4$. Consequently, for the ground state the anti-squeezed quadrature is meaningful. We make the similar analysis for the blue lines which correspond to the results obtained including dissipation $\kappa \neq 0$, $\kappa_\phi \neq 0$ and $\kappa_2 \neq 0$) and presented as a function of the temperature (n_{th}). We observe that, while the Heisenberg equality is saturated for the ground state, it is not satisfied when dissipations (κ , κ_2 and κ_ϕ) are included, even at zero thermal photon levels. Despite being squeezed along one quadrature $\Delta x^2 < 1/2$ at zero thermal photon, the product of the two quadrature is $\Delta \hat{x}^2 \Delta \hat{p}^2 \geq 1/4$. This clearly shows that looking at the anti-squeezed quadrature does not give information on how quantum the system is.

In our analysis, measuring the variance of the anti-squeezed quadrature can still be used as a sanity check for the state to be physical, by checking the Heisenberg relation, which we did. Indeed, for completeness and transparency, we still present measurement of the anti-squeezed quadrature ($L = 1$) below. In the top panel, we show the product of the squeezed (labeled Δx^2) and anti-squeezed (labeled Δp^2) variances. The product always satisfies the expected relation $\Delta x^2 \Delta p^2 \geq 1/4$ within error bars. When the detuning is highly negative, the state is pure as the system is in the vacuum, and the equality holds. For the reasons explained above, the Heisenberg inequality is not saturated when going towards the second-order DPT. When the detuning becomes positive, the system transitions to a superposition of two coherent-like states of opposite phases, resulting in a very large variance along the anti-squeezed quadrature. This is shown in the lower panel, where the variance of squeezed (black line) and anti-squeezed quadratures (red line) are plotted independently. A zoom is provided in the inset and plotted with an horizontal black line indicating 0.5 for reference. The detuning corresponding to the maximum $\partial^2 \tilde{n}_{ss}$ is shown by the grey dotted line. We have added this figure to the supplementary material.

Non-classicality :

- For Gaussian states, where the Gaussianity check is meaningful, having (close to) zero thermal photon is not needed for a “solid” proof of quantumness. As an example, one can compare a pure state with low squeezing (e.g., $\Delta x^2 = 0.4$, $\Delta p^2 = 0.625$, such that $\Delta x^2 \Delta p^2 = 1/4$) with a highly non-pure state with larger squeezing (e.g., $\Delta x^2 = 0.2$, $\Delta p^2 = 10^4$). It is well understood in the literature that the second state (a highly non-pure state but more squeezed below vacuum) is “more quantum” than the former, as, e.g., it generates more entanglement when we let it interact with a classical coherent state via a beamsplitter (see, e.g., PRL 94 (17), 173602 (2005)). Let us elaborate more on this point.

Such potential of generating entanglement can be quantified with the so-called Entanglement Potential (EP), see PRL 94 (17), 173602 (2005), which is a measure of non-classicality. The EP does not depend at all on the purity of the state, but only on the variance of the squeezed quadrature. By considering a state $S(r)\rho(n_T)S(-r)$ where $S(r) = e^{r/2(a^{\dagger 2} - a^2)}$ is the squeezing operation and $\rho(n_T)$ a thermal state with n_T photons. For such state, the EP is $EP(r, n_T) = \max \left\{ \frac{r - \ln(1 + 2n_T)/2}{\ln(2)}, 0 \right\}$ (see Eq. 7 of PRL 94 (17), 173602 (2005)). Since $\Delta x^2 = e^{-2r}(1 + 2n_T)/2$ for the squeezed quadrature, we have that $r = \frac{1}{2}[\ln(1 + 2n_T) - \ln(2\Delta x^2)]$, so that

$$EP = \max \left\{ -\frac{\ln(2\Delta x^2)}{\ln(2)}, 0 \right\}. \quad (10)$$

In other words, the amount of non-classicality of a Gaussian state depends only on how much the state is squeezed below vacuum, and not on its purity or effective temperature. Notice that purity does not even put a quantum mechanical constraint on how much a state can be squeezed.

- Our state is not Gaussian, so we do not have trivially access to the effective temperature, or the purity, of the state. However, having squeezing below vacuum is still a clear witness of non-classicality. In fact, if a Gaussian state is entangled, then every non-Gaussian state with the same second moments is necessarily entangled (New J. Phys. 8 51 (2005)). Since the Entanglement Potential of a Gaussian state with the same second moments of our non-Gaussian state depends only on the squeezed quadrature (see Equation above), our argument trivially follows.

Notice that the Entanglement Potential provides an operative way to assess the non-classicality of

a state, directly connected to a feature that is universally recognized as “quantum” in its nature: Entanglement. Other witnesses or measures of non-classicality can be analyzed, e.g., PRL 89, 283601 (2002), but substantially the result is the same. We have therefore shown that, regardless of being Gaussian or non-Gaussian, squeezing below vacuum alone is a “solid” witness of non-classicality.

We want to stress that we do believe that, for *non-Gaussian* states, purity and other features may have a role for quantifying non-classicality. However, defining such role shall require a rigorous and meticulous theoretical analysis which goes well beyond the scope of this article, as the “quantum” feature is not central for the story of this article.

The authors also disagreed that (vi) comparing $\langle a \rangle^2$ with 0 and $\langle a^\dagger a \rangle$, (vii) quantum state tomography, and (viii) $g^{(2)}(0)$ measurements are more difficult measurements. This comment was originally made on whether the current study is “comprehensive”. In principle, I agree with the authors that the histogram they recorded should be adequate to generate these results. Indeed, a plot of these results may be very helpful here to support the “quantum” claim, especially $g^{(2)}(0)$.

The claims regarding “quantum” effects have been reduced and are now discussed with more nuance. Nevertheless, additional support for the outlook on the potential role of “quantum” effects, beyond what has been mentioned above, would always be beneficial. However, we disagree with the Reviewer that $g^{(2)}(0)$ would provide strong indication on the “quantum” claim. From theoretical simulations, we expect a peak in $g^{(2)}(0) > 1$ at both transitions. This photon bunching effect is known to occur in both classical and quantum systems, contrary to anti-bunching $g^{(2)}(0) < 1$. Consequently, this observation would not be helpful in supporting any discussion distinguishing between “quantum” or classical behavior.

One related comment on the state tomography: To the best of my knowledge, there have been two approaches for this purpose: (i) Assuming a small photon-number truncation and fit the density matrix via optimization, or (ii) assuming an analytical state and put in moments. Both methods rely solely on the measured histogram or moments. In Fig. S2, the authors have used $\langle a^\dagger a \rangle$, λ_{SSB} , and $\lambda_{1\text{st}}$ for tomography. It may be acceptable in the sense that all the variables are determined from the experiment, but I think it is very biased since a small or large value of $\lambda_{\text{SSB}, 1\text{st}}$ would already exclude most of the possibilities. I suggest use only moments for a less biased tomography.

We agree with the Reviewer that experimental state tomography is typically performed following (i) or (ii). In Fig. S2, we followed the analysis presented in Nature Communications, volume 14, Article number: 2896 (2023), a reference suggested by the Reviewer for obtaining a reconstructed Wigner function. To avoid any potential confusion, we have clarified both the caption and the main text.

In our approach, we considered first-order momenta ($\langle \hat{a} \rangle = 0$) and second-order one ($\langle \hat{a}^\dagger \hat{a} \rangle \neq 0$), and complemented these results with the Liouvillian analysis. We argue that the Liouvillian analysis constraining the problem is advantageous, as it reduces the likelihood of multiple minima or shallow gradients in the study. Of course, this remains a theoretical reconstruction of the Wigner function from the measurement parameters, not a direct measurement of it.

*** Fig. S3.** *I thank the authors for plotting all the relevant branches of metastability. I agree with the authors that considering a finite κ_2 leads to a better fitting between theory and experiment, so that it is a reasonable choice. What I do not understand is the interpretation of κ_2 : Why a negligibly small two-photon loss would indicate “quantum”? Shouldn’t single-photon loss sounds even more “quantum”?*

We agree with the Reviewer that including κ_2 alone does not make a model quantum, and we apologize if this idea was mistakenly conveyed in our paper. As the Reviewer state, our claim is that from our numerical simulations, the effect of κ_2 is needed to correctly fit the experiment. We have removed any discussion of κ_2 and quantum in the updated version of the manuscript.

I have two more confusions on this plot: First, the claim that “only for $\kappa_2 \neq 0$ that the bright phase becomes unstable” is drawn from the semiclassical calculation. Second, I think the semiclassical calculation indicates that $\kappa_2 \neq 0$ results in hysteresis instead of instability. Third, I am wondering that if a different choice of parameter lead to very different comparison results? I agree that we should consider

κ_2 , but I cannot follow the necessity of “quantum” here.

This section has been removed from the manuscript for clarity, as it was distracting from the main message on first- and second-order DPTs and how our Liouvillian model captures them. As such, this figure is no longer referenced in the main text and has been removed from the supplementary figures. We again only meant that from our full quantum model κ_2 is needed to correctly fit the experiment as explained in the comment above.

For transparency and completeness, we nevertheless address the Reviewer’s questions. Yes, we agree with the Reviewer that the claim that “only for $\kappa_2 \neq 0$ that the bright phase becomes unstable” is drawn from the semiclassical calculation. If $\kappa_2 = 0$, in a coherent-state approximation, the bright phase never jumps to the vacuum. However, if $\kappa_2 \neq 0$, the coherent-state approximation predicts a jump and a region with multiple steady states with different photon number. The Reviewer is correct that there is hysteresis in the region of multiple solutions predicted by the semiclassical approximation. The instability we referred to is the jump that occurs at sufficiently large detuning, where the semiclassical prediction indicates that any high-photon-number state will become unstable and decay towards the vacuum.

The necessity of quantum comes from the fact that the coherent-state approximation is insufficient to accurately predict the position of the transition from high to low population (first-order DPT) . To determine this point, a full quantum model is required. Furthermore, following the intuition gained from the semiclassical approximation (this is also corroborated by simulations of the full quantum model), the quantum model should also correctly include $\kappa_2 \neq 0$ since this term has great influence on the observed hysteresis (λ_{1st}), as shown in supplementary Fig. S7. This only means that from our full quantum model κ_2 is needed to correctly fit the experiment as explained in the comment above.

** In the rebuttal letter the authors states that “In the main text, we have tried to clarify this by specifying that the data are not quantum trajectories but rather collected along a single trajectories.” But I fail to see this change in the revised manuscript.*

I raised up this concern because photon fluctuation itself is broadband. In my opinion, using a narrow-band filter results in classical trajectories that jump from one cloud to another. The so called “quantum trajectories” are indeed very classical. Perhaps “single measurement trace” may be a better terminology?

We apologize to the Reviewer for not making this change sufficiently clear in our previous version of the manuscript. We have clarified the terminology by systematically writing in the caption and main text that we are measuring the quadratures *along quantum trajectories*.

In the new version, we further modify to “single measurement trace”. We only keep the terminology “quantum trajectories” when we referring to the theory, i.e. “rooted in a quantum trajectories interpretation of the Liouvillian dynamics.”

*It leads to the following questions on Figs. 4a and 5d: * Figs. 4a and 5d. In my understanding, the two metastable states, ρ_α and $\rho_{-\alpha}$, should converge to a mixed state $\rho_{ss} = (\rho_\alpha + \rho_{-\alpha})/2$ in the symmetry-breaking region. If this understanding is correct, why the I quadratures jump between two values with a notable time interval? I would expect that ρ_{ss} leads to $I = 0$, considering that filter performs average. Even if the average is not sufficient, I would still expect a fast switching between $\pm I$. Why the trajectory seems to stuck at either of the clouds for a while?*

The Reviewer is correct that in the symmetry-broken region, the steady state is $\rho_{ss} = (\rho_\alpha + \rho_{-\alpha})/2$. This of course implies that $\langle a \rangle = 0$ and both $\langle I \rangle = 0$ and $\langle Q \rangle = 0$ on average.

It is important to keep in mind that each $\rho_{\pm\alpha}$ is a long lived metastable state, which decays to ρ_{ss} in a time $1/\lambda_{SSB}$. This implies that for a single trajectory, the time between jumps is of the order $1/\lambda_{SSB}$. Consequently, we have the following situation :

- A single “short” measurement trace ($t < 1/\lambda_{SSB}$) will show one of the metastable state $\rho_{\pm\alpha}$. The time average of this single short measurement trace will result in $\langle a \rangle \neq 0$.
- A longer measurement ($t \gg 1/\lambda_{SSB}$) will still show that $\langle a \rangle \neq 0$ at each time, but many phase

flips events will be visible during the measurement. Indeed, as long as the acquisition rate is much faster than λ_{SSB} and that the measurement time is $t \gg 1/\lambda_{\text{SSB}}$, we will observe many successive measurements of either ρ_{α} or $\rho_{-\alpha}$ with jumps in between. This is why we see that the trajectory is “stuck” on a cloud for a while.

The fact that $\langle I \rangle = \langle Q \rangle = 0$ results from averaging over many repetition of the experiment. Given the stochastic nature of each experimental realization, the system randomly chooses the phase $+\alpha$ or $-\alpha$, which is why the steady state always has $\langle a \rangle = 0$. A similar result can also be obtained by averaging the signal over a time period during which many jumps occur. Both averaging procedures result in states that are equal mixture of $\rho_{\pm\alpha}$, i.e. $\rho_{\text{ss}} = (\rho_{\alpha} + \rho_{-\alpha})/2$. Consequently to have $\langle I \rangle \simeq 0$ on a single measurement point would require an acquisition rate significantly larger than λ_{SSB} , so that on each sampled point we would recover the average of many jumping event. A study of single quantum trajectories (this time the object ensuing from an interpretation of the Lindblad master equation as a quantum stochastic process) can be found in [Eur. Phys. J Special Topics 226 (12), 2705 (2017)]. There, one sees that, despite *averaging* to zero, *single quantum trajectories* can still display nonzero x and p (I and Q).

If I understand it correctly, the $t = 0$ starts after a waiting time. What is the expected state at $t = 0$? Is it a random choice between ρ_{α} and $\rho_{-\alpha}$? I am wondering whether $\langle a \rangle$ is a more straightforward measure of λ_{SSB} than the autocorrelation? Or, is this measurement takes place after a sufficiently long waiting time where the system has reached ρ_{ss} ? If this is the case, why would autocorrelation function of steady-state measurement predict time dependance?

Yes, the Reviewer is correct that $t = 0$ is after a waiting time. The Reviewer is also correct that in the symmetry broken region, we expect a random choice between the two metastable states ρ_{α} and $\rho_{-\alpha}$. To be even more precise, closer to the first-order phase transition, there is also a probability of being in the vacuum state (see Fig. 5d).

$\langle a \rangle$ as a measure of λ_{SSB} :

The measure of $\langle a \rangle$ is not a more straightforward measure of λ_{SSB} . Since the Hamiltonian is invariant under the transformation $a \rightarrow -a$, one has that $\langle a \rangle = 0$. Experimentally, as explained in the previous comment and noted by the Reviewer in their question, this manifests in the system being initialized with a random choice between the two metastable states ρ_{α} and $\rho_{-\alpha}$, such that $\langle a \rangle = 0$. To further clarify this, we consider the normalized measurement shown below :

Maybe the Reviewer’s suggestion is to calculate the time average of this trace, i.e. $\frac{1}{T} \int_0^T I(t)dt$ for increasing T . However, as shown below, this quantity will simply oscillate before eventually converging to zero (since $\langle a \rangle = 0$) after a sufficiently long time, making it complicated to extract a characteristic time scale.

Maybe the Reviewer's suggestion is instead to initialize the system in the state $|\alpha\rangle$ (or $|\alpha\rangle$) at a time $t = 0$, and continuously measure until a jump to $|\alpha\rangle$ (or $|\alpha\rangle$) is observed. This process could be repeated multiple times and averaged to observe the exponential decay of the state $|\alpha\rangle$ toward $|\alpha\rangle$ (or vice versa). This approach is correct and would give the wanted characteristic time. It is actually very similar to what is done in Fig.5 a and c to measure λ_{1st} . However, for the case of λ_{ssb} it is not ideal since the system will randomly be initialized in $|\pm\alpha\rangle$, meaning half of the measurements would need to be discarded from the average, as $\langle a \rangle = 0$.

Justification of the autocorrelation function to measure λ_{ssb}

The autocorrelation calculation effectively performs what the Reviewer is proposing in a self-contained and mathematically rigorous way. Roughly speaking, the autocorrelation determines, on average, when the next jump will occur by calculating the correlation of the signal with a delayed copy of itself. Consequently, even if the system is, on average, in the steady state, this quantity still captures the switching behavior observed in individual experimental realizations.

For example, if we calculate the autocorrelation for a delay time $t = 0$ ($C_{ss}(0)$), the signal is, of course, perfectly correlated with itself. As we increase the time delay t , since the signal jumps between $|\pm\alpha\rangle$, some sections of the initial signal in $|\alpha\rangle$ will begin to align with sections of the delayed signal that are in $|\alpha\rangle$ (and inversely). This means the signal and its delayed version are less correlated. At very large delay time, the two signals are completely uncorrelated. This can be seen by looking at the experimental autocorrelation shown Fig 4 b. The correlation time of our signal directly corresponds to the time the signal spends in $|\pm\alpha\rangle$ and consequently to λ_{ssb} . Since the autocorrelation function provides the information on the correlation time, λ_{ssb} can be obtained by fitting the exponential decay of $C_{ss}(t)$ (see Fig 4 b).

From a pragmatic point of view, the autocorrelation function is our tool of choice. (i) It discards no data, giving a better precision. (ii) It measures the correlation time λ_{ssb} of the signal without any implicit assumption: the formal validity of this statement is given in Supplementary material section C. (iii) It only requires a single, very long measurement trace to observe enough jumps (note that the figures above and in the manuscript show only segments of the entire trace).

3. Z_2 symmetry *The spontaneous breaking of Z_2 symmetry is very interesting, but I think it needs more data support. Although λ_{SSB} provides sort of information, I think a direct observation related to the symmetry breaking is missing. In the symmetry-breaking phase, the two SSB states, ρ_α and $\rho_{-\alpha}$, would acquire a very long lifetime. The $\langle a \rangle$ measurement will show a very slow decay towards zero. Is this consistent with the measurement?*

Yes, this is perfectly consistent with the measurement. As discussed above, $\langle a \rangle$ is always zero and therefore cannot be used to estimate λ_{ssb} . Instead, the autocorrelation provides the information on the decay rate of $\langle a \rangle$, if we were to initialize the system in, e.g., $\rho_{+\alpha}$ and let it decay to ρ_{ss} . From the value of λ_{ssb} extracted from the autocorrelation calculation, we can clearly see that the two SSB states, ρ_α and $\rho_{-\alpha}$, acquire a very long lifetime. This observation is reported in Fig.4 a,b and c. The gap λ_{SSB} gives the lifetime of ρ_α and $\rho_{-\alpha}$. As shown in Fig 4 a, as we increase the detuning from the second-order critical point, the lifetime of $\rho_{\pm\alpha}$ increases (remark the different x-axis in the different panels). Furthermore, as we increase the value of L , the lifetime becomes longer as shown in Fig.4 c. We further show in the inset of panel of Fig.4 c that the maximum lifetime of $\rho_{\pm\alpha}$ actually follows an exponential law $\lambda_{SSB} \propto \exp(\alpha L)$ with the scaling parameter L demonstrating that acquire very long lifetime $\rho_{\pm\alpha}$.

As commented above, I think $\langle a \rangle$ is a straightforward measure of λ_{SSB} ? What is the consideration that leads to the choice of the autocorrelation function?

As further detailed above, $\langle a \rangle$ is always zero because the Hamiltonian is invariant under the transformation $a \rightarrow -a$. Since $\langle a \rangle = 0$, it cannot be directly used to measure λ_{SSB} .

From a pragmatic point of view, the autocorrelation function is our tool of choice. (i) It discards no data, giving a better precision. (ii) It measures the correlation time λ_{ssb} of the signal without any implicit assumption: the formal validity of this statement is given in Supplementary material section

C. (iii) It only requires a single, very long measurement trace to observe enough jumps.

** Fig. 4c. In my understanding, the range for a small λ_{SSB} should cover the origin, since both positive and negative Δ would allow 2nd-order DPT. Two questions: (i) Where is the right boundary of the symmetry-breaking region? (ii) Why does the symmetry-breaking region looks solely positive?*

Before answering (i) and (ii), we must explain why we respectfully disagree with the Reviewer.

There is no 2nd-order DPT at positive detuning. The second-order DPT occurs at a single point in parameter space ($\tilde{\Delta} \simeq -0.7$ MHz for our parameters). At this point, the system shows a second-order discontinuity in the steady state in the thermodynamic limit. This discontinuity does not occur elsewhere, and thus, there is no second-order DPT in the positive detuning region.

This confusion might arise from a direct comparison between $\lambda_{1\text{st}}$ and λ_{ssb} . A second-order DPT normally leads to SSB, which manifest in the thermodynamic limit. For finite-size systems, we observe the precursors of this eventual phenomenon. The point where λ_{SSB} is minimal does not necessarily coincide with the point where the second-order DPT occurs. This differs from the critical slowing down of the first-order DPT, where the minimum $\lambda_{1\text{st}}$ occurs at the critical point. The minimum can actually be significantly removed from the critical point as in our system. Showing that, however, the minimum of $\lambda_{\text{SSB}} \rightarrow 0$ with increasing L strongly indicates that SSB is emerging.

(i) The theoretical data in [PRX Quantum 4, 020337 (2023)] suggest that the right boundary should occur in the region of hysteresis of the first-order DPT. Thus, SSB can also occur in the metastable state associated with the first-order DPT. In our system, measuring the rate of SSB in the region of metastability is extremely difficult since $\lambda_{1\text{st}} > \lambda_{\text{SSB}}$ in that region. This implies that when initialized in $\rho_{\pm\alpha}$, the system will typically decay to the vacuum before a phase flip event occurs.

(ii) The scaling towards the thermodynamic limit of λ_{SSB} is point-dependent. As we also show in the main text, not all curves converge to their thermodynamic value at the same rate. Nonetheless, in all the region for $\tilde{\Delta} \gtrsim -0.7$ MHz we observe that, increasing L , λ_{SSB} decreases.

4. Additional comments

** Fig. 1a. I know from the rebuttal letter that the red-dashed lines are covered by the blue lines. I would recommend making the dashed lines visible in the hysteresis region for clarity. Alternatively, replacing the label of “metastable” by “hysteretic” may be more appropriate?*

As shown in the schematic in the rebuttal letter, the x-axis can be understood as Δ . Because Δ can be either positive or negative, I think there should exist another critical point of the 2nd-order DPT on the positive side. Two SSB branches would exist between them, and may be drawn for clarity

I understand that the authors want to keep the schematic general, but I do feel that the schematic in the rebuttal letter, with Δ as the x-axis and $\langle a \rangle$ or $\langle a^\dagger a \rangle$ as the y-axis, is much clearer to me.

We thank the Reviewer for their suggestion and have updated Fig. 1a accordingly. It now includes both $\langle a \rangle$ and $\langle a^\dagger a \rangle$ as function of Δ . We have also replace “metastable” by “hysteretic” and further clarified the difference between the steady state, the rates given by $\lambda_{1\text{st}}$ and λ_{ssb} .

We disagree with the Reviewer concerning the second-order DPT at positive Δ because there is no second-order discontinuity in that region. The minimum of λ_{SSB} is *not* associated with the second-order critical point. The lifetime of the metastable states $\rho_{\pm\alpha}$ can be longlived (λ_{SSB} small) even across a first-order transition.

As the authors explained in the rebuttal letter, $\lambda_{1\text{st}}$ and λ_{SSB} are the minimum non-zero eigenvalues of different symmetry sectors corresponding to the order parameter ± 1 , respectively. But a Liouvillian gap, as defined by the minimum non-zero eigenvalue of the whole matrix, should be the minimum of them, i.e., $\min\{|\lambda_{1\text{st}}|, |\lambda_{\text{SSB}}|\}$. Naming both of them as the Liouvillian gap is rather confusing to me. It is acceptable if this is the convention, otherwise I would suggest renaming them.

It is the convention to call these two quantities Liouvillian gaps. See for example : Phys. Rev. A 98, 042118 (2018), arXiv:2406.19381, Phys. Rev. Lett. 126, 160401 (2021). We nevertheless tried to make

this distinction clear by calling them λ_{SSB} and $\lambda_{1\text{st}}$.

** Fig. 2d. I understand from the rebuttal letter that the units of the x- and y-axes are not $\sqrt{\text{photon number}}$. I recommend add the “arb. units” label here and in Figs. 4a and 5d.*

We apologize to the Reviewer for the confusion. The data we present in the x- and y-axis and in Figs. 4a and 5d are the rescaled field quadratures convoluted with the amplifier noise (see supplementary section F). Consequently, they are unitless quantities, expressed in terms of the number $\sqrt{\text{photon number}}$

This quantity is unitless since it counts the number of photons from $E/\hbar\omega$, where E is an energy (Joule) and $\hbar\omega$ the energy (Joule) of a single photon. This is analogous to plots of $\langle a^\dagger a \rangle$ or $\langle a \rangle$ which typically do not have units.

** Fig. 5e. In my understanding, the 1st-order DPT exists inside the symmetry-breaking region. Then, ρ_α and $\rho_{-\alpha}$ would also have a very long lifetime. How is ρ_{ss} distinguished from the metastable states ρ_α and $\rho_{-\alpha}$? I am thinking that looking only at $\langle a^\dagger a \rangle$ may end up in ρ_α and $\rho_{-\alpha}$ instead of ρ_{ss} . Will it mix up $\lambda_{1\text{st}}$ and λ_{SSB} ? What does $\langle a \rangle$ look like?*

Yes, the Reviewer is correct that the 1st-order DPT exists inside the symmetry-broken region. Nonetheless, the two decay rates $\lambda_{1\text{st}}$ and λ_{SSB} can be extrapolated independently.

To extrapolate $\lambda_{1\text{st}}$ we measure how $\hat{a}^\dagger \hat{a}$ evolves in time. Notice that $\text{Tr}(\hat{a}^\dagger \hat{a} \rho_\alpha) = \text{Tr}(\hat{a}^\dagger \hat{a} \rho_{-\alpha})$, and consequently, the dynamics of $\langle \hat{a}^\dagger \hat{a}(t) \rangle$ does not show signatures of λ_{SSB} . Measuring it therefore allows us to directly determine $\lambda_{1\text{st}}$ without seeing the influence of λ_{SSB} .

As detailed above, $\langle \hat{a} \rangle$ is always zero. A state initialized in $\rho_{\pm\alpha}$ will instead evolve with the same decay rate as calculated from the autocorrelation function.

In the last review round, I have asked whether the gap $\lambda_{1\text{st}}$ looks wide. In the rebuttal letter, the authors provide a zoom out and indicate that it is indeed a narrow region. I am now wondering if it should be wide? I am considering that the symmetry-breaking region is wide with a small eigenvalue.

We show again the previous figure for completeness.

We disagree with the Reviewer that the region of closure of $\lambda_{1\text{st}}$ is narrow. The region of closure of $\lambda_{1\text{st}}$ is very wide for positive detuning as is shown below and in the main text. We recall that the first-order DPT occurs near $\tilde{\Delta} \sim 0.13$ MHz and the region of closure of $\lambda_{1\text{st}}$ extends to $\tilde{\Delta} \sim 2.5$ MHz. This is perfectly consistent with our simulation and with the hysteresis measurement shown in Fig. 6, which shows a large hysteresis region for the up-sweep and a smaller one for the down-sweep. This is however not linked to the eigenvalue of the symmetry-broken region λ_{SSB} since this plot only shows $\lambda_{1\text{st}}$, a different quantity.

As detailed above, in our system, measuring the rate of SSB in the region of metastability is extremely difficult since $\lambda_{1\text{st}} > \lambda_{\text{SSB}}$ in that region. This implies that when initialized in $\rho_{\pm\alpha}$, the system will typically decay to the vacuum before a phase flip event occurs. Furthermore, measuring this quantity would not provide additional information on the scaling of the minimum of λ_{SSB} .

*** Fig. 6.** *I do not understand this claim: “we do not observe hysteresis in the region where $|\lambda_{\text{SSB}}| \ll |\lambda_{1\text{st}}|$.” In my understanding, observation of hysteresis depends on the sweeping time, T , and the gap, $\lambda_{1\text{st}}$. The condition $|\lambda_{\text{SSB}}| \ll |\lambda_{1\text{st}}|$ should not exclude the existence of hysteresis for a small T . Do the authors mean a very small $|\lambda_{\text{SSB}}|$ but a very large $|\lambda_{1\text{st}}|$?*

Yes the Reviewer is correct, we have checked that $|\lambda_{\text{SSB}}| \ll 1$, but a very large $|\lambda_{1\text{st}}|$ does not lead to hysteresis. In other words, this complements the Referee’s question above: there are effects dependent of the number of photons, such as hysteresis, which only reveal the presence of $|\lambda_{1\text{st}}|$ and not $|\lambda_{\text{SSB}}|$, making them decorrelated.

*** Reply to Reviewer 2 ***

In the revised version of the manuscript, the authors have addressed both my major and minor concerns while improving the presentation of their results. In particular, they provide a more systematic treatment of experimental errors and clarify the relation between their results and the expectations of Liouvillian theory.

We thank the Reviewer for helping to improve the quality of the manuscript through their review and for supporting its publication in Nature Communications.

*** Reply to Reviewer 3 ***

I believe that this work has a notable gap between the presented data and the interpretation offered by the authors. The quality of the data is relatively weak, the signals of interest are elusive, and the authors tend to make very strong claims. As a consequence, the experiment does not support the claims made. I think the authors believe too much in their theory and see in their data what they want to see. I do not find the work convincing. The work itself is valuable as it sets a precedent for more experiments in an original research direction. If it is toned down further, it deserves publication. In its current form, I just find it misleading, claiming an experimental real state that is out of bounds for the setup.

Below, I review the point on which I based my judgment.

We thank the Reviewer for their thorough reading of our revised manuscript. We apologize if we misunderstood some of the initial comments and did not address them adequately in the rebuttal letter. We appreciate the further clarification provided by the Reviewer regarding their concerns and a second opportunity to address them.

We, of course, want to avoid any overrepresentation and provide an analysis that accurately reflects

the collected data. As such, we have reviewed the manuscript extensively. Following the Reviewer's comments, we have removed the section titled "Quantum interpretation of the transition" and moved any discussion on the *possible* quantum nature of the transitions to the conclusion as an outlook for future experiments which could strengthen these claims. In the outlook, we discuss with nuance some experimental and theoretical observations that *hint* at the possibility that certain quantum processes *could* play a role in this device. We also contrast these observations with previous classical experiments, provide further references, and clearly state that while we find good agreement with our quantum model, we cannot entirely rule out a classical explanation for our observations.

We hope that the reply below, along with the updated version of the manuscript, will convince the reviewer that our work is honest, sufficiently valuable and novel to recommend publication in Nature Communications.

The experiment is done in the classical regime

As far as I am concerned, the experiment is done in an entirely classical regime. In particular, I completely disagree with Referee 2 (and with the authors who agree with the referee here) that "The observed quantum jumps (...) constitute a clear observation of the significant role of quantum fluctuations", and they have a "pivotal role". Please see Figure 2e in PhysRevLett.123.254102. These jumps are entirely classical and entirely analogous to the jumps observed here by Beaulieu et al. In fact, I propose to replace the words "jump" and "quantum jump" in the manuscript with "hop" or "classical jump" to avoid the confusion that Referee 2 was led into.

I find this misconception, purposefully induced by the article, entirely unacceptable.

In general, the authors have not provided convincing experimental evidence that what they observed will not be observed too in the setup of PhysRevLett.123.254102. This is a question beyond the theoretical model.

The theory analysis is indeed based on quantum trajectories and jumps. The experiment does not require it or demonstrate at all its need.

We apologize for any misunderstanding. We agree with the Reviewer that it is a well-known fact that jumps can occur in classical nonlinear parametric resonators, and we never intended to claim otherwise. To avoid any confusion regarding this point, we now state this in our outlook and provide further classical references to illustrate this. We also provide further clarification below.

We contacted some of the authors of PhysRevLett.123.254102 for extra clarification. They disagree with the Reviewer that the observed jumps are completely analogous to the one we present in our manuscript. The experiment conducted in PhysRevLett.123.254102 and shown in Fig.2e differs *fundamentally* from the one presented in our manuscript. In the cited paper, the jumping events are not random, but are periodically and deliberately induced by deforming the potential. Specifically, Frimmer et al. confine the parametron to one of the two stable phase states using a parametric drive, then abruptly switch the natural oscillation frequency to $f_0/2$ for a time t_{def} . Choosing this time $t_{\text{def}} = 1/f_0$ results in the particle moving to the other stable state, resulting in a phase flip of π when the parametric drive is reinstated. In our experiment, *we do not apply* such a protocol; there is no abrupt change in the fundamental frequency or any other method to deform the potential. Instead, the system undergoes random and spontaneous jumps between the two states that emerge when the Kerr resonator is subjected to a parametric drive. For these reasons, we fail to see how this is "entirely analogous" to the jumps we observe. Furthermore, given the different nature of the study, Frimmer et al. do not discuss or present any graph of the switching rate, as it is fixed by $1/f_0$ in their protocol (see Fig.2e). In fact, their observation of only periodic jumps, along with their control of the switching probability through t_{def} and the rate through f_0 , provides direct evidence that they are reporting a different effect.

We nevertheless understand the Reviewer's point. We are aware of existing entirely classical and more closely related experiment, e.g., Appl. Phys. Lett. 121, 164101 (2022), conducted by many of the

same authors as the paper cited by the Reviewer. The authors implement a classical Kerr Parametric Oscillator (KPO) by parametrically driving a nonlinear micro-electromechanical resonator at room temperature. They demonstrate that by *voluntarily introducing white electrical noise*, the state of the resonator fluctuates around its initial solutions, causing it to switch between the two attractors (see Fig. 1c). This reference along with others have been added when discussing the occurrence of similar phenomena in entirely classical settings in the outlook.

Concerning, the use of the term “jump”. The term “jump” is commonly used in numerous articles studying parametrically driven classical systems across various platforms. For example, in the article cited above (Appl. Phys. Lett. 121, 164101 (2022)), the authors state, “At this point, the resonator jumps to a positive or negative response with equal probability.” This use of the word “jump” is consistent with how we use it. Additional references which use “jump” in the same way as we do in fully classical settings include Phys. Rev. Lett. 99, 060601; Phys. Rev. E 99, 062205 (2019); and Appl. Phys. Lett. 112, 233105 (2018). From the current literature, it appears that the term “jump” does not carry any quantum connotation unless paired with “quantum jump”, a term that does not appear in the manuscript and was never used by us.

While the term “jump” is frequently used in classical settings to describe the dynamic of oscillator, the term “hop” is more abundant in the study of quantum systems (i.e hopping of particles between lattice sites), but is not commonly found in the literature on oscillator. As for “classical jump”, we are uncomfortable using this term, as it is in direct opposition to the conclusions of many studies, such as “Quantum behavior of the Duffing oscillator at the dissipative phase” (**Nature Communications** volume 14, Article number: 2896 (2023)), which claims that a quantum description is necessary to correctly interpret the first-order DPT in a *single photon driven* Kerr resonator with $U < \kappa$. We believe it is more honest to use neutral terms in the main text and present references for both classical and quantum interpretations in the outlook.

b) In their reply to my first review, the authors state that “As Heisenberg scaling cannot be obtained using classical resources, we argue that some quantum effect must be present around the critical point, despite the very dissipative nature of the system”.

However, the paper does not contain any equivalent explanation or justification. The paper does not demonstrate any “metrological advantage” nor mentions that “the Heisenberg scaling” is a signature that (some) “quantum mechanics” must be present.

The word “some” used in the author’s private reply to me seems to be much more adequate and honest in spirit. Still, I do not believe the toned-down argument either and I still wonder what do the authors really mean by “quantum” (I explicitly asked for a clarification on this in my first review and I do not find their answer).

Our statement about Heisenberg scaling was intended as an additional theoretical support for evidence that some quantum properties *should* be present in the current system, even in the regime where $U < \kappa$ (npj Quantum Information 9, 23 (2023)). As correctly pointed out by the Reviewer, we only have theoretical support for this (npj Quantum Information 9, 23 (2023)) and do not demonstrate this in the manuscript. As such, we have adjusted the text to remove claims regarding the demonstration of quantum effects in the manuscript. We now only mention the *potential* quantum nature of the device in the conclusion as a perspective for future more refined experiments which might verify this for our setup. We believe that this outlook is justified given the extended literature on this topic.

We apologize to the Reviewer for not providing sufficient answer in our first rebuttal letter. Our system is “more quantum” than the classical one cited above because the operating temperature (~ 10 mK) is much lower than the photon energy of the signal frequency $k_B T \ll \hbar\omega$ and the device is properly shielded and the lines filtered and attenuated such that thermal and classical noise sources should be negligible relative to quantum noise. By quantum noise and quantum fluctuations, we mean the effects that require some order of $\hbar \neq 0$ to be observed. In the context of bosonic quantum systems, these can be formally investigated by considering a formal expansion using the Moyal brackets. The lowest-order Moyal bracket is the coherent state approximation, that predicts no fluctuations in the dynamics of the system. Higher-order corrections to \hbar include, for instance, noise induced by photon

loss, dephasing, and Kerr nonlinearity. A discussion of some of these issues is given in New J. Phys. 26 023022 (2024). Following this formal approach to the classical limit, if in our system we assume $\hbar \rightarrow 0$, no quadrature can achieve a variance below $1/2$ (no squeezing below vacuum).

It is well known that the conditions mentioned above under which these effects become visible are typically achieved using the cQED platform with both linear resonator (Nature volume 454, pages 310–314 (2008); Nature volume 459, pages 546–549 (2009)) and nonlinear resonator with $U < \kappa$ (Nature volume 479, pages 376–379; Appl. Phys. Lett. 91, 083509; Phys. Rev. B 83, 134501), allowing for the observation of quantum effects such as Fock state generation, quantum-limited amplification, and the dynamical Casimir effect (to name a few). Since we only provide squeezing below vacuum as evidence that our setup is governed by quantum fluctuations, we limit the discussion of the quantum nature of the experiment to the outlook section.

Quantum fluctuations are related to the circuit properties (it's "mass"), and one needs quantum-limited amplification to directly observe these quantum fluctuations. The experiment does not have a quantum-limited amplifier and, therefore, needs to subtract extra noise in post-processing. This is not an elegant experimental technique. This is why quantum-limited amplifiers are important. They are hardly optional if the aim is to demonstrate the purely quantum nature of an observation.

Noise removal in post-processing can lead to all kinds of experimental artifacts, in particular, an overestimation of vacuum fluctuations. I judge that there is no evidence in the paper to decide if this calibration was made with sufficient care. If the fluctuation obtained after noise subtraction (a very delicate and error-prone process) contains classical noise, it is easy to reduce the fluctuations with squeezing without ever having witnessed any quantum mechanics. This is related to the first referees' concerns about the calibration of the quadrature measurement. (See referee one comment 19) At no time is it made clear (by the data) that " $(\Delta x)(\Delta p)=1/4$ ", showing the measurements reach the Gaussian quantum noise limit. How was this uncertainty relation calibrated?

To put it in simpler language, the authors have not proven that the fluctuations they see are quantum in origin (even without any squeezing involved). In fact, I do not believe they are.

We agree with the Reviewer that quantum fluctuations can be *directly* observed with a quantum-limited *parametric* amplifier, such as a degenerated JPA. However, for our claims -where only expectation values are needed- directly accessing quantum fluctuations is not necessary. In this case, using a phase-insensitive amplifier combined with a noise-removal technique is sufficient to derive the expectation values. In our case, using quantum-limited parametric amplifiers would improve the efficiency of the experiment by reducing the number of measurements (or the time required) to estimate the expectation values of the quadrature and their moments, but they are not essential to our approach. In other words, the use of quantum-limited parametric amplifiers constitutes a more advanced and efficient measurement technique than the one used in our experiment, but it is not needed at all.

For instance, it is well documented that squeezing below vacuum can be observed without a quantum-limited amplifier by performing the noise-removal procedure as done in our experiment (PRL 107, 113601 (2011), Phys. Rev. Lett. 65, 1419; L. Zhong et al. 2013 New J. Phys. 15, 125013).

Regarding the noise-removal technique, we are afraid that the technique based on a quantum-limited parametric amplifier is also not shielded from gain calibration, see e.g., Phys. Rev. Lett. 106, 220502 (2011). Moreover, since we are interested in the state *inside* the resonator and a parametric amplifier would not be perfectly quantum-limited (i.e., it would add some noise), this method would still require noise-removal to reconstruct observables of our interest, see Phys. Rev. Lett. 106, 220502 (2011).

Regarding the robustness, the noise removal process used in the paper is standard and well-documented in the literature (e.g., Phys. Rev. A 86, 032106 (2012); New J. Phys. 16, 015001 (2014); Phys. Rev. Lett. 105, 133601), and is solely based on vacuum calibration. A very similar noise removal process and calibration was used in the article [Nature Communications volume 14, Article number: 2896 (2023)] recently published in this journal. Furthermore, the noise removal and calibration procedure are explained in great detail in the supplementary section F and E. We have now added more emphasis on this point in the main text. Furthermore, one of the co-authors has a strong publishing background

on papers concerning such calibration (e.g., Phys. Rev. Lett. 109, 250502; New J. Phys. 15, 125013, Science advances 7 (52), eabk0891 (2021), Physical Review Letters 117, 020502 (2016), Scientific Reports volume 8, Article number: 6416 (2018)). Finally, with our further clarification on the noise removal process the Reviewer 1 now clearly state that he is "not very skeptical on the claim "squeezing below vacuum". Consequently, we do not understand on which basis the Reviewer judged that there is no evidence that the calibration is correct.

In our analysis, measuring the variance of the anti-squeezed quadrature can still be used as a sanity check for the state to be physical, by checking the Heisenberg relation, which we did. We show here the results for the antisqueezed quadrature for $L = 1$. We define, as in the main text, the quadratures $x = \frac{a+a^\dagger}{\sqrt{2}}$ and $p = \frac{a-a^\dagger}{\sqrt{2}}$, such that the minimum of $\Delta x^2 = \Delta p^2 = \frac{1}{2}$. In the top panel of the figure below, we show that the product of the squeezed (labeled Δx^2) and anti squeezed (labeled Δp^2) quadratures variances follows a clear trend. When the detuning is highly negative, the state is in the vacuum and the heisenberg inequality is saturated $\Delta x^2 \Delta p^2 = 1/4$. When the detuning approach the second-order critical point, the state becomes mixed, resulting $\Delta x^2 \Delta p^2 > 1/4$. This is reflected in the very large variance along the antisqueezed quadrature. In the lower panel, we show the variance of the squeezed (black line) and antisqueezed quadratures (red line). A zoom is provided in the inset with the horizontal black line indicating 0.5. The detuning corresponding to the maximum $\partial^2 n_{ss}$ is shown by the grey dotted line. This figure was added in the supplementary material.

Finally, we want to highlight that the most recognized example of a quantum-limited amplifier, is the Josephson Parametric Amplifier (JPA). A JPA generally consists of a SQUID-shunted resonators operating in the regime $U \ll \kappa$. A JPA based on 4-wave mixing is not only in the *same regime* as our device, but also has the *exact same arrangement and operates in the same manner* (a flux pump at $2\omega_r$ and an output line connected to an HEMT). For a concrete example, consider Fig.1 a of Appl. Phys. Lett. 93, 042510 (2008) which shows exactly the same schematic as in Fig. S15. At the second-order point, our device is essentially a JPA without an input signal, a configuration well known to squeeze the vacuum when brought near the parametric threshold (New J. Phys. 15, 125013, Low Temp. Phys. 45, 848–869 (2019)).

I find uncomfortable the Wigner plots they now show as "experimental reconstruction". The plots are clearly not a tomographic reconstruction. Tomographic reconstruction is a usual technique nowadays but the experiment in question does not have the possibility to measure it. In the field of quantum

optics and quantum circuits “Wigner reconstruction” means full tomography. It is disappointing to read “Wigner reconstruction” and find the analysis made.

I think the Wigner plots shown are some sort of theory fit to some partial data, but they reflect the full phase-space distribution. Therefore, the “reconstruction” conveys much more information than the information the authors have experimental access to. It is, yet again, a way to overrepresent the experimental data.

The Reviewer his correct that the plot do not show a full experimental tomographic measurement and that the current experiment does not have the possibility to measure it.

The Wigner plots were added to the revised manuscript based on recommendation of Reviewer 1. We followed the procedure clearly explained in the paper cited by Reviewer 1 [Nat. Commun. 14, 2896 (2023) Supplementary Note 4, paragraph C]. To make sure the data are not overrepresented, we adapted the title of the supplementary figure to “Theoretical Wigner functions.” Similarly, the caption and places where this figure is referenced in the manuscript have been revised. We have no issue removing this figure from the supplementary materials if Reviewer 1 agrees.

The data does not show discontinuities

a) The authors insist on the “discontinuous jump” in Fig. 2(b) and the “a discontinuous jump at positive detuning can be observed” and the “continuous but non-differentiable change in photon number at negative detuning” in Fig. 3(c). I see no discontinuities or nondifferentiabilities in the data, because there are none. Sporadically the authors tone down the claim by introducing the word “emergence”. This word should be systematically used and in the title of the paper.

It is true that the observation the authors are after is very (very) hard to measure experimentally and the word “emergence” is sufficiently ambiguous to be tolerated. The terrible experimental challenge faced is why this observation has not been reported directly in previous literature (to my knowledge) and I judge it has not yet been overcome yet. To do so, a highly refined experiment will be needed.

b) Regarding Figures 2b and c. It is understood that here there are no discontinuities but only the “emergence of a discontinuity.” This is fine. As we already know from theory that in some limit ($L \rightarrow \infty$) there should be a discontinuity, we theoretically interpret this set of experimental points following over a smooth curve as the “emergence”.

We agree with the Reviewer that the data only reflect the emergence of a discontinuity, as a true discontinuity occurs only in the limit of $L \rightarrow \infty$. The title as been adjusted to “Observation of the emergence of first- and second-order dissipative phase transitions in a two-photon driven Kerr resonator”.

Here we have theory speaking for the data, but it is reasonable and acceptable as $L > 2$ is hard to handle experimentally as the authors clearly explain. Now, where do we see this is caused by unambiguous quantum effects? Where do we see this is caused by κ_2 ? I believe that introducing a phenomenological κ_2 to the simpler model makes the effect emerge. But what about the alternatives? What about also including phenomenologically κ_4 ? Why not? or Hamiltonian terms other than Kerr which naturally become relevant at higher amplitudes? Or just broad-band noise? (All of this, including κ_2 , admits a classical explanation unless several caveats are addressed experimentally and theoretically).

As noted by the Reviewer, we do not provide “smoking-gun” evidence that quantum fluctuations are responsible for driving the observed switching rate in our experiment or that no other classical noise sources are coupled to the system. As such, we have removed the section on the quantum nature of the transition and provide a nuanced discussion on the potential quantum effect in the the outlook.

We agree with the Reviewer that higher-order terms could indeed play a relevant role when discussing the emergence of the first-order transition. Nonetheless, our numerical simulations follow the experimental data up to $\tilde{\Delta}/2\pi \simeq 1\text{MHz}$, where the photon number in the metastable manifold is of order 100 (see, e.g., the characterization of λ_{1st} and the corresponding hysteresis data allowing extracting

the photon number). Since we observe no deviation from the theoretical curve, and higher-order terms are expected to become more relevant at larger photon numbers, our data do not allow for fitting these terms. We tried including these extra terms, but given our data the fitting procedure didn't converge to meaningful values of higher order nonlinearity. Nevertheless, we cannot exclude the possibility that they exist.

In other words, I would not use this type of experiment to calibrate a kappa2 in a setup (as the authors propose). It is too indirect evidence of the existence of a two-photon dissipation, and the effect could be caused by anything else. Furthermore, the authors do not spend any effort justifying kappa2 microscopically for their flux-driven SQUID. Microscopically, kappa2 can be generated by noise in the flux drive at the two-photon drive frequency. What noise-temperature do the authors have in their drive at the sample level at the drive frequency? Why they do not consider D[adag^2] too? There are thermal photons at 4GHz (n_th) but not at 8GHz? Is this temperature roughly compatible with the kappa2 inferred from the model applied to the data?

We agree with the Reviewer that we cannot exclude different origins for κ_2 , and we do not provide a microscopic explanation for this term. Consequently, we have removed the discussion on the calibration of κ_2 from the manuscript and its quantum origin. Two-photon dissipation is introduced as a phenomenological term which is necessary to accurately capture the experimental results using our model.

As for the presence of terms $D[a^\dagger^2]$, the presence of thermal photons at twice the frequency could result in this injection, and naturally, there are some thermal photons at any frequency. However, we do not observe any effects that would allow us to calibrate these terms. According to our simulation, the effect of κ_2 is to reduce the rate of jump associated with λ_{SSB} (fewer passages from $\alpha \rightarrow -\alpha$), while thermal photons injected one- and two-at-time would increase λ_{SSB} . However, the interplay of this term with other unaccounted terms cannot be excluded.

c) Regarding Figure 3c (Where are the error bars? Where is the theory curve for the derivatives?) Indeed, the authors are after a quantity that is very elusive experimentally. The data needs to be presented with more touch or a better experiment needs to be designed. Let me make this clear by the following example. Consider the function $f(x)$ which is $f(x) = 0$ if $x < 0$ and $f(x) = x^2$ if $x \geq 0$. This function is continuous, its first derivative is continuous, and its second derivative is discontinuous. Consider now a signal consisting of this function plus a very small amount of noise and compute numerically its derivatives. I have done this in the plot below (the Python code to generate this figure is attached at the end of the report).

Just like the author's true experimental data, the blue numerical data I created has very good signal-to-noise. But what can one say, from the data (dots), about the second derivative? As the authors explain to referee one (comment 18), they can smooth the noise in the calculation of the second derivative (taking a midpoint derivative) to see the "peak," but this average is a poor experimental technique (what are the error bars?). I do not think the information about the "continuity" is possibly there.

I think the "peak" in Figure 3c is essentially noise, an artifact of the freedom in choosing which midpoint derivative using many points. Exploiting that freedom, the plot could have been made to look very different. I believe it looks like a peak because that is what the authors are expecting to see from theory. I, as a reader, yet ignore completely what an unbiased analysis of the experimental data revealed to the authors in their exploration.

We thank the Reviewer for providing a clear example to illustrate their point and apologize for the confusion caused by our response to Reviewer 1. The derivative calculation is performed using a finite-difference method over two points. Specifically, we first calculate the first derivative as follows:

$$\partial \tilde{n}_{\text{ss}}(\tilde{\Delta}_i) = \frac{\tilde{n}_{\text{ss}}(\tilde{\Delta}_{i+1}) - \tilde{n}_{\text{ss}}(\tilde{\Delta}_i)}{\tilde{\Delta}_{i+1} - \tilde{\Delta}_i}, \quad (11)$$

where \tilde{n}_{ss} is the rescaled steady state photon number, $\tilde{\Delta}$ the rescaled detuning. The second derivative

is then given by :

$$\partial^2 \tilde{n}_{ss}(\tilde{\Delta}_i) = \frac{\partial \tilde{n}_{ss}(\tilde{\Delta}_{i+1}) - \partial \tilde{n}_{ss}(\tilde{\Delta}_i)}{\tilde{\Delta}_{i+1} - \tilde{\Delta}_i}. \quad (12)$$

However, before performing these calculations, we first average over multiple points to reduce the noise on the calculation of the first and second derivative. Mathematically, we are performing a convolution of a sharp function with a smoother one, making it impossible to increase the sharpness of the feature by averaging. This is the reason why we collected so many points and decided to average for different values of detuning: to be sure to observe a feature and not spurious data.

We can demonstrate the validity of our approach using the code provided by the Reviewer. First, the numerical calculation of the second-order derivative follows the theory curve. More importantly, the second-derivative will consistently display a peak corresponding to the sampled curve, with increasing the number of averages simply smoothing the peak.. The effect of our procedure for the function $f(x) = 0$ if $x \leq 0$ and $f(x) = x^2$ if $x > 0$ is shown in all the vertical left panels in the figure below. To clearly highlight the effect of averaging, we have increased the number of points and, for each panel, show an increasing number of averages. In all the vertical right panels, we have also repeated this analysis for the function $f(x) = 0$ if $x \leq 0$ and $f(x) = x$ if $x > 0$, which has a peak in the second derivative similar to our experiment. Our procedure accurately captures the trend of the second derivative, and increasing the number of points averaged simply results in a smoother peak. Over averaging, removes any peak.

Furthermore, we also show below the full unbiased analysis of the experimental data by plotting the

photon number \tilde{n}_{ss} (first column), first derivative $\partial_{\Delta}\tilde{n}_{ss}$ (second column) and second derivative $\partial_{\Delta}^2\tilde{n}_{ss}$ below. The observed behavior is similar to the test function treated above. The python code to generate these figures and the ones above are provided with this report. We have added this figure in the supplementary material with more explanation on the calculation of the derivative so that the reader can be aware of the complete unbiased analysis of the data.

Concerning the theory curve, the example for the function $f(x) = 0$ if $x \leq 0$ and $f(x) = x$ if $x > 0$ clearly demonstrates that, although the averaging process correctly recovers a peak, the amplitude and shape of the peak are significantly smoothed, making direct comparison difficult. This is also valid for the error bars, as adding statistical error to a significantly reduced peak would not be meaningful. Nevertheless, the trend of the derivatives remains correct.

How is “the dephasing rate mainly due to the nonlinearity of the resonator”? Isn’t it flux noise?

Yes, the Reviewer is correct that dephasing rate is due to flux noise. We have clarified this by changing the sentence to “dephasing rate mainly due to flux noise”. The flux noise is mainly due to noise on the nonlinear element affecting the system’s frequency, and our previous statement was assuming this connection between the two phenomena.

-Why is the “five order of magnitude slowdown” a demonstration of quantum criticality? What would one expect for a classical parametric oscillator? Isn’t the strong slowdown a well-known fact? This slowdown has been observed in classical and quantum systems already (citations of this scaling shine by their absence. Quantum dissipative cats dominated by kappa2 are a bad example for this work, where kappa2 is very small and the two bright state is due to Kerr)

We apologize for any confusion, but in the manuscript we claim that the system is critical (not “quantum critical”), as quantum criticality often means that the element triggering criticality is the

non-commutative nature of the Hamiltonian terms. Quantum phase transitions and quantum critical points need not to be related with the phenomenology of dissipative phase transition.

That being said, the Reviewer is correct that critical slowing is expected also for a classical parametric oscillator (see for i.e. Appl. Phys. Lett. 121, 164101 (2022)). We have added further references for classical systems in the outlook.

Concerning citations, we provide several, including those on quantum nonlinear resonator based on superconducting circuits. For instance, for a single photon driven Kerr resonator critical slowing down near the 1st-order was shown in Sci. Adv.7,eabe9492(2021) and Nature Communications volume 14, Article 7 number: 2896 (2023). However, this has never been demonstrated with a formal scaling and model for both the first- and *second-order* critical points in a *parametrically* driven quantum nonlinear resonator. These papers studying DPTs in a quantum setting, *along with others*, are already cited in the manuscript. However, we are certainly open to adding additional references if the Reviewer has any suggestions.

-“Our work impacts all major quantum technological platforms.” I think the claims about quantum technology and quantum information processing are all unwarranted and beyond the scope of this setup. The experiment is not close to a regime that would be useful for quantum information. Turning this experiment into a quantum information experiment will be a mayor task and it is unfair to extrapolate this observations to that setting. That defeats the point of experimental science.

Following the Reviewer’s suggestion, we have removed this statement from the manuscript.

-The words “establish” or “demonstrated” are too strong, and the data cannot back them up. Maybe “seen signatures of” would be better.

We have revised this section according to the Reviewer’s suggestion.

-“Quantum fluctuations and quantum dissipative processes could be the main drives of the observed transitions’. This is a better statement but it contradicts the statements made before. “Could be” or “demonstrably is”? “They show how quantum processes could trigger DPTs, highlighting the necessity to study DPTs within a quantum framework.” I disagree with the necessity. It contradicts the previous statement. “Could” or “necessitates”?

We revised this section based on the Reviewer’s comment.